# Blebbisomes are large, organelle-rich extracellular vesicles with cell-like properties

Dennis K. Jeppesen [1,5] ✉, Zachary C. Sanchez [2,5], Noah M. Kelley [2], James B. Hayes[2], Jessica Ambroise[2], Emma N. Koory[2], Evan Krystofiak [2], Nilay Taneja [2], Qin Zhang[1], Matthew M. Dungan[3], Olivia L. Perkins[2], Matthew J. Tyska [2], Ela W. Knapik[1,2], Kevin M. Dean [4], Amanda C. Doran[3], Robert J. Coffey [1,2] & Dylan T. Burnette [2] ✉

Cells secrete a large variety of extracellular vesicles (EVs) to engage in cell-to-cell and cell-to-environment intercellular communication. EVs are functionally involved in many physiological and pathological processes by interacting with cells that facilitate transfer of proteins, lipids and genetic information. However, our knowledge of EVs is incomplete. Here we show that cells actively release exceptionally large (up to 20 μm) membrane-enclosed vesicles that exhibit active blebbing behavior, and we, therefore, have termed them blebbisomes. Blebbisomes contain an array of cellular organelles that include functional mitochondria and multivesicular endosomes, yet lack a definable nucleus. We show that blebbisomes can both secrete and internalize exosomes and microvesicles. Blebbisomes are released from normal and cancer cells, can be observed by direct imaging of cancer cells in vivo and are present in normal bone marrow. We demonstrate that cancer-derived blebbisomes contain a plethora of inhibitory immune checkpoint proteins, including PD-L1, PD-L2, B7-H3, VISTA, PVR and HLA-E. These data identify a very large, organelle-containing functional EV that act as cell-autonomous mobile communication centres capable of integrating and responding to signals in the extracellular environment.

Cells release a variety of 30- to 10,000-nm lipid-bilayer-enclosed extracellular vesicles (EVs) to facilitate cell-to-cell and cell-to-environment communication by packaging signalling molecules to avoid degradation[1–5] and escape immune surveillance[6–9]. EVs may interact with target cells through contact between molecules on the EV surface with receptors on the cell surface to relay signals. In addition, modulation of recipient cell behavior may follow uptake of EVs cargo, including bioactive proteins, lipids and nucleic acids. EVs have emerged as important actors and agents of intercellular communication in normal cell biology and pathological conditions[2,4,6].

Here, we identify blebbisomes, an exceptionally large functional EVs, that are actively released by human and mouse cells, remain motile independently of cells and have the capacity to both take up EVs and secrete exosomes and microvesicles. Blebbisomes are the largest type of EV described so far with an average diameter of 10 μm but can be as large as 20 μm, with an area commonly larger than 50 μm². After being released from motile cells, blebbisomes display marked

[1]Department of Medicine, Vanderbilt University Medical Center, Nashville, TN, USA. [2]Department of Cell and Developmental Biology, Vanderbilt University School of Medicine Basic Sciences, Nashville, TN, USA. [3]Department of Molecular Pathology and Immunology, Vanderbilt University Medical Center, Nashville, TN, USA. [4]Lydia Hill Department of Bioinformatics, University of Texas Southwestern, Dallas, TX, USA. [5]These authors contributed equally: Dennis K. Jeppesen, Zachary C. Sanchez. ✉e-mail: dennis.k.jeppesen@vumc.org; dylan.burnette@vanderbilt.edu

**Fig. 1 | Blebbisomes form with functional mitochondria. a**, Representative blebbisomes imaged with DIC microscopy. The arrowheads show blebs. **b**, The diameter of blebbisomes. Average blebbisome diameter: 14.9 ± 3.0 s.e.m., 8.92 ± 0.8 s.e.m., 10.9 ± 1.2 s.e.m. in B16-F1, DKO-1 and MDA-MB-231 cells, respectively. $n$ = 167, 183 and 236 blebbisomes for B16-F1, DKO-1 and MDA-MB-231 cells, respectively. **c**, An SEM micrograph of B16-F1 blebbisome displaying large characteristic bleb (arrowhead). **d**, Correlative light and electron microscopy of a B16-F1 blebbisome using iSIM (top and middle) and SEM (bottom). The colour bar denotes the relative $Z$ height, and the dotted line denotes the height of the single $z$-slice. The arrowheads show the blebs. **e**, A timelapse DIC and epifluorescence (MitoTracker) microscopy time montage showing blebbisome formation. The arrow denotes the blebbisome formation, and the arrowhead denotes mitochondria. **f**, Imaging of actin filaments and mitochondria in blebbisomes by iSIM. The boxes indicate blebbisomes. Bottom: enlarged boxes are shown below ($n$ = 3). **g**, TMRE fluorescence before and after FCCP treatment. The arrow shows a cell, and the arrowhead shows a blebbisome ($n$ = 5). **h**, A timelapse of

TMRE fluorescence before and after FCCP treatment. The fluorescence levels between the cells and blebbisomes after FCCP treatment were not significantly different as determined by a two-tailed Student's $t$-test. For DKO-1 post FCCP treatment, blebbisome mean: 0.42 ± 0.46 s.e.m., cell mean: 0.35 ± 0.46 s.e.m. For B16-F1 post FCCP treatment, blebbisome mean: 0.55 ± 0.058 s.e.m, cell mean: 0.47 ± 0.058 s.e.m ($n$ = 5 cells and blebbisomes each for both DKO-1 and B16-F1 cells). **i**, Kaplan–Meier survival curves for blebbisomes treated with DMSO, FCCP, staurosporine or raptinal. Time of death was denoted by a loss of membrane integrity (Supplementary Video 7). The mean survival time for DMSO, FCCP, staurosporine and raptinal are 23.18 ± 1.23 s.e.m., 19.59 ± 1.57 s.e.m., 20.36 ± 1.46 s.e.m. and 20.46 ± 1.68 s.e.m., respectively. DMSO was significantly different from FCCP, staurosporine and raptinal with a $P$ value of 2.3 × 10⁻⁵, 0.00041 and 0.00040, respectively, as determined by a two-tailed Student's $t$-test ($n$ = 29 for DMSO, 41 for FCCP, 29 for staurosporine and 39 for raptinal; across three independent experiments).

contractility-dependent 'blebbing' behaviour. Both normal and cancer cells release blebbisomes that contain active, healthy, mitochondria further distinguishing them from other large EVs (lEVs) such as exophers[10,11] and migrasomes[12] that function in the removal of damaged mitochondria from cells under stress conditions. In addition, blebbisomes contain many other cellular organelles including endoplasmic reticulum (ER), Golgi apparatus, ribosomes, lysosomes, endosomes, multivesicular endosomes (MVEs) and autophagosomes/amphisomes, as well as cytoskeletal elements; however, they lack a definable nucleus.

Blebbisomes can take up and internalize smaller EVs from their extracellular environment. The presence of MVEs and functional cytoskeletons inside blebbisomes suggests they themselves actively secrete smaller EVs. Indeed, purified blebbisomes can release syntenin-1- and TSG101-positive EVs as well as annexin A1- and A2-positive EVs, indicative of 30–120 nm exosomes[2,13,14] and 150–1,000 nm microvesicles[1,2,15], respectively. In a zebrafish embryo model system, cancer cells can be observed to release blebbisomes in vivo, and blebbisomes can be isolated from normal mouse bone marrow. PD-L1 on tumour exosomes has been reported to contribute to immunosuppression[7,9]. We find that cancer cells release blebbisomes containing an abundance of immune evasion and inhibitory immune checkpoint proteins, including PD-L1, PD-L2, B7-H3, VISTA, PVR, Nectin-2, HLA-E, CD73 and CD47. These findings establish an additional layer of complexity in EV-mediated intercellular signalling and interaction in the extracellular microenvironment, with blebbisomes representing potential mobile communication centres.

## Results

### Blebbisomes form with intact mitochondria

While studying the release of vesicles and nanoparticles from cancer cells, we observed a population of exceptionally lEVs that exhibited pronounced membrane blebbing post-release (Fig. 1a–c and Extended Data Fig. 1). Membrane blebbing in cells has been previously studied during cell division, cell migration and apoptosis, and recently, it has been demonstrated that in melanoma cells, blebbing can confer resistance to anoikis[16]. A cellular membrane bleb is a portion of the plasma membrane that has been pushed out by intracellular pressure, similar to blowing up a balloon[17,18]. After one of these membrane blebs protrudes, actomyosin contractility retracts the bleb back towards the cell. Since this population of very lEVs exhibited constant membrane blebbing (Supplementary Videos 1–3), we named them 'blebbisomes'. Their size and continuous blebbing makes blebbisomes easily identifiable with a variety of light and electron microscopic techniques using both live and fixed samples (Fig. 1a,c,d and Extended Data Figs. 1 and 2a,b). We first used a transmitted light microscopic technique—differential interference contrast (DIC)—to investigate how blebbisomes arise. Timelapse DIC microscopy revealed that blebbisomes do not originate from a membrane bleb on the cell, as is typically assumed as the mechanism of release for other lEVs. Instead, the mechanism behind blebbisome formation involves a single retraction event, during which a portion of the cell remains attached to the substrate (Fig. 1e and Supplementary Videos 4 and 5). A membrane nanotube connects the cell body with the portion of the cell destined to become a blebbisome. The release of the blebbisome occurs when the nanotube is severed. The forces that drive cellular retractions are generated by the molecular motor, non-muscle myosin IIB[19]. To test the hypothesis that myosin IIB is required not only for the retraction event but also for the formation of blebbisomes, we knocked down expression of myosin IIB with siRNA and found that this depletion significantly reduced blebbisome release (Extended Data Fig. 3a,b). In contrast, knockdown of the two endosomal sorting complex required for transport (ESCRT) III proteins CHMP2A and CHMP4B, involved in membrane budding and scission[20], did not cause a reduction in blebbisome release (Extended Data Fig. 3a,c).

Timelapse imaging revealed that blebbisomes continuously bleb for at least 72–96 h. As membrane blebbing requires a constant supply of ATP[18], we predicted that blebbisomes contain functional mitochondria. Epifluorescence timelapse microscopy revealed that mitochondria were present at sites where blebbisomes form (Fig. 1f and Supplementary Video 6). The presence of mitochondria was confirmed using super-resolution instant structured illumination microscopy (iSIM) (Fig. 1f and Extended Data Fig. 2b) and expansion iSIM microscopy (Extended Data Fig. 2c). We found that every blebbisome examined contained mitochondria. We next determined if mitochondria within blebbisomes were functional. Functional mitochondria have a polarized membrane potential that is capable of generating ATP. Lipophilic cationic dyes, such as tetramethylrhodamine ethyl ester (TMRE), accumulate within polarized mitochondria due to their negative charge. TMRE localized to mitochondria in cells and was lost when the proton gradient was uncoupled by treatment with carbonyl cyanide-$p$-trifluoromethoxyphenylhydrazone (FCCP), as previously reported[21] (Fig. 1g,h). We found that TMRE had a similar fluorescent intensity within blebbisomes compared with that in cells (Fig. 1g). TMRE fluorescence was also reduced to a similar degree upon FCCP treatment as in cells (Fig. 1g,h). In addition to multiple human cancer cell lines (breast, colorectal and glioblastoma) and mouse melanoma cells, we also observed blebbisome formation from normal human colon fibroblasts and cardiomyocytes and mouse embryonic fibroblasts, indicating that blebbisome release also occurs from non-cancer cells. We next wondered if mitochondria have functions in blebbisomes other than producing ATP, which is obviously driving the membrane blebbing and motility. Mitochondria also play a central role in regulating cell death pathways such as apoptosis. Apoptosis can be triggered by multiple small molecules that target mitochondria through different mechanisms (for example, FCCP, staurosporine and raptinal). Staurosporine is a broad kinase inhibitor[22] and raptinal is a capase 3 activator[23]. Staurosporine and raptinal caused more blebbisome death than in dimethyl sulfoxide (DMSO) controls while FCCP was trending towards a similar result (Fig. 1i and Supplementary Video 7). These data support the concept that blebbisomes, similar to cells, can go through apoptosis. In summary, human and murine cells release blebbisomes, exceptionally lEVs ranging in size from 5 to 20 μm, that exhibit pronounced and prolonged blebbing and contain intact healthy mitochondria.

### Blebbisomes are distinct from other EVs at the protein level

Cells secrete a great diversity of EV types varying in size from 30 to 10 μm in size[2,4,6]. Due to the inherent difficulty of isolating specific subpopulations of EVs that may overlap in both size and composition, this heterogeneity is often compressed into two main categories; lEVs that are more than 200 nm in diameter (including microvesicles, apoptotic bodies and large oncosomes) and small EVs (sEVs) that are less than 200 nm in diameter (including exosomes, small ectosomes and arrestin domain-containing protein 1-mediated microvesicles (ARMMs))[1,2,4,24]. To determine how blebbisomes might differ from other types of secreted EV at the protein level, we first devised an isolation strategy to generate samples of purified blebbisomes (Methods, Fig. 2a, Extended Data Fig. 4a and Supplementary Video 8). It is well established that crude samples of lEVs and sEVs contain a multitude of non-vesicular particles and materials that can contaminate or complicate EV analysis[1,2,24–26]. To ensure the purity of isolated EVs we therefore employed high-resolution density-gradient fractionation[1,24] of lEV and sEV samples (Methods and Fig. 2a). Proteomic analysis (Supplementary Table 1) revealed that purified blebbisomes can be distinguished from cells, lEVs and sEVs at the protein level and are more similar to lEVs than sEVs (Fig. 2b). However, blebbisomes are distinctly different from lEVs in that they contain a large abundance of mitochondrial proteins (VDAC2 and VDAC1), cytoskeleton proteins (myosin IIa, alpha tubulin, actinin-4 and beta actin), ribosomal proteins (RPS8 and EEF2), ER proteins (calreticulin) and Golgi proteins (TGN protein 2), whereas lEVs (and sEVs) contain much less of these proteins (Fig. 2c,d and Extended

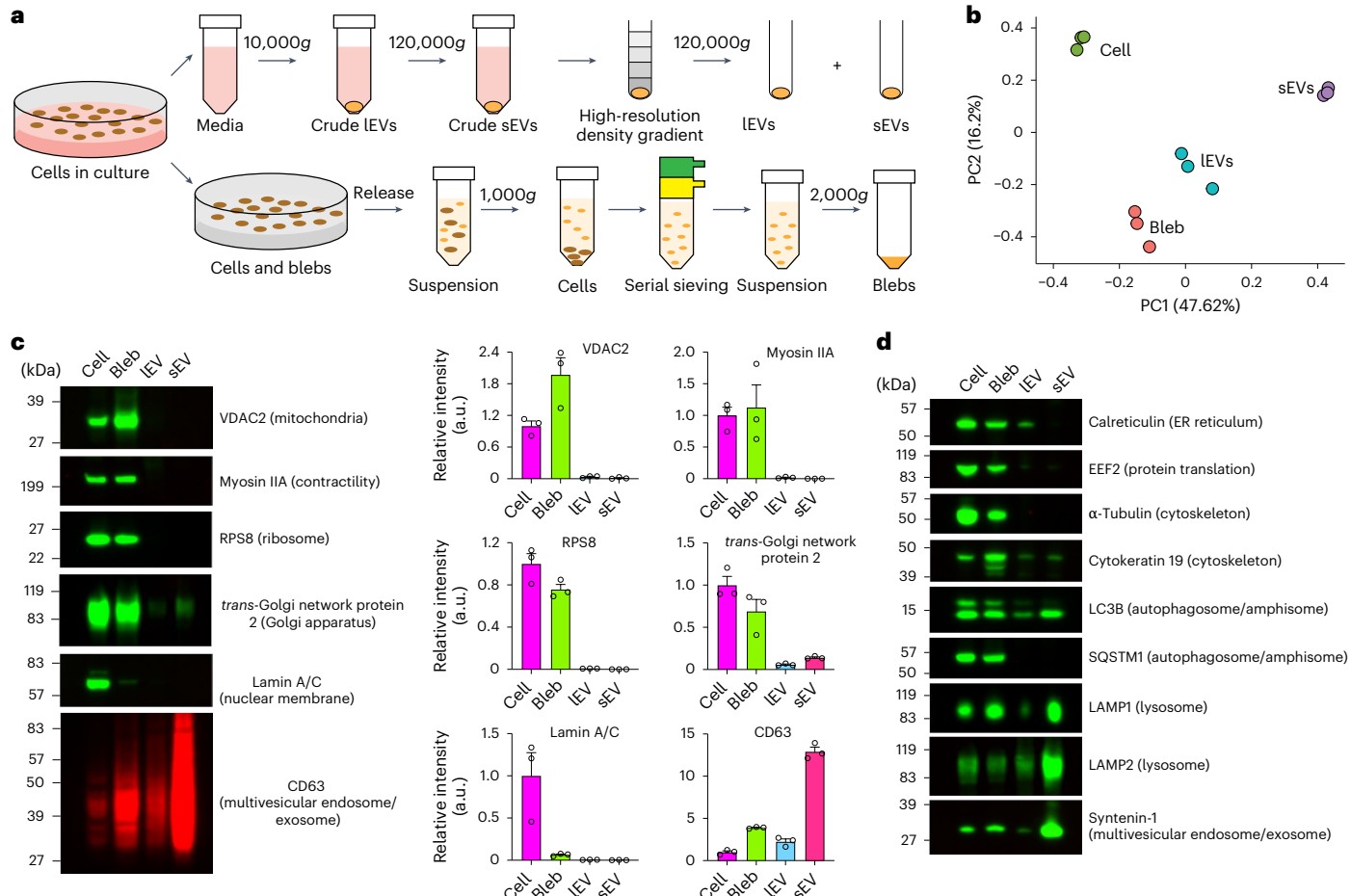

**Fig. 2 | Blebbisomes are distinct from other EVs at the protein level.**
**a**, A simplified experimental setup for purification of blebbisomes (blebs), lEVs and sEVs. **b**, A principal component analysis of normalized MDA-MB-231 proteomic mass spectral counts for cells, blebs, lEVs and sEVs (*n* = 3). **c**, Immunoblot analysis of MDA-MB-231 cells, bleb, lEV and sEV for select proteins (left) and quantification of relative fluorescence signal intensity (right). Each data point represents one independent experiment (see Extended Data Fig. 4b for replicate immunoblots) (*n* = 3 and data are displayed as mean ± s.e.m.). The images are representative of three independent experiments. **d**, An immunoblot analysis of MDA-MB-231 cells, bleb, lEV and sEV for select proteins. The images are representative of three independent experiments. a.u., arbitrary units.

Data Fig. 4b–d). In this regard, blebbisomes more closely resemble cells than either of the other two EV categories. Unlike cells, though, blebbisomes contain much less nuclear protein (Fig. 2c), as expected from their observed lack of a defined nuclear structure. In addition to mitochondrial, ribosomal, ER and Golgi proteins, we also found blebbisomes to contain proteins associated with other cellular organelles, including MVE, intraluminal vesicle (ILV), (CD63 and syntenin-1), lysosomal (LAMP1 and LAMP2) and autophagosome/amphisome (LC3B and SQSTM1/p62) proteins (Fig. 2c,d). Interestingly, proteomic analysis revealed that blebbisomes also express CD47, CD73 and Nectin-2/CD112 (Extended Data Fig. 4c and Supplementary Table 1), proteins that are involved in immune evasion and suppression. In conclusion, blebbisomes contain a great abundance of mitochondrial, cytoskeletal, ribosomal, ER and Golgi proteins in contrast to lEVs and sEVs but fewer nuclear membrane proteins than cells.

**Blebbisomes are distinct from large oncosomes**
Our proteomic analysis revealed that blebbisomes were most closely related to other lEVs including large oncosomes that are released from cancer cells[27–29]. Unlike blebbisomes, large oncosomes and microvesicles are believed to be produced from plasma membrane blebbing[1,29]. A membrane bleb typically retracts back to the cell, but if a bleb is released it becomes a microvesicles or large oncosomes, usually defined as one or the other based on size. Large oncosomes

are typically larger (1–10 µm)[1,29] than microvesicles (<1,000 nm)[1] and may be up to 5–10 mm, which can put them in a relatively similar size range as blebbisomes. Therefore, we wondered if large oncosomes also exhibited membrane blebbing. To test this, we added a membrane dye to purified large oncosomes. Large oncosomes displayed a round morphology and did not exhibit membrane blebbing (Fig. 3a). Proteomic analysis has suggested that large oncosomes also contain mitochondrial proteins, such as TU translation elongation factor (TUFM)[28]. Our proteomic data also detected TUFM in lEVs/large oncosomes, albeit with lower enrichment compared with blebbisomes (Fig. 3b). To further compare the the amount of TUFM, we localized it with immunofluorescence in both large oncosomes and blebbisomes (Fig. 3c). Similar to the proteomic analysis, quantification of the fluorescence levels showed that TUFM was enriched in blebbisomes (Fig. 3d). Given the presence of TUFM in large oncosomes, we next wanted to know if these mitochondria were functional. Therefore, we labelled purified large oncosomes and blebbisomes with TMRE. We found that 54% of the large oncosomes did not have signal above background. Of the large oncosomes that did, the TMRE signal was significantly lower compared with blebbisomes (Fig. 3e,f). Note the faint ring, which probably shows the plasma membrane (Fig. 3e, arrowhead). These data are consistent with previous reports suggesting that lEVs generally contain damaged mitochondria[2,10,30]. In addition, we also quantitatively compared other proteins that have been reported to be enriched in large oncosomes

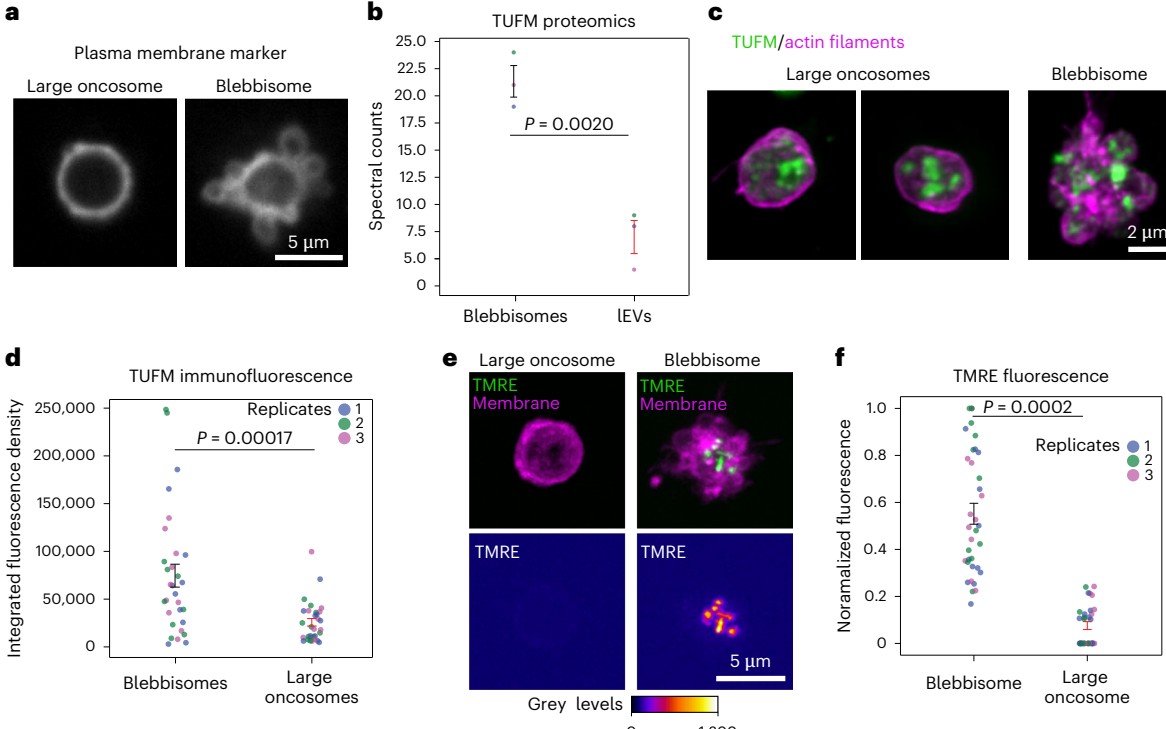

**Fig. 3 | Blebbisomes are distinct from large oncosomes. a**, A large oncosome and a blebisome from purified preparations stained with CellMask Deep Red to label the plasma membrane. **b**, Proteomics comparison of TUFM between blebbisomes and lEVs was shown to be significant by a Student's *t*-test. The mean spectral counts for blebbisomes is 21.33 ± 1.45 s.e.m. and for lEVs 7.00 ± 1.53 s.e.m. (*n* = 3 separate isolation preps). **c**, Maximum intensity projections of purified large oncosomes and a blebbisome labelled for actin filaments (magenta) and TUFM (green). **d**, An immunofluorescence comparison based on the integrated density of maximum intensity projections of the extracellular vesicle (*n* = 35 blebbisomes and 35 large oncosomes; representing three independent experiments each). Mean integrated fluorescent density, blebbisomes: 72,825.715 ± 11,992.082 s.e.m., oncosomes mean: 22,202.855 ± 3,901.163 s.e.m.

(*n* = 30 blebbisomes and 30 large oncosomes; representing three independent experiments each). **e**, Maximum intensity projections of a large oncosome and a blebbisome stained with CellMask Deep Red to label the plasma membrane (magenta) and TMRE (green) to stain for active mitochondria. **f**, The TMRE fluorescence was normalized on the basis of maximum intensity per *N* and then compared between blebbisomes and large oncosomes. Normalized fluorescence, blebbisomes mean: 0.55 ± 0.044 s.e.m., oncosomes mean: 0.066 ± 0.014 s.e.m. (*n* = 35 blebbisomes and 35 large oncosomes; representing three independent experiments each). The average diameter of large oncosomes was 4.72 ± 0.28 μm. The *P* values displayed in the graphs were derived from a two-tailed Student's *t*-test.

(Extended Data Fig. 5). As both large oncosomes and blebbisomes are formed from the plasma membrane and portions of the cytoplasm, it is not surprising that these proteins were found both types of EV, albeit with different enrichment. Taken together, these data indicate that large oncosomes and blebbisomes are distinct types of very lEV.

**Blebbisomes contain multiple organelles**

As we discovered blebbisomes to contain proteins associated with various cellular organelles, we next wanted to determine if any other intact organelles were present besides mitochondria. First, we examined blebbisomes with immunofluorescence and confirmed the staining for markers of the Golgi (GM130), cytoskeleton (myosin IIA and actin) and ribosomes (RPS8 and RPS10) (Fig. 4a, Extended Data Fig. 2b,c and Supplementary Table 2). Next, we examined purified blebbisomes with transmission electron microcopy (TEM). As expected from our previous observations (Fig. 1), an abundance of intact mitochondria was found in blebbisomes (Fig. 3b,c). Moreover, ER structures were readily observable by TEM. Consistent with the presence of endosomal and lysosomal proteins (Fig. 2), the organelles related to the endolysomal system were present in blebbisomes, including lysosomes, endosomes and MVEs, and what appeared to be an ongoing process of endocytosis could be observed; also consistent with our finding that blebbisomes contain LC3B and SQSTM1/p62, autophagosomal organelles were readily detectable. Blebbisomes also appeared to have an abundance of actin protrusions. In addition to the organelles mentioned above,

blebbisomes also consistently contain peroxisomes (PEX14) (Supplementary Table 2). As we had observed that blebbisomes contain ribosomes, this suggested that they may also contain RNA and indeed, fluorescence in situ hybridization (FISH) poly(A) probes confirmed the presence of RNA in blebbisomes (Extended Data Fig. 6). As already suggested by their lack of nuclear staining (Fig. 1) and protein composition (Fig. 2), TEM confirmed that a defining feature of blebbisomes is their lack of a nucleus structure. To summarize, blebbisomes are characterized by the consistent presence of many types of cellular organelle, including mitochondria, ER and Golgi, endosomes, lysosomes, peroxisomes and autophagosome/amphisomes, as well as actin protrusions.

**Blebbisomes take up and secrete exosomes and microvesicles**

As blebbisomes are extreme lEVs that contain endocytic pathway organelles (Fig. 4), we wondered if blebbisomes could take up other EVs smaller than themselves. To test this hypothesis, we first labelled purified EVs with Alexa Fluor-647 (ref. 26) and then incubated the cells and the blebbisomes they made with labelled EVs (Fig. 5a). The *XZ* and *YZ* slices confirmed that labelled EVs were present inside blebbisomes and not attached to the outside (Fig. 5a). We quantified both the presence and number of puncta in blebbisomes actively released by cells (Fig. 5b,c). We next wanted to confirm that blebbisomes could take up EVs independent of cell contribution. Therefore, we incubated purified MDA-MB-231 blebbisome preparations with labelled EVs and also found that they contained similar puncta (Fig. 5a–c). Taken

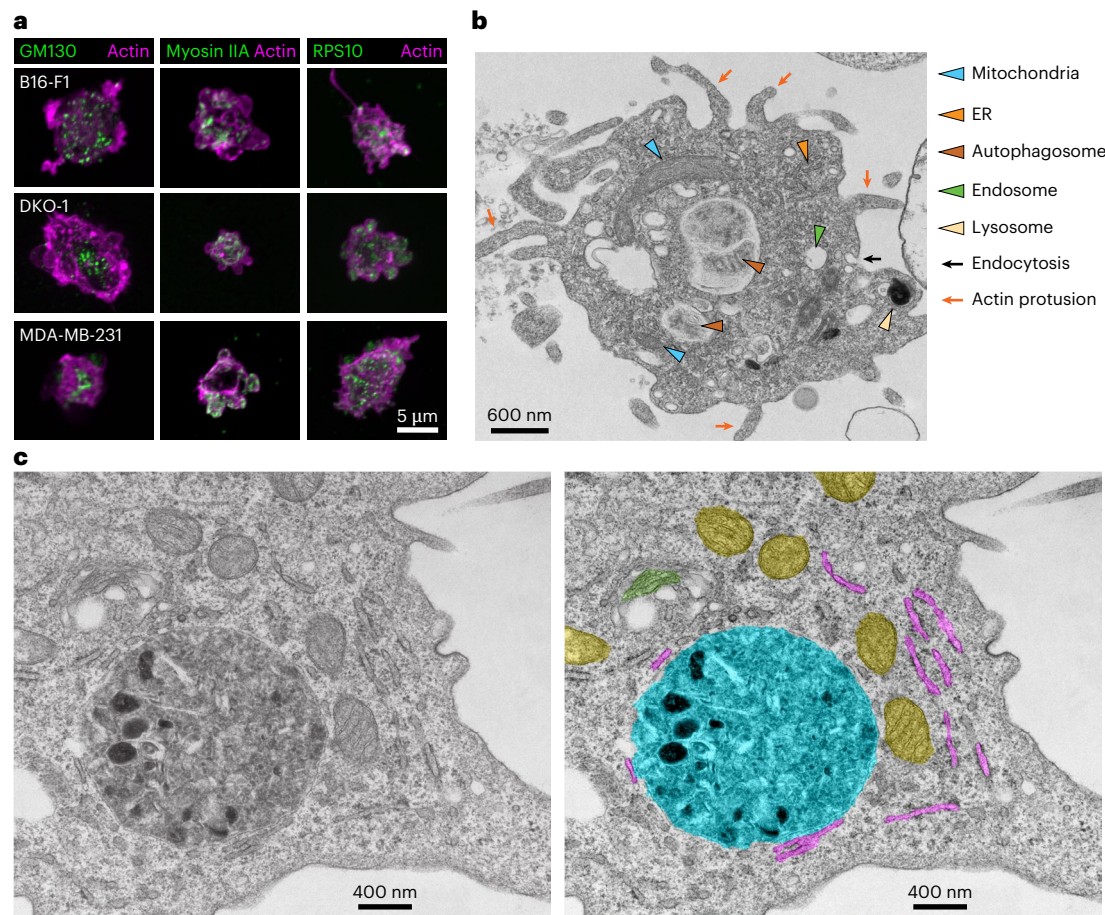

**Fig. 4 | Blebbisomes contain multiple organelles. a**, Immunofluorescence imaging of GM130, myosin IIA and RPS10 in B16-F1, DKO-1 and MDA-MB- 231 blebbisomes stained for actin by iSIM. The images are representative of three independent experiments. **b**, TEM imaging of purified MDA-MB-231 blebbisome. The coloured arrowheads indicate organelles or ultrastructures as indicated, the black arrows show endocytosis and the red arrows show actin protusions. **c**, A TEM image of a purified MDA-MB-231 blebbisomes (left) and a colour-coded (false colour) image (right). Yellow, mitochondria; green, Golgi apparatus; purple, ER; and turquoise, autophagosome–lysosome. Example micrographs from *n* = 2 independent blebbisome purifications are shown.

together, these data indicate that blebbisomes can take up EVs from their extracellular environment.

With the appearance of MVEs containing ILVs inside blebbisomes, it raised the question if blebbisomes might be able to secrete exosomes. Previously, structures resembling MVEs have been reported to be released in microvesicle-like protrusions from endothelial cells[31], and clusters of sEVs that express CD63 and ALIX can be released en bloc as MVE-like structures by colorectal cancer cells[32]. Supporting this possibility, it was easy to find more examples of blebbisomes with MVEs contained ILVs inside (Fig. 6a and Extended Data Fig. 7a). CD63 is a marker protein for exosomes and MVEs[1,2,4], and we found that blebbisomes contained CD63-positive compartments with sizes consistent for MVEs[1] (Fig. 6b). Furthermore, purified blebbisomes contained RAB27A and RAB27B (Extended Data Fig. 7b) that regulates transport of MVEs to the surface for release of exosomes[2,33,34] and RAB13 (Extended Data Fig. 7b) that is involved in release of sEVs from the plasma membrane[35]. In addition to CD63 (Fig. 2c and Extended Data Fig. 4b,c), blebbisomes also contained marker proteins for several EV subpopulations (Extended Data Fig. 7b), including syntenin-1, CD81 (exosomes), TSG101 (exosomes and ARMMs), annexin A1 and A2 (microvesicles) and CD147 (small ectosomes). To test the hypothesis that blebbisomes secrete EVs, we first cultured purified MDA-MB-231 blebbisomes for 48 h in serum-containing media that had been predepleted for contaminating bovine EVs by ultracentrifugation and filtration. The media was collected and processed for lEV and sEV isolation in a manner similar to EV purification from cell cultures. Due to the relatively small recoverable

amounts, for subsequent analysis by immunoblotting, we combined samples of lEVs and sEVs. Based on their respective marker proteins, we were able to detect both exosomes and microvesicles secreted into the media from blebbisomes (Fig. 6c,d and Extended Data Fig. 7c,d). Perhaps not surprisingly given the pronounced, continuous, blebbing and the abundance of functional mitochondria, purified MDA-MB-231 blebbisomes contained the glucose transporters GLUT1 and GLUT3 (Supplementary Table 1). GLUT1 has previously been demonstrated to be secreted in EVs from cells[36,37], and this protein was also present in the EVs released from blebbisomes (Fig. 6c,d and Extended Data Fig. 7d). Together, these data strongly suggest that blebbisomes can both take up and secrete EVs, thereby representing dynamic hubs for interactions in the extracellular environment.

**Blebbisomes are in vivo and have immune checkpoint proteins**
We next wanted to investigate if blebbisomes could be formed in more physiologically relevant environments. We found that human melanoma MV3 cells within a three-dimensional (3D) collagen matrix release EVs that appear to be blebbisomes (Fig. 7a and Supplementary Video 9). In addition, when mouse melanoma B16-F1 cells were injected into zebrafish embryos, they also released lEVs that displayed the characteristic blebbing behavior of blebbisomes (Fig. 7b and Supplementary Video 10). We wanted to test if blebbisomes are released in an in vivo context, by cells within an animal. Therefore, we extracted bone marrow from mice, immediately performed blebbisome purification. Several cell types in bone marrow are relatively small and in the same size range

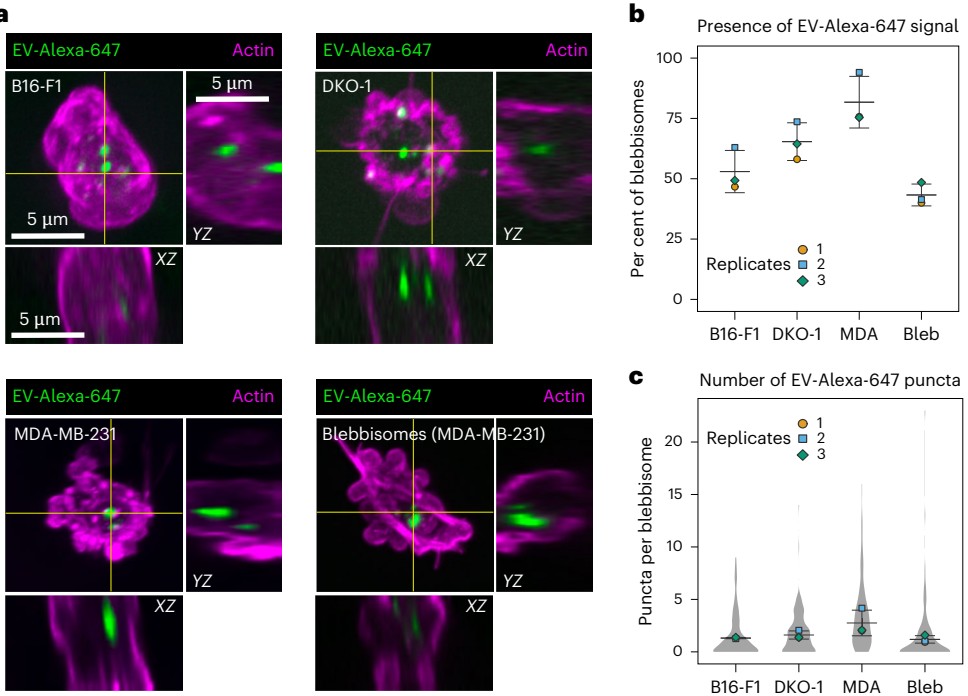

**Fig. 5 | Blebbisomes take up extracellular vesicles. a**, iSIM images of the uptake of EVs labelled with Alexa-647 in actin-stained blebbisomes released by B16-F1, DKO-1 and MDA-MB-231 cells or in prepurified samples of MDA-MB-231 blebbisomes. In the maximum projections, there are horizontal and vertical yellow lines that denote the *XZ* and *YZ* orthogonal view. The images are representative of three independent experiments. **b**, A quantification of the percentage of blebbisomes that contained EV-Alexa-647 signal. The per cent of blebbisomes containing EVs: 53 ± 9% s.e.m., 65 ± 8% s.e.m., 82 ± 11% s.e.m.,

43 ± 5% s.e.m. in B16-F1, DKO-1, MDA-MB-231 and prepurified MDA-MB-231 blebbisomes, respectively. The data are from three independent experiments. **c**, A quantification of the number of EV-Alexa-647 puncta per blebbisome. The number of puncta per blebbisome: 1.31 ± 0.06 s.e.m., 1.61 ± 0.39 s.e.m., 2.75 ± 1.22 s.e.m., 1.19 ± 0.36 s.e.m. in B16-F1, DKO-1, MDA-MB-231 and prepurified MDA-MB-231 blebbisomes, respectively. The data are from three independent experiments. Bleb, purified MDA-MB-231 blebbisomes.

as blebbisomes. As such, we needed to be able to distinguish been cells and large blebbisome-like EVs. Therefore, we labelled mitochondria as they are present in both blebbisomes and cells, as well as nuclei, which only cells have (Fig. 7c). Of note, red blood cells contain neither mitochondria or nuclei but can be distinguished by their concave appearance in DIC and lack of membrane blebbing (Fig. 7c). Blebbisome-like EVs were identified by their characteristic blebbing nature as well as a positive signal for mitochondria and negative signal for nuclei. The cells were identified by morphology and positive signal for nuclei and mitochondria (Fig. 7c and Supplementary Video 11). CD47 is a transmembrane cell surface molecule that functions through the monocyte and macrophage receptor signal-regulatory protein alpha (SIRPα) leading to inhibition of phagocytosis and, thus, serves as a 'don't eat me signal' to macrophages of the immune system[38,39], and the CD47-SIRPα immune checkpoint plays a broad role in cancer immune evasion[38,40]. The presence of CD47 on cancer-cell-derived sEVs allow them to escape immune rejection and extends their lifetime in circulation[41,42]. As blebbisomes derived from breast cancer, glioblastoma and melanoma cells all contained CD47 (Extended Data Fig. 4c), and since we had noticed blebbisome expression of the immunosuppressive effector molecules CD73 (refs. 6,26) and Nectin-2/CD112 (ref. 43) (Supplementary Table 1 and Extended Data Fig. 4c), we wondered if cancer blebbisomes contain inhibitory immune checkpoint ligands[6,44]. MDA-MB-231 cells express many of these ligands (Extended Data Fig. 8a), and they were present in purified MDA-MB-231 (breast cancer), Gli36 (glioblastoma) and B16-F1 (melanoma) blebbisomes (Fig. 7d and Extended Data Fig. 8b,c), including PD-L1, PD-L2, B7-H3, VISTA, PVR and HLA-E. MDA-MB-231 blebbisomes not only contained more PD-L2, VISTA and HLA-E than the corresponding lEVs and sEVs (Fig. 7d) but also released smaller EVs that expressed B7-H3 and PVR (Extended Data Fig. 8d). Taken together,

our data indicate that blebbisomes are released from cells in vivo and that cancer blebbisomes express a plethora of immune evasion and inhibitory checkpoint proteins.

## Discussion

Cellular blebbing, the formation of plasma membrane protrusions as the membrane decouples from the actomyosin cortex, is a common property of all cells. In healthy cells, it is associated with detachment, mitosis and migration[17,18]. In cancer, cellular blebbing has been linked to tumour cell motility[45]. Accumulation of signalling factors in the blebs has been linked to tumour cell survival and resistance to anoikis[16], and this process also occurs in human and mouse non-cancer cells[16]. Consistent with this, we observed that blebbisomes are continuously produced not only by human (colorectal, breast and glioblastoma) and mouse (melanoma) cancer cells but also by normal human and mouse embryonic fibroblasts. Blebbisome formation occurs in vivo as they can be observed being released from melanoma cells implanted in zebrafish embryos and are present in the bone marrow of normal mice. It, thus, appears that production of blebbisomes is a common cellular phenomenon and is not restricted to a specific cell type.

We were able to show that blebbisomes contain endocytic pathway components and compartments, appear to display visible signs of endocytosis and demonstrate that blebbisomes can actively take up other secreted EVs from the extracellular environment. In cells, MVEs of the endocytic pathway are transported to the cell surface, with the later steps regulated by RAB27A and RAB27B[2,33,34]. Fusion of MVEs with the plasma membrane results in release of ILVs contained within exosomes[3,5]. The cells can also release EVs by direct budding from the plasma membrane, including small-to-large EV sized microvesicles[1,2,5], small ectosomes[2,46] and sEV-sized ARMMs[1,2,47]. We report here that

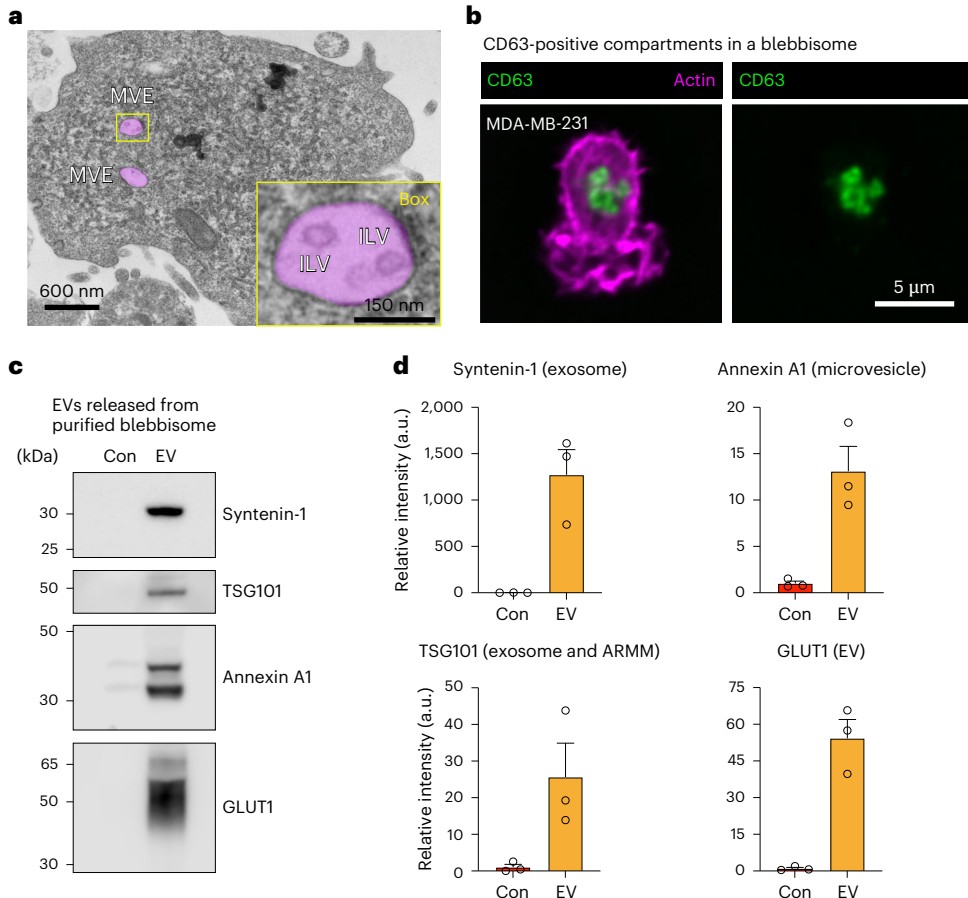

**Fig. 6 | Blebbisomes secrete exosomes and microvesicles. a**, A TEM micrograph of purified MDA-MB-231 blebbisome displaying MVE. The MVEs are pseudo-coloured magenta. Inset: enlarged box of a single MVE displaying multiple ILVs. **b**, Immunofluorescence imaging by iSIM of CD63 and actin in a MDA-MB-231 blebbisome. **c**, Immunoblot analysis of control media (Con) and EVs secreted from purified MDA-MB-231 blebbisomes for select EV marker proteins. The images are representative of three independent experiments. **d**, A quantification of relative signal intensity from **c**. Each data point represents one independent experiment (see Extended Data Fig. 7d for replicate immunoblots) (*n* = 3 and data are displayed as mean ± s.e.m.).

blebbisomes contain CD63-positive compartments of a size consistent with MVEs[1], and we directly observed purified blebbisomes that contain MVEs with ILVs inside. Of note, there is electron microscopy evidence that some smaller, microvesicle-like EVs can also contain MVEs[31]. Blebbisomes also contain TSG101, ALIX, RAB27A and RAB27B proteins associated with exosome secretion, as well as RAB13 that is associated with secretion of sEVs, probably directly from the plasma membrane[35]. Our experiments furthermore indicate that purified blebbisomes themselves can release EV that expresses syntenin-1, consistent with exosomes[2,13], TSG101, consistent with exosomes, and ARMMs[2,47], as well as annexin A1 and annexin A2, consistent with microvesicles[1,2]. Because of the relatively small amounts of secreted EVs obtainable from parental EVs (blebbisomes) compared with conventional secretion of EVs from parental cells, only small amounts of EV protein lysates (<1 µg) were available for immunoblotting. This might inform why we did not succeed in obtaining data for some common EV markers, including CD63 (exosomes)[1,2], ARRDC1 (ARMMs)[1,47] and CD147 (small ectosomes)[2,46]. In addition, we uncovered evidence that blebbisomes can undergo apoptosis. with concurrent generation of EVs, highlighting that not only cells but also blebbisomes may be sources of apoptotic EVs.

Blebbisomes at 10 µm average diameter but as big as 20 µm are the largest EVs reported so far, but other types of >1 µm vesicle are known, including migrasomes (0.5–3 µm), exophers (3.5–4 µm) and large oncosomes (1–10 µm)[2,4,6]. Mechanistically, blebbisome release occurs in a single retraction event in which the blebbisome is released from the plasma membrane and is left behind as the cell moves (Extended

Data Fig. 9). In contrast, multiple of the smaller migrasomes can be created per retraction event (Extended Data Fig. 9), which is consistent with previously published observations[30]. Of note, we never saw a blebbisome and migrasome being formed during the same cellular retraction event. Furthermore, we were able to obtain some insights into the formation of blebbisomes. Mechanistically, blebbisome release depends on non-muscle myosin IIB, a contractile protein of the myosin superfamily of motor proteins[19], consistent with the roles of myosin IIB in cellular motility and retraction. The ESCRT machinery function in remodelling of membranes such as sealing and repair, formation of MVEs, release of exosomes and ectosomes/microvesicles[20]. The ESCRT-III complex mediates budding and scission of membranes to form vesicles, both at MVEs and the plasma membrane[20]. We investigated a potential role of the ESCRT-III complex in blebbisome formation, but knockdown of the ESCRT-III proteins CHMP2A and CHMP4B did not reduce release of blebbisomes. The biogenesis of blebbisomes is clearly distinct from that of migrasomes, which form by ballooning of the membrane of a thin retraction fiber (Supplementary Video 12), and migrasomes are distinctly much smaller than blebbisomes (Supplementary Video 13)[12,30], while the biogenesis of exophers is currently unknown but has been speculated to be regulated by the autophagy pathway[10,11]. Unlike exophers and migrasomes, which are reported to perform the task of removing damaged mitochondria and toxic protein aggregates from cells under stress[10–12], blebbisomes are formed with intact, functional mitochondria that allows them to continue to bleb and interact with their surroundings for days.

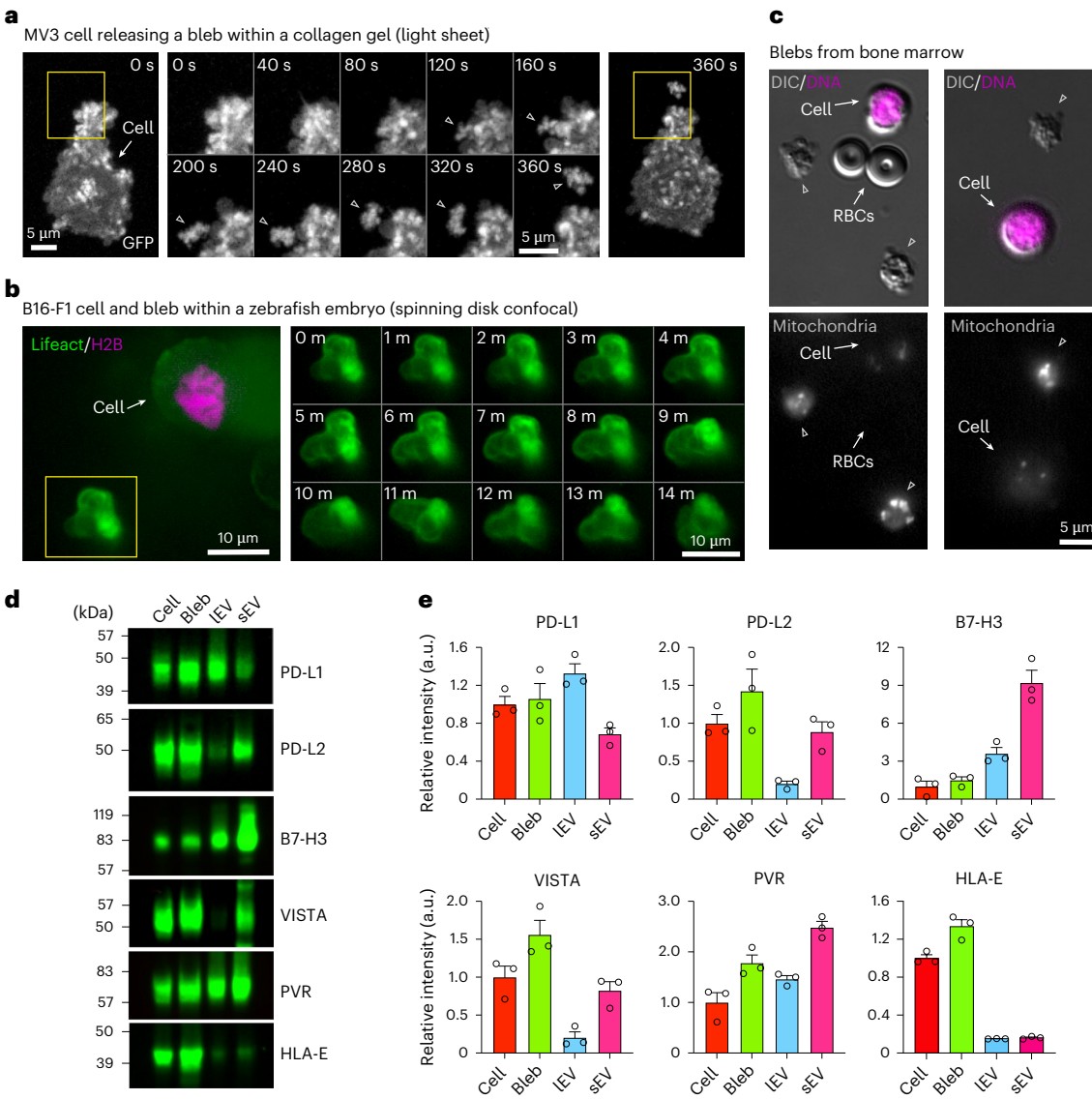

**Fig. 7 | Blebbisome-like EVs are present in vivo and contain immune checkpoint ligands. a**, A timelapse axially swept light sheet microscopy montage showing blebbisome (bleb) formation from MV3 cells embedded in collagen. Boxes: areas of bleb formation and release. Middle: the timelapse is for the enlarged area in the yellow box. **b**, A timelapse spinning-disk confocal montage of a bleb released from B16-F1 cell implanted in a zebrafish embryo and stained for actin (Lifeact) and Histone H2B. Box: a blebbisome displaying blebbing behavior.

**c**, A DIC microscopy montage showing bleb in bone marrow. RBC, red blood cell (*n* = 3 independent isolations). **d**, An immunoblot analysis of MDA-MB-231 cells, blebs, lEV and sEV for inhibitory immune checkpoint proteins. **e**, A quantification of relative signal intensity from **d**. Each data point represents one independent experiment (see Extended Data Fig. 5c for replicate immunoblots) (*n* = 3, and data are displayed as mean ± s.e.m).

Large oncosomes are atypically large microvesicles/ectosomes released by pinching off bulky protrusions or blebs from the plasma membrane of some cancer cells[1,28] as a means to influence and reprogramme the tumour microenvironment and distant sites; thereby, promoting disease progression and metastasis[27,48]. Large oncosomes can contain mitochondrial proteins[29] but, unlike blebbisomes, do not contain much ER protein (calnexin) or endosome/MVE protein (CD63 and TSG101)[48]. Here, we show that the presence of mitochondrial proteins within large oncosomes does not represent functional but damaged mitochondrial organelles as in exophers and migrasomes. Blebbisomes, exophers, migrasomes and large oncosomes can all be said to be very large ectosomes (plasma membrane-derived EVs) and share some common characteristics; however, only blebbisomes display pronounced and continuous membrane blebbing. Given their similarity in size, we have thought about the possibility that large oncosomes could represent inactive/dead blebbisomes. While it will

take more investiagion to fully answer this question, we feel that there are two reasons that this relationship is very unlikely. First, blebbisomes lose membrane integrity when they die similiy to cells and do not round up and float away into the media, as we would expect if they transformed into large oncosomes. Second, blebbisomes do not form from plasma membrane blebs as large oncosomes and microvesicles are thought to. Finally, it is important to note that there is much more to learn about blebbisomes. For example, it is not clear at this stage whether blebbisomes represent one homogeneous population of very lEVs or whether there are subcategories of blebbisomes.

T cells can recognize antigens on antigen-presenting cells, endowing a host with immunity to malignancies that produce neoantigens. However, the induction of T cell immunity responses can be heightened or dampened by ligands on tumour cells that transmit costimulatory or coinhibitory signals through receptors present on T cells, thereby, forming an immune checkpoint[44,49]. Engagement of immune

checkpoints can also restrain natural killer (NK) cells in a similar fashion[50]. Cancer-cell-derived exosomes (sEVs) that express the inhibitory immune checkpoint ligand PD-L1 have been reported to suppress CD8+ T cells[7,9], and suppression of exosomal PD-L1 is reported to induce antitumour immunity[9]. We systematically surveyed the expression of inhibitory immune checkpoint ligands across MDA-MB-231 (breast), Gli36 (glioblastoma) and B16-F1 (melanoma) cancer cells, blebbisomes and highly purified lEVs and sEVs. We found that cancer-cell-derived blebbisomes express not only PD-L1 but also a plethora of other inhibitory ligands, including PD-L2, B7-H3 (CD276), VISTA (B7-H5), HLA-E, PVR (CD155) and Nectin-2 (CD112). Circulating tumour cells that express HLA-E can escape NK-cell-mediated immune surveillance by engaging the heterodimer CD94-NKG2A[51]. VISTA mediates immune suppression by binding P-selectin glycoprotein ligand-1 (PSGL-1) on T cells[52]. MDA-MB-231 blebbisomes contained significantly higher levels of HLA-E, VISTA and PD-L2 than either lEVs or sEVs, and Gli36 blebbisomes also contained high levels of these proteins. Given the great abundance of different inhibitory immune checkpoint proteins on blebbisomes and other EVs, they should be investigated for their roles in immunosuppression and evasion in the tumour microenvironment.

Blebbisome production is a previously unrecognized common mechanism in mammalian cells for release of exceptionally lEVs characterized by pronounced long-lived membrane blebbing behaviour. Blebbisomes are unique from previously described EVs in at least four ways: (1) they are motile independent of cells; (2) they can secrete exosomes and microvesicles, as well as internalize EVs from their extracellular environment; (3) they contain functional mitochondria and are characterized by the presence of numerous cellular organelles that may enable them to perform a multitude of processes independently of cells; (4) they can make cell-like decisions, such as going through apoptosis. Taken together, our data suggest blebbisomes could function as motile cell-autonomous communication centres. Mechanistic dissection of this mode of secretion could, thus, inform fundamental mechanisms of intercellular communication in both physiological and pathological processes.

## Online content

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

## Methods

### Ethical approval

All animal studies were done in accordance with NIH, the US Department of Agriculture Animal Welfare Act, and the US Public Health Service Policy on Humane Care and Use of Laboratory Animals and were approved by Vanderbilt University Medical Center's Institutional Animal Care and Use Committee. The bone marrow from mice were collected according to M1800191-01. Zebrafish embryo experiments were conducted in accordance with M2100073-00-S2300172.

### Statistics and reproducibility

All experiments were performed with at least three independent biological replicates unless otherwise specified. Each replicate was derived from separate cell culture preparations or animals to ensure biological variability. Technical replicates were included for each experiment as detailed in the corresponding figure legends. The quantitative data are presented as mean ± standard error of the mean (s.e.m.) as indicated in the text or figure legends. Statistical analyses were conducted using GraphPad Prism v10.4.1, SuperPlots or Python (version 3.11.5). The statistical tests were chosen on the basis of the data distribution and experimental design. For comparisons between two groups, unpaired two-tailed Student's $t$-tests were used for normally distributed data. No statistical methods were used to predetermine sample sizes; the sample sizes were determined on the basis of previously published studies and pilot experiments. For microscopy experiments, random fields of view were selected for imaging but were not performed blind to the conditions of the experiments. No data were excluded from the analyses. All key findings were independently validated in at least three separate experiments, and reproducibility was confirmed across different experimental setups or cell lines when applicable.

### Key resources

For immunofluorescence experiments, we used the following primary antibodies: monoclonal mouse anti-APEX nuclease I (NeoBiotechnologies, 328-MSM1-P0), monoclonal mouse anti-Cytokeratin 18 (Proteintech, 66187-1-Ig), polyclonal rabbit anti-TUFM (Proteintech, 26730-1-AP), polyclonal rabbit anti-NMIIB (BioLegend, 909902), polyclonal mouse anti-TOM20 (Proteintech, 11802-1-AP), polyclonal rabbit anti-RPS8 (Proteintech, 18228-1-AP), polyclonal rabbit anti-PEX14 (Proteintech, 10594-1-AP), monoclonal mouse anti-GM130 (BD Biosciences, 610822), polyclonal rabbit anti-NMIIA (BioLegend, PRB-440P) and monoclonal rabbit anti-RPS10 (Abcam, ab151550). Additional antibodies included monoclonal mouse anti-mitochondria (Abcam, ab92824), monoclonal rabbit anti-HSP60 (Cell Signaling Technology, 12165) and antibodies against various intracellular proteins, such as annexin A1, annexin A2, syntenin and VDAC2 (all from Abcam: ab214486, ab178677, ab133267 and ab155803, respectively).

For immunoblotting, we employed primary antibodies against NMIIA (Abcam, ab138498), RPS8 (Abcam, ab201454), Lamin A/C (Abcam, ab108595 and ab169532), EEF2 (Abcam, ab75748), calreticulin (Abcam, ab92516), alpha tubulin (Abcam, ab52866) and cytokeratin 19 (Abcam, ab52625), as well as autophagy markers LC3B and SQSTM1/p62 (Abcam, ab192890 and ab109012). The lysosomal proteins were detected with monoclonal antibodies against LAMP1 (Abcam, ab108597 and ab208943) and LAMP2 (Abcam, ab199946). GLUT1 and TSG101 were detected using Abcam antibodies (ab115730 and ab125011, respectively), and additional proteins of interest were probed using antibodies from Abcam, Cell Signaling Technology, Thermo Fisher Scientific and BD Transduction Laboratories.

For secondary detection, goat anti-rabbit and goat anti-mouse IgG antibodies conjugated with Alexa Fluor dyes (Life Technologies, A11001, A11034, A11004, A11036, A32728 and A32733) or HRP-linked antibodies (Cell Signaling Technology, no. 7074) were used. Donkey anti-mouse and anti-rabbit IgG antibodies conjugated to Alexa Fluor Plus 680 or 800 (Thermo Fisher Scientific, A32788, A32730 and A32808) were employed for fluorescence imaging and immunoblotting.

Bovine serum albumin (RPI, A30075-100) was used as a blocking reagent. For labelling, we employed Phalloidin conjugates with Alexa Fluor 488, 568 and 647 (Invitrogen, A12379, A12380 and A22287), CellMask Deep Red (Thermo Fisher, C10046), TMRE (Thermo Fisher, T669), Mitotracker Green FM (Thermo Fisher, M7514) and SPY555-DNA (Cytoskeleton, CY-SC201).

The experimental models included MDA-MB-231 (ATCC, CRM-HTB-26), B16-F1 (ATCC, CRL-6323), DKO-1 (gift from Dr Takehiko Sasazuki), CCD-18Co (ATCC, CRL-1459) and MEF cells (gift from Dr Jennifer Lippincott-Schwartz). Imaging analysis was performed using Fiji software (NIH).

### Cell lines and culture

Human DKO-1 (male) colon cancer, human Gli36 glioblastoma, human MDA-MB-231 (female) breast cancer cells lines, human CCD-18Co (female) colon fibroblast cells, murine B16-F1 (male) melanoma cells and mouse embryonic fibroblasts from Dr Jennifer Lippincott-Schwartz were cultured in Dulbecco's modified Eagle's medium (DMEM) supplemented with 10% (v/v) fetal bovine serum (FBS) and 100 µg ml$^{-1}$ penicillin–streptomycin (Gibco Invitrogen) at 37 °C in a 5% $CO_2$ humidified incubator. The cells were plated on glass coverslips (Cellvis chamber glass-bottom dish, 35 mm dish with 10 mm bottom well and #1.5 glass) coated with 10 µg ml$^{-1}$ of fibronectin in the case of MDA and DKO cells. The B16 cells were plated on glass coverslips coated with 25 µg ml$^{-1}$ laminin. For immunofluorescence assays, the cells were fixed with 4% paraformaldehyde for 20 min and then permeabilized with 1% triton for 5 min. The cells were washed with 5 ml of 1× phosphate-buffered saline (PBS) three times after permeabilization. The cells were then blocked with bovine serum albumin for 20 min. The primary antibodies were added at a 1:200 dilution for 1 h at 37 °C. The cells were then washed with 5 ml of bovine serum albumin three times. The secondary antibodies were added at a 1:100 dilution for 1 h at room temperature (RT), after which the cells were washed with 5 ml of 1× PBS three times. Viafect with 4′,6-diamidino-2-phenylindole was added at the end to stain for DNA.

### Blebbisome isolation

The cells were seeded and cultured in DMEM medium with FBS depleted for EVs as previously described[53] to 30–50% confluence at 37 °C in a 5% $CO_2$ humidified incubator. The medium was removed, and the cells were washed in PBS. The cells and blebbisomes were trypsinized and collected in 15 ml conical tubes. All subsequent manipulations were performed at RT to preserve blebbisome activity. The cells were removed by centrifugation at 1,000$g$ for 5 min. The supernatants, containing blebbisomes, were successively filtered through stacked 10 and 5 µm sterile pluriStrainer (pluriSelect) sieves into 50 ml conical tubes to remove remaining cells. The supernatant was redistributed to new 15 ml conical tubes and subjected to centrifugation at 2,000$g$ for 10 min to pellet blebbisomes. The blebbisome pellet was resuspended in PBS for washing and repelleted at 2,000$g$ for 10 min. For microscopy and functional assays, the isolated blebbisomes were then kept at 37 °C, while for protein analysis, the blebbisomes were suspended in lysis buffer on ice and protein extracted as described below.

### Large and small extracellular vesicle isolation

The cell-conditioned medium was collected from MDA-MB-231, Gli36 and B16-F1 cells cultured for 48 h in DMEM with FBS depleted for EVs at 37 °C in a 5% $CO_2$ humidified incubator. The cell viability was assessed using trypan blue exclusion and only medium from cultures with >90% viability was used for isolation of EVs. lEVs and sEVs were isolated as previously described[24], with a few modifications. Briefly, the collected media was first subjected to a centrifugation step of 400$g$ for 10 min at RT to pellet and remove cells. All following centrifugation steps were performed at 4 °C. Next, the supernatant was spun at 2,000$g$ for 20 min

to remove debris and apoptotic bodies. Then, to pellet and collect crude lEV samples, the supernatant was centrifuged at 10,000$g$ for 40 min. The resulting lEV pellet was resuspended in a large volume of PBS followed by ultracentrifugation at 10,000$g$ for 40 min to wash the sample. To remove any remaining lEVs, the media supernatant from the first 10,000$g$ step was passed through a 0.22 µm pore PES filter (Millipore). This supernatant was subjected to ultracentrifugation at 120,000$g$ for 4 h in a SW 32 Ti Rotor Swinging Bucket rotor ($k$ factor of 204, Beckman Coulter) to sediment crude sEV samples. The crude sEV pellet (P120) was resuspended in a large volume of PBS followed by ultracentrifugation at 120,000$g$ for 4 h to wash the sample. Large oncosomes were purified as previously described[29], and immunofluorescence experiments performed on large oncosomes that were in the size range of 1–10 µm.

## High-resolution, 12–36% iodixanol, density-gradient fractionation of large and small extracellular vesicles

To remove non-vesicular contaminating material and further purify EVs, samples of crude lEVs and sEVs were subjected to high-resolution density-gradient fractionation as described previously[1,24]. Briefly, iodixanol (OptiPrep) density media (Sigma-Aldrich) were prepared in ice-cold PBS immediately before use to generate discontinuous step (12–36%) gradients as previously described[1,24]. Briefly, crude pellets of lEVs (P10) or sEVs (P120) were resuspended in ice-cold PBS and mixed with ice-cold iodixanol/PBS for a final 36% iodixanol solution. The suspension was added to the bottom of a centrifugation tube and solutions of descending concentrations of iodixanol in PBS were carefully layered on top yielding the complete gradient. The bottom-loaded 12–36% gradients were subjected to ultracentrifugation at 120,000$g$ for 15 h at 4 °C using a SW41 TI Swinging Bucket rotor ($k$ factor of 124, Beckman Coulter). Twelve individual fractions of 1 ml were collected from the top of the gradient. Each individual 1-ml fraction was transferred to new ultracentrifugation tubes, diluted 12-fold in PBS and subjected to ultracentrifugation at 120,000$g$ for 4 h at 4 °C using a SW41 TI swinging bucket rotor. The resulting pellets were lysed in cell lysis buffer for protein extraction (see below) for 30 min on ice.

## Protein extraction from cells, blebbisomes and extracellular vesicles

To extract cellular proteins, cultured cells were collected, washed twice with ice-cold PBS and solubilized in cell lysis buffer (20 mM Tris–HCl (pH 7.5), 150 mM NaCl, 1 mM Na$_2$EDTA, 1 mM EGTA, 1% Triton X, 2.5 mM sodium pyrophosphate, 1 mM beta-glycerophosphate, 1 mM Na$_3$VO$_4$, 1 µg ml$^{-1}$ leupeptin, 60 mM octyl β-D-glucopyranoside), to which complete Mini Protease Inhibitor Cocktail and PhosSTOP phosphatase inhibitor cocktail (both from Roche) and 2.0 mM Pefabloc (Sigma-Aldrich) was added immediately before use. The lysed samples were incubated on ice for 30 min. The protein content of cell lysates was quantified by a Direct Detect Infrared Spectrometer (Millipore). After the final wash step in PBS by ultracentrifugation, the blebbisome, lEV and sEV samples were lysed, and the proteins were extracted and quantified as described above for cell samples.

## Immunoblot analysis

The samples were prepared in lithium dodecyl sulfate buffer, heated to 70 °C for 10 min or incubated at RT for 20 min, before being loaded on gels. The samples were separated on 4–12% SDS–polyacrylamide gel electrophoresis Bis–Tris gels (Life Technologies) under either reducing or non-reducing conditions, depending on the subsequent use of primary antibody, before being transferred to Immobilon-FL polyvinylidenefluoride transfer membranes (EMD-Millipore). The membranes were blocked for 1 h in 5% non-fat dry milk or intercept (TBS) blocking buffer (LI-COR Biosciences), depending on the primary antibody subsequently used. For chemiluminescence detection of proteins, HRP-conjugated anti-rabbit IgG, anti-mouse IgG (Cell Signaling

Technology) secondary antibodies and western lightning Plus-ECL substrate (PerkinElmer) was used. For fluorescence detection of proteins, IRDye 680RD anti-mouse IgG (H+L), highly cross adsorbed, IRDye 800CW anti-rabbit IgG (H+L), highly cross adsorbed and IRDye 800CW anti-rat IgG (H+L) and highly cross adsorbed (LI-COR) secondary antibodies was used. Detection and quantification were performed with an Odyssey Fc Imaging System and Image Studio 5.2.5 software (LI-COR).

## Proteomics and proteomic analysis

The protein samples were brought to a final concentration of 5% SDS, were reduced with tris(2-carboxyethyl)phosphine (10 mM), alkylated with iodoacetamide (20 mM) and prepared by S-Trap (ProtiFi) digestion. Aqueous phosphoric acid and S-trap binding buffer (90% MeOH, 100 mM triethylammonium bicarbonate were added to each sample, and the samples were transferred to S-Trap micro spin columns according to manufacturer's instructions. The proteins were digested with trypsin (1:10 ratio) at 47 °C for 1 h. The peptides were eluted from the S-trap columns, and the eluates were dried by vacuum centrifugation. The peptides were reconstituted in 0.2% formic acid. Cells, purified blebbisomes, density-gradient purified lEVs and sEVs were analyzed as previously described[26]. Briefly, the samples were analyzed by nanoflow liquid chromatography with tandem mass spectrometry using a Dionex Ultimate 3000 nanoLC and a Q Exactive Plus Orbitrap mass spectrometer. The peptides were gradient-eluted with a 120 min reverse-phase gradient and analysed using a data-dependent method. For protein identification, data were searched with Sequest (Thermo Fisher Scientific) against a Homo sapiens UniprotKB database, including modifications of +15.9949 on Met (oxidation) and +57.0214 on Cys (carbamidomethylation). The search results were assembled in Scaffold 5.1.0 (Proteome Software) using a minimum filtering criteria of 95% peptide probability and 99% protein probability. The proteins with an average count of ≥1 in each fraction were considered detectable. The spectral counts of proteins were normalized to the total spectral counts. A principal component analysis was performed to assess the similarity between samples.

## Light microscopy

DIC and epifluorescence microscopy was performed using a Nikon Ti equipped with a Nikon DS-Qi2 cMOS camera and a 40× objective (NA of 0.95, Plan Apo, air). iSIM was performed using a Nikon Ti2 equipped with a Visitech iSIM, ORCA-Fusion CMOS camera (model C14440-20UP), a Nikon 60× objective (NA of 1.49, Plan Apo, oil) and a 100× objective (NA of 1.49, Plan Apo, oil). VisiView (Visitron Systems) software was used for acquisition. $Z$ planes were acquired using a 0.2 mm axial step size. Microvolution software installed in FIJI (Fiji Is Just ImageJ) was used to deconvolve the iSIM-data over 20 iterations.

## Expansion microscopy

The cells were plated on Cellvis chamber glass-bottom dish (35 mm dish with 10 mm bottom well, #1.5 glass) and stained for proteins of interest. The cells were then imaged with iSIM to acquire pre-expansion images. The cells were then incubated with an anchoring solution consisting of 186.25 µl of PBS, 5 µl acrylamide, 8.75 µl 16% paraformaldehyde at 37 °C overnight. Anchoring solution was then aspirated off and then 100 µl of cold polymerization solution, which consists of 980 µl of stock X, 10 µl 10% ammonium persulfate and 10 µl of $N,N,N',N'$-tetramethylethylenediamine diluted 1:7.75 in deionized (DI) water was added directly to the plate. Stock X is a solution of 2.5 ml of 38% sodium acrylate, 0.5 ml of acrylamide, 0.75 ml of $N,N'$-methylenebisacrylamide, 4 ml of 5 M NaCl, 1 ml of 10× PBS and 1 ml of DI water. After addition of the polymerization solution, the cells were incubated in a humidifying chamber, an old pipette box with water filling the bottom and a plastic insert for the plate to rest on, at 37 °C in a 5% CO$_2$ humidified incubator for 1 h. After the gel forms, proteinase K is added at 1:100 dilution in digestion buffer to the cells and incubated for at least 6 h at 4 °C. The digestion buffer is made by combining 10 ml of Triton X-100, 0.2 ml of

EDTA, 5 ml of 1 M Tris, 20 ml of 5 M NaCl and 64.8 ml of DI water. The gel is then washed three times with DI water, waiting 10 mins between washes. The gel is then transferred to a MatTek 50 mm glass-bottom dish no. 0 and then imaged.

## SEM

The cells were plated on gridded glass Coverslips (iBidi, catalogue no. 10816). The cells were fixed with 2% glutaraldehyde for 2 h and postfixed sequentially in 1% tannic acid, 1% $OsO_4$ and 1% uranyl acetate. The samples were dehydrated in a graded ethanol series and dried with a Tousimis Samdri-PVT-3D critical point dryer and coated with 2-nm-thick platinum using a Leica ACE600 ebeam system and then imaged with a Zeiss Crossbeam 550.

## Correlation electron microscopy

For correlation electron microscopy, the cells were plated on gridded glass Coverslips (iBidi, catalogue no. 10816). The cells were fixed with 4/% paraformaldehyde for 20 min and then permeabilized with 0.5% triton for 5 min before three washes with PBS and incubation with Alexa 488-phalloidin (1:40 in PBS) for 2 h. The blebbisomes were identified and imaged using an iSIM microscope and each position on the grid noted. The cells were then fixed with 2% glutaraldehyde for 2 h and prepared for scanning electron microscopy (SEM) as above.

## TEM

The isolated blebbisomes were fixed with 2.5% glutaraldehyde for 60 min followed by sequential postfixation in 1% tannic acid, 1% $OsO_4$ and en blocked stained in 1% uranyl acetate. The samples were dehydrated in a graded ethanol series and infiltrated with Quetol 651 based Spurrs using propylene oxide as the transition solvent. The resin was polymerized at 60 C for 48 h and samples were sectioned at 70 nm nominal thickness on a Leica UC7 ultramicrotome and collected onto 300 mesh Ni grids. Grids were stained with lead citrate and 1% uranyl acetate. Transmission electron microscopy (TEM) imaging was performed on a Tecnai T12 operating at 100 keV using an AMT nanosprint 5 CMOS camera.

## Blebbisome production assay

The MDA-MB-231 cells were plated onto 35-mm glass-bottom dishes with 20 mm microwell #0 cover glass. They were allowed to grow for 48 h to give ample time for blebbisomes to form. The plates were then fixed with 4% paraformaldehyde and stained with phalloidin 568 and 4′,6-diamidino-2-phenylindole. A 2046.74 µm × 2046.74 µm stiched image with 15% overlap is taken of the plate. All of the cells and blebbisomes are counted in this image from which a ratio of the number of blebbisomes to cells is derived.

## TMRE assay

The cells were plated on a Cellvis 4-chamber glass-bottom dish (35-mm dish with 20 mm bottom well, #1.5 glass) for 24 h before experimentation. TMRE was then added at a concentration 1.5 nM and incubated for 30 mins before imaging. A cell with a blebbisome nearby was selected for imaging. The region of interest was then imaged every 100 ms for 1 s to get the baseline fluorescence. FCCP was then added at a concentration of 6 µM. A total of 5 s following addition, the same region of interest (ROI) was imaged every 100 ms for 1 s. The fluorescence measurements were calculated by generating a square ROI (22.891 µm²) over the blebbisome encompassing the mitochondria and then measuring the mean fluorescence. The same ROI was used to measure the mean fluorescence of the cell encompassing a similar area of mitochondria. The mean fluorescence was the normalized to the background fluorescence of the original ROI containing both the cell and blebbisome.

## EV uptake assay

The samples of purified EVs (see 'Large and small extracellular vesicle isolation') were labelled with Alexa Fluor-647 (Invitrogen, A20173) as previously described[26]. To monitor the uptake of EVs, B16-F1, DKO-1 and MDA-MB-231 cells (20,000 cells per well) were seeded on a 35-mm dish (P35G-0.170-14-C, MatTek Corporation) in DMEM culture medium overnight to produce blebbisomes. Prepurified MDA-MB-231 blebbisomes (see above) were also seeded. The samples were then treated with Alexa Fluor-647-labelled EVs (40 µg ml$^{-1}$) in serum-free DMEM media. The images were acquired using a ×60 objective on a VisiTech iSIM with a Nikon Ti base. Fluorescence (640 far red, 10% laser power, 100 ms exposure time) images were taken. Three z-slices, 1 µm apart, were taken of each fluorescent field, and the maximum z-projection was analysed.

## EV secretion assay

The blebbisomes were purified as described above, resuspended in DMEM with FBS depleted for EVs and plated in cell culture dishes. After 48 h, the blebbisome-conditioned media were collected. Crude blebbisome-derived lEVs and sEVs were isolated from the media, as described above. After isolation, lEVs and sEVs were combined. In parallel, the samples of the same type of media but without blebbisomes were processed for isolation of lEVs and sEVs to serve as a control for any contaminating bovine EVs in subsequent analyses. Blebbisome-derived EVs were lysed for protein extraction, and immunoblot analyses were performed as described above.

## Zebrafish embryo injection and imaging

The zebrafish line LH1066 was used according to institutional ethical guidelines. B16-F1 melanoma cells were transfected with lipofectamine 3000 to express LifeAct–GFP and H2B–mCherry. These cells were then injected intracranially into zebrafish 48 hpf. The zebrafish were imaged starting at 56 hpf with a CSU-W1 yokogawa spinning-disk microscope, 95B sCMOS camera and a Plan Fluor 40×/1.30 NA oil objective.

## RNA FISH

FISH was performed in 10 cm, 35 mm well Mattek dishes according to the manufacturer's protocol for adherent cells (Biosearch Technologies) with minor adjustments. Permeabilization was performed with 0.01% Triton X-100 for 20 mins at RT. For hybridization, a positive control probe to the 5′ poly(A) tail was used (catalogue no. T30-Calfluor 590-1). Following wash buffer B incubation, immunofluorescence was performed as previously described in 'Cell lines and culture', starting with blocking.

## 3D collagen assay

All 3D imaging of MV3 melanoma cells in collagen extracellular matrix environments was performed as previously described[54,55]. Specifically, to evaluate cell morphology and vesicle shedding events, MV3 cells were lentivirally transduced with GFP–Tractin (pLVX-GFP-TRactin-IRES-PURO) and isolated using either flow cytometry or antibiotic selection. Thereafter, stably fluorescent cells were treated with trypsin and placed in a pH-neutral rat-tail collagen I solution (4 mg ml$^{-1}$, Corning 354249). The mixture was subsequently transferred to a custom polytetrafluoroethylene holder and polymerized at a temperature of 37 °C. Once the polymerization process was complete, the sample was transferred into culture media and allowed to incubate for a duration of ~24 h before imaging with Axially Swept Light-Sheet Microscopy. All data shown are raw (for example, no denoising or deconvolution was performed).

## Blebbisome isolation from bone marrow

The 8-week-old female C57BL/6J mice were euthanized with isoflurane and hind legs were removed. The marrow was flushed from bilateral femurs and tibias using ice-cold DMEM containing 4.5 g l$^{-1}$ glucose and a 26G needle. The cell suspensions were passed over a 40-mm filter and collected in fresh DMEM. The suspension of bone marrow was then subjected to centrifugation at 1,000g for 5 min at RT to sediment bone

marrow cells. Next, the blebbisomes were isolated from the supernatant as outlined above. Isolated blebbisomes were stained with mitotracker green FM as well as SPY555-DNA and then imaged by DIC microscopy and widefield epifluorescence as outlined above 30 min after plating.

## Reporting summary

Further information on research design is available in the Nature Portfolio Reporting Summary linked to this article.

## Data availability

Mass spectrometry data have been deposited in ProteomeXchange with the primary accession code PXD059407. All other data supporting the findings of this article are available from the corresponding authors on reasonable request. Source data are provided with this paper.

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

## Acknowledgements

We thank the members of the Burnette and Coffey labs for insightful discussions. We also thank K. Hyde for assistance culturing cells, and the Vanderbilt Nikon Center for Excellence and the Vanderbilt Center for Imaging Shared Resources for experimental and imaging assistance. We thank the Vanderbilt MSRC Proteomics Core for experimental assistance. We thank Vanderbilt's Program in Developmental Biology, Microtubules and Motors Club and Molecular Biophysics Training Program for project feedback and discussion.

## Author contributions

D.T.B. discovered the blebbisomes, and D.K.J. developed the method for isolation of blebbisomes. D.T.B. and D.K.J. conceived the study. D.T.B., Z.C.S. and D.K.J. designed and performed the majority of the experiments with assistance from the other coauthors. Z.C.S., J.B.H., O.L.P., M.J.T. and E.W.K. performed the zebrafish experiments. Z.C.S., D.K.J., M.M.D. and A.C.D. performed the bone marrow experiments. Z.C.S., J.A. and N.T. performed the myosin IIB knockdown experiments. D.K.J., Z.C.S. and E.K. performed the electron microscopy experiments. K.M.D. performed the collagen gel experiment. D.K.J. and Q.Z. performed extracellular vesicle purifications and immunoblotting, with R.J.C. assisting in data analysis and interpretation. Z.C.S. and E.N.K. performed the poly(A) FISH. Z.C.S., D.K.J., N.M.K. and D.T.B analysed the data. D.T.B., D.K.J. and Z.C.S. wrote the paper, and all authors read and contributed to its preparation.

## Competing interests

The authors declare no competing interests.

## Additional information

**Extended data** is available for this paper at https://doi.org/10.1038/s41556-025-01621-0.

**Correspondence and requests for materials** should be addressed to Dennis K. Jeppesen or Dylan T. Burnette.

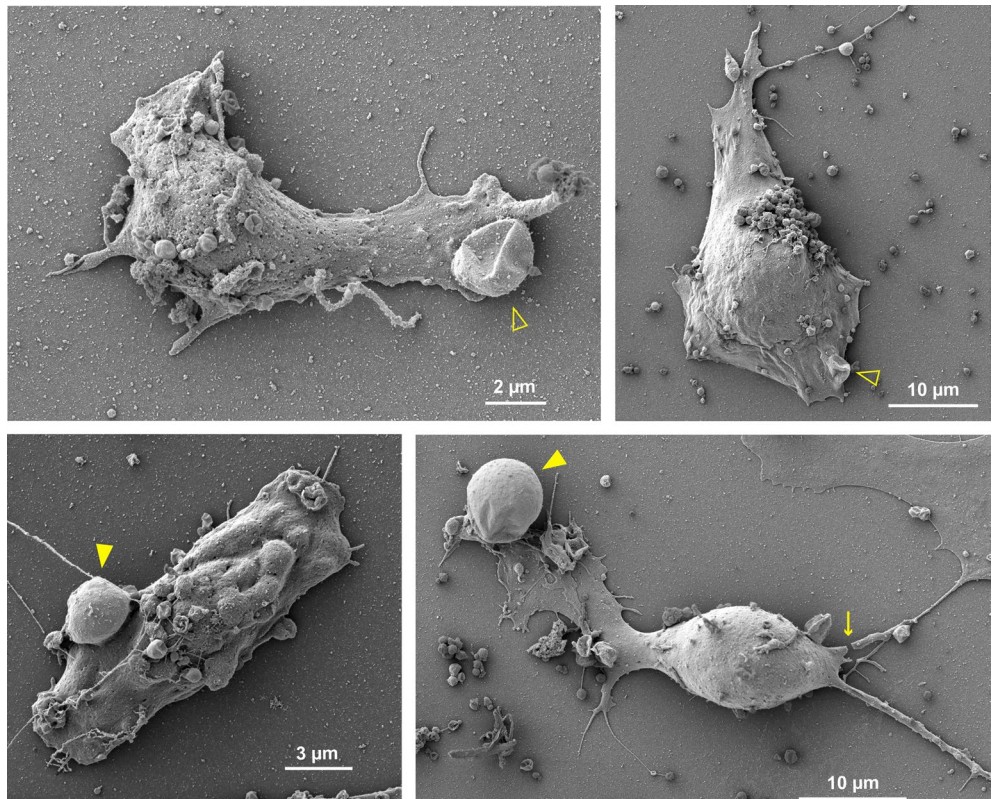

**Extended Data Fig. 1 | Scanning electron microscopy images of blebbisomes.** Scanning electron microscopy (SEM) micrographs of B16-F1 blebbisomes displaying characteristic blebs. Open arrowheads denote blebs that are likely retracting and closed arrowheads denote blebs that are likely growing. Lower right panel shows a potential blebbisome still attached to or just released from the parent cell. The arrow denotes the separation between the blebbisome and cell, which could have been an artifact of specimen preparation. SEM was performed 2 times independently with similar results.

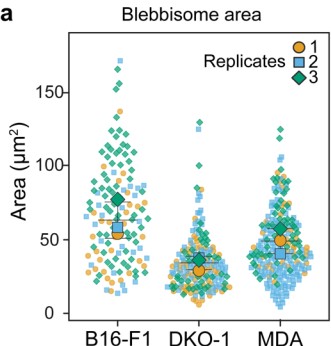

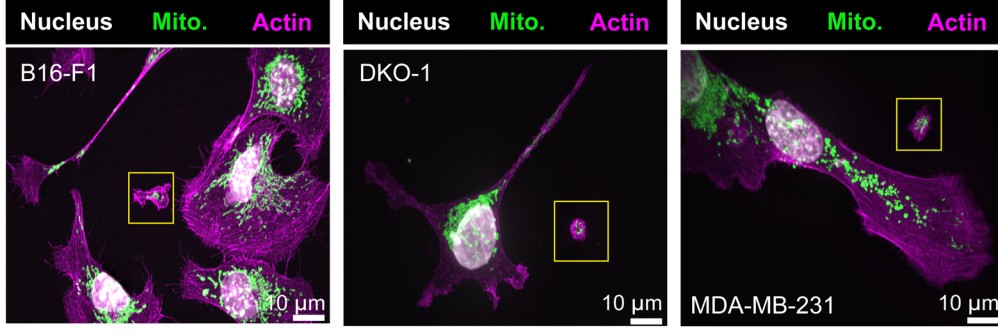

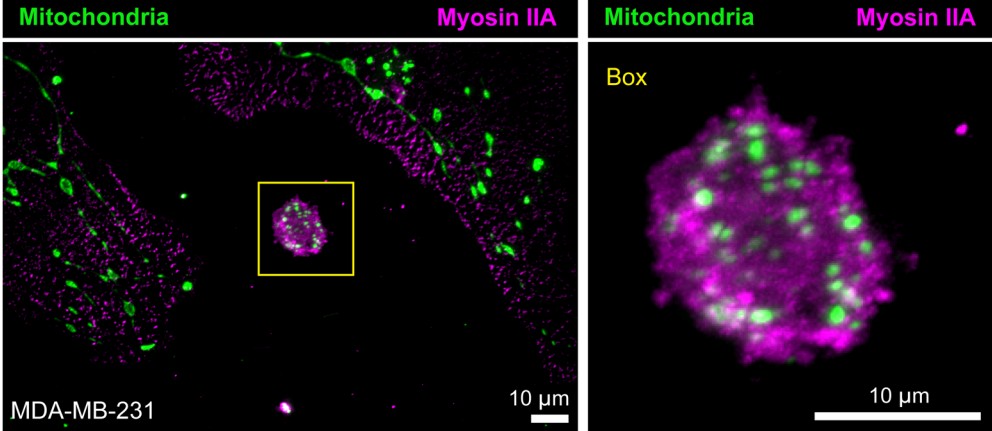

**Extended Data Fig. 2 | Blebbisomes form with functional mitochondria. a**, Average blebbisome area: 89.4 +/- 18.6 SEM, 36.0 +/- 6.1 SEM, 51.2 +/- 9.1 SEM µm2 for B16-F1, DKO-1 and MDA-MB-231 cells, respectively. n = 167, 183 and 236 blebbisomes across three independent experiments for B16-F1, DKO-1 and MDA-MB-231 cells, respectively. **b**, Imaging of nucleus, actin filaments and mitochondria in cells and blebbisomes by iSIM. Boxes indicate blebbisomes. Higher magnification version of images from Fig. 1f. **c**, Imaging of mitochondria and myosin IIa in blebbisomes by expansion microscopy using iSIM (Ex-iSIM). Box indicate a blebbisome. Enlarged inset is display on the right. Source numerical data are available in source data.

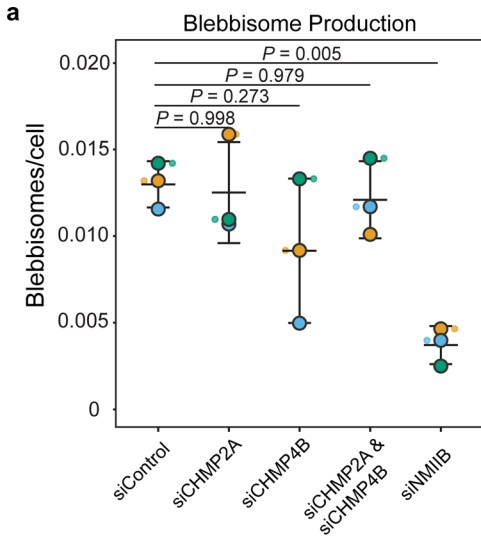

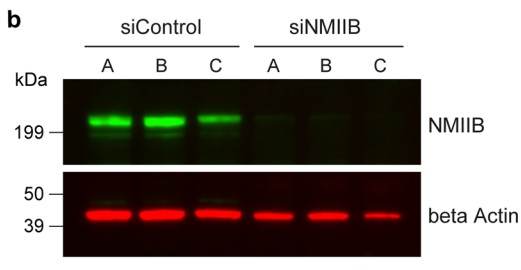

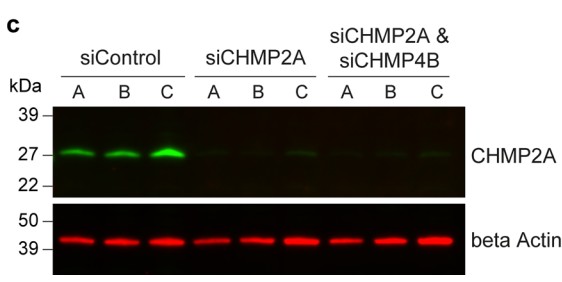

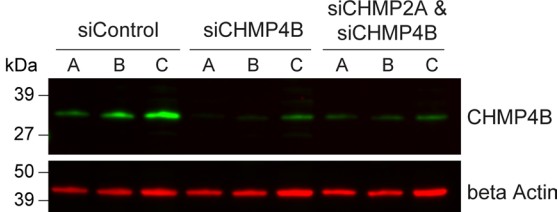

**Extended Data Fig. 3 | NMIIB is necessary for blebbisome formation. a**, Quantification of blebbisomes produced by MDA-MB-231 cells after siRNA knockdown of CHMP2A, CHMP4B and NMIIB. The mean ratio of blebbisomes/cell, siControl: 0.01 +/− 0.0, siCHMP2A: 0.001 +/- 0.00, siCHMP4B: 0.01 +/- 0.00, and siNMIIB 0 +/- 0.00. n = 3 independent experiments. **b**, Immunoblots of NMIIB and beta Actin in MDA-MB-231 cells following NMIIB siRNA knockdown. **c**, Immunoblots of CHMP2A, CHMP4B and beta Actin in MDA-MB-231 cells following CHMP2A and CHMP4B siRNA knockdown. Data are from three independent siRNA knockdown experiments. P values displayed in graphs were derived from a two-tailed student's t-test. Source numerical data are unprocessed western blots are available in source data.

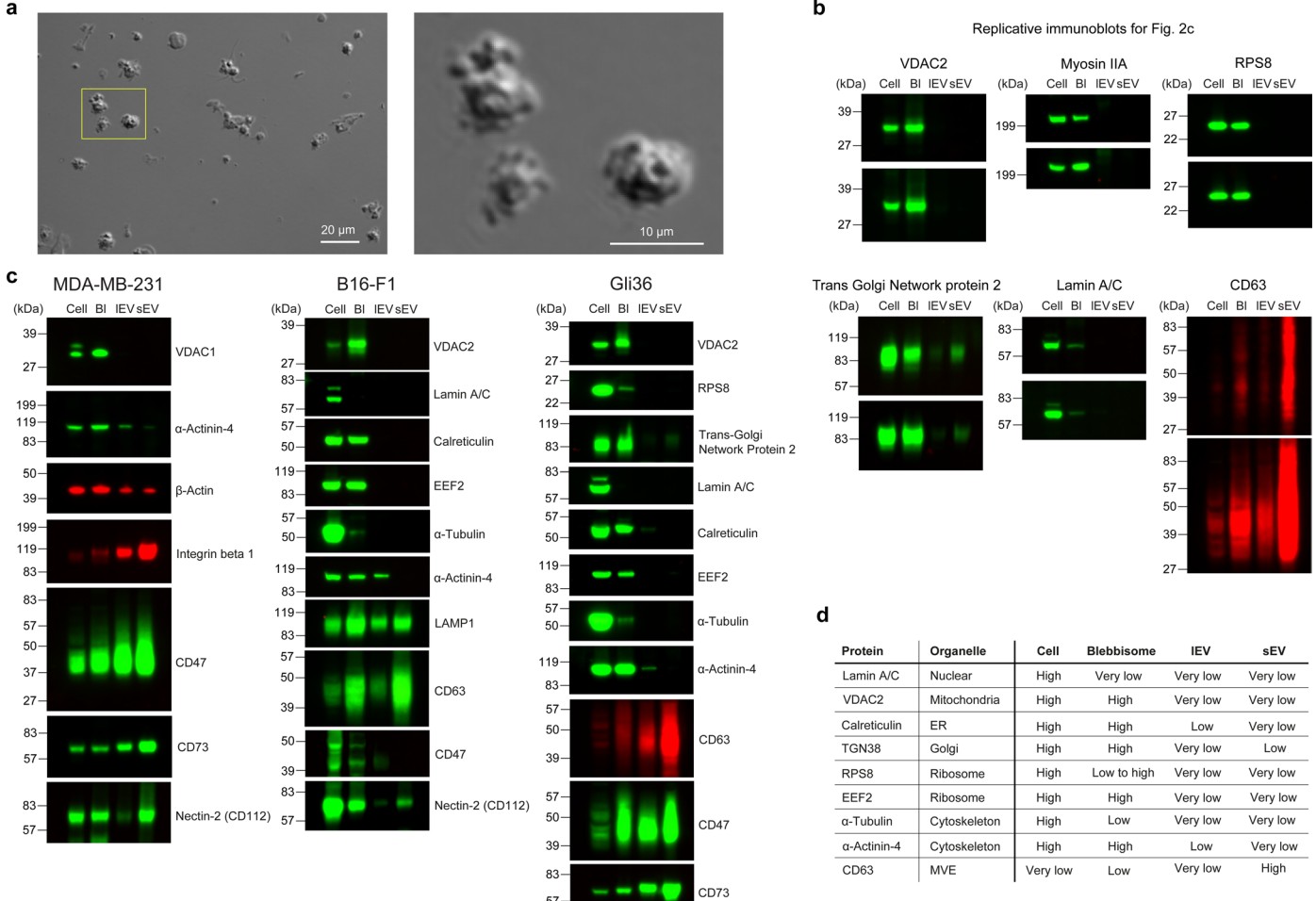

**Extended Data Fig. 4 | Purified blebbisomes and immunoblots of select proteins. a**, DIC image of MDA-MB-231 blebbisomes purified as outlined in Fig. 2a. Enlarged inset is display on the right. **b**, Replicative immunoblot experiments of MDA-MB-231 cells, purified blebbisomes (bleb), large Evs (lEV) and small Evs (sEV) for select proteins. Related to Fig. 2c. **c**, Immunoblot analysis of MDA-MB-231 (breast cancer), Gli36 (glioblastoma) and B16-F1 (melanoma) cells, blebbisomes (bleb), large Evs (lEV) and small Evs (sEV) for select proteins.

Images are representative of three independent experiments. **d**, Summary of common protein expression pattern for MDA-MB-231, B16-F1 and Gli36 cells, purified blebbisomes, lEVs and sEVs based validation by immunoblot analysis. Based on data from Fig. 2c, d and Extended Data Fig. 3b, c. TGN38 is Trans Golgi Network Protein 2. Source numerical data and unprocessed blots are available in source data.

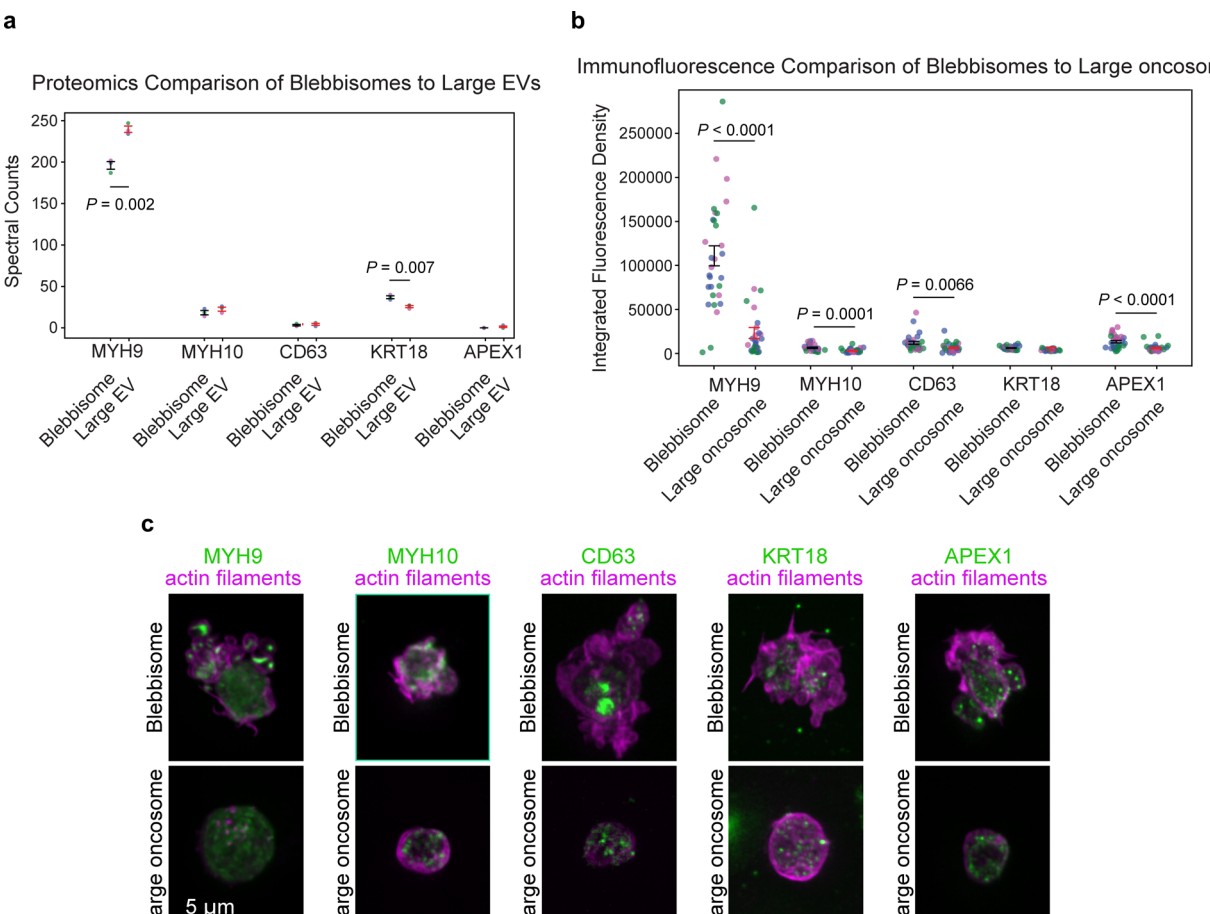

**Extended Data Fig. 5 | Further comparison between blebbisomes and large oncosomes. a**, Proteomics comparison of proteins known to be enriched in large oncosomes (MYH9, MYH10, KRT18, and APEX1) as well as CD63 to serve as a known EV marker. MYH9 (p = 0.002), blebbisome mean: 196.00 +/- 4.58 SEM, large EV mean: 239.67 +/- 3.84 SEM. MYH10, blebbisome mean:18.33 +/- 2.60 SEM, large EV mean: 22.67 +/- 2.40 SEM. CD63, blebbisome mean: 3.33 +/− 0.88 SEM, large EV mean: 4.33 +/- 1.20 SEM. KRT18 (p-value = 0.007), blebbisome mean: 37.00 +/- 1.73 SEM, large EV mean: 25.67 +/- 1.33 SEM. APEX1, blebbisome mean: 0.00, large EV mean:1.33 +/- 0.88 SEM. n = 3 independent isolation preps. **b**, Immunofluorescence comparison based on the integrated density of maximum intensity projections of the blebbisome and large oncosome. The proteins chosen are known to be enriched in large oncosomes (MYH9, MYH10, KRT18, and APEX1) as well as CD63 to serve as a known EV marker.

MYH9 (p-value < 0.0001), Blebbisome mean: 109857.81 +/-14895.17 SEM, Large oncosome mean: 22157.01 +/- 6882.76 SEM. MYH10 (p-value = 0.0001), Blebbisome mean: 11796.96 +/- 2742.26 SEM, Large oncosome mean: 6285.41 +/- 1027.22 SEM. CD63 (p-value = 0.0066), Blebbisome mean: 5904.93 +/-1592.75 SEM, Large oncosome mean: 2730.8 +/- 445.56 SEM. KRT18, Blebbisome mean: 4145.94 +/- 516.41 SEM, Large oncosome mean: 1943.83 +/- 105.23 SEM. APEX1 (p-value < 0.0001), Blebbisome mean: 11969.16 +/- 4898.36 SEM, Large oncosome mean: 4703.29 +/-1537.73 SEM. n = 3 independent isolation preps. **c**, Maximum intensity projections of blebbisomes and large oncosomes stained for actin (magenta) and a protein of interest (green). P-values displayed in graph were determined using a two tailed student's t-test. Source numerical data are available in source data.

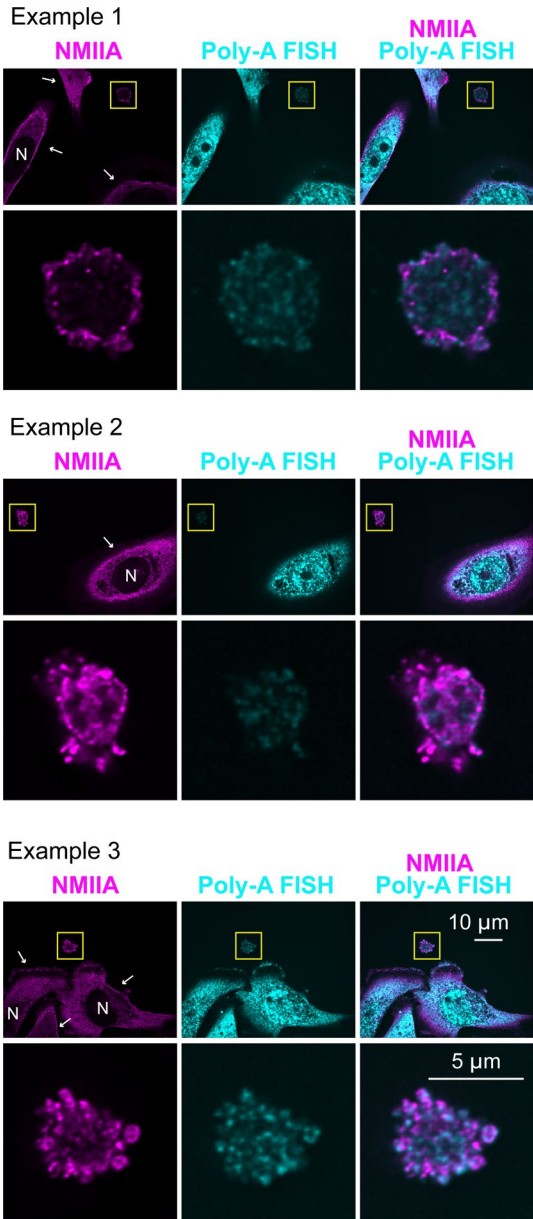

**Extended Data Fig. 6 | Blebbisomes contain RNA.** RNA molecules in cultures of MDA-MB-231 cells were labeled by fluorescence in situ hybridization (FISH) using probes directed towards poly-A sequences (cyan). NMIIA (magenta) was localized to facilitate the identify of blebbisomes. Arrows denote cells and Ns denote nuclei in the field of view. Blebbisomes are denoted by yellow boxes and insets show high magnification images. Images are representative of three independent experiments.

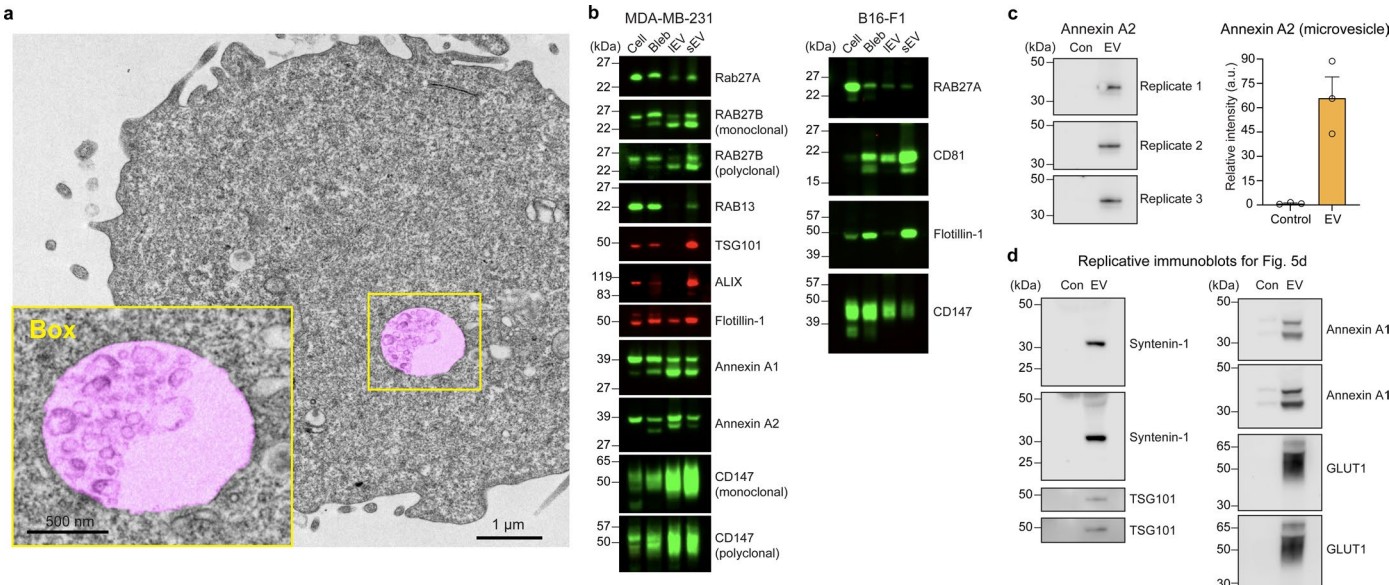

**Extended Data Fig. 7 | Blebbisomes contain RAB proteins and EV marker proteins. a**, TEM micrograph of purified MDA-MB-231 blebbisome displaying a multivesicular endosome. The MVE are pseudo-colored magenta. Box represents enlarged inset of a single MVE displaying multiple intraluminal vesicles. **b**, Immunoblot analysis of select proteins in MDA-MB-231 and B16-F1 cells, purified blebbisomes (bleb), large EVs (lEV) and small EVs (sEV). **c**, Immunoblot analysis of annexin A2-positive microvesicles secreted from MDA-MB-231 blebbisomes. n = 3 and data are displayed as mean ± s.e.m. Images represents three independent experiments. **d**, Replicative immunoblot experiments of EVs secreted from purified MDA-MB-231 blebbisomes. Related to Fig. 5d. Con, control media; EV, secreted EVs from blebbisomes. Source numerical data and unprocessed blots are available in source data.

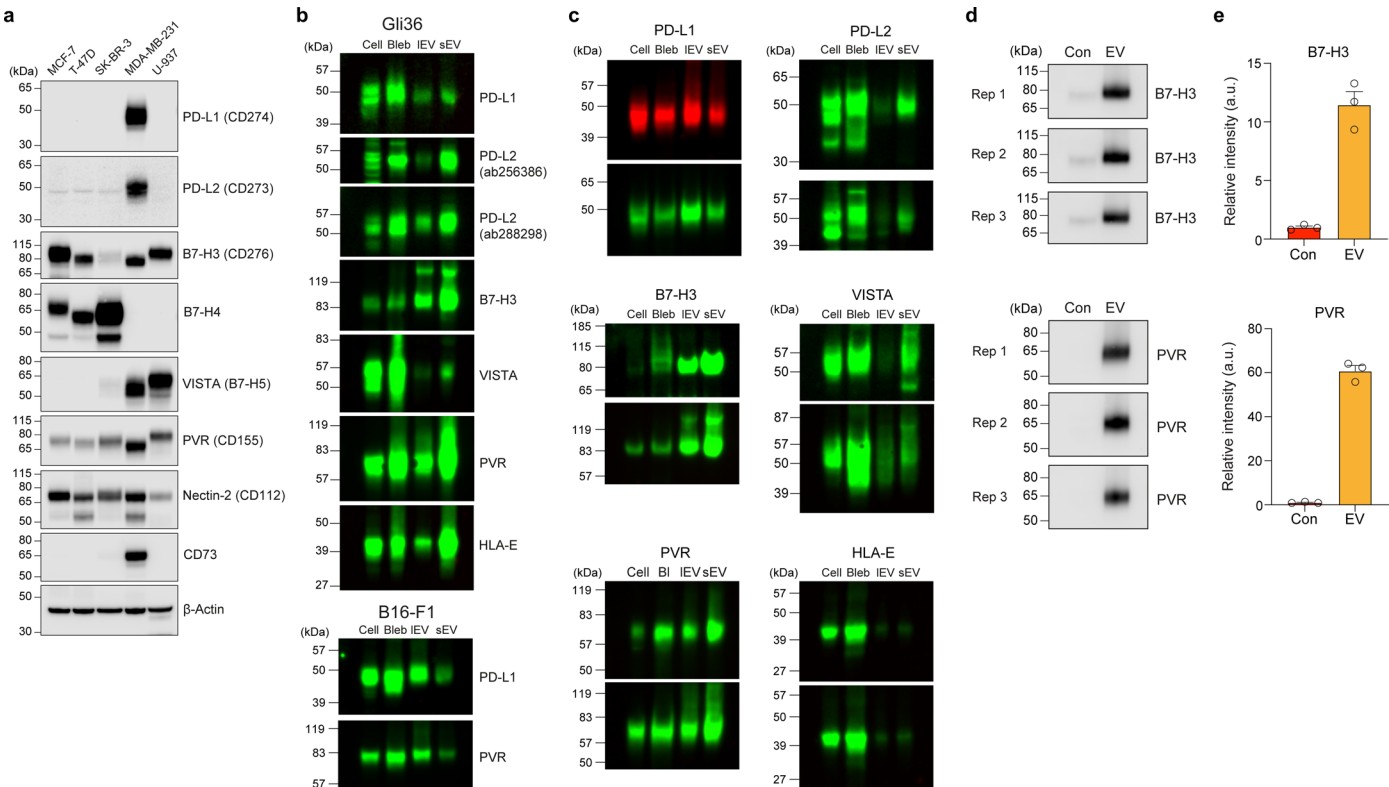

**Extended Data Fig. 8 | Blebbisomes contain inhibitory immune checkpoint ligands. a**, Immunoblot analysis of human breast cancer cells (MCF-7, T-47D, SK-BR-3, MDA-MB-231) and pro-monocytic, human histiocytic lymphoma cells (U-937) for inhibitory immune checkpoint proteins. **b**, Immunoblot analysis of Gli36 and B16 cells, blebbisomes (bleb), large EVs (lEV) and small EVs (sEV) for inhibitory immune checkpoint proteins. **c**, Replicative immunoblot experiments of MDA-MB-231 cells, purified blebbisomes (bleb), large EVs (lEV) and small EVs

(sEV) for immune checkpoint proteins. Related to Fig. 6d, e. **d**, Immunoblot analysis of control media (Con) and EVs secreted from purified MDA-MB-231 blebbisomes for B7-H3 and PVR from three independent experiments. Rep, replicate. **e**, Quantification of relative signal intensity from (d). Each data point represents one independent immunoblot experiment (replicate immunoblots). n = 3 and data are displayed as mean ± s.e.m. Source numerical data and unprocessed blots are available in source data.

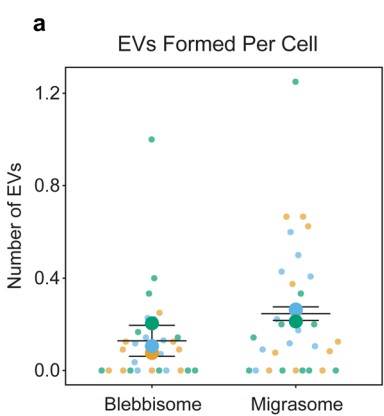

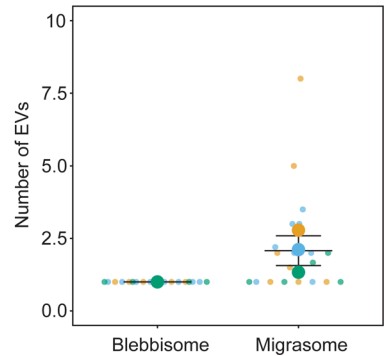

**Extended Data Fig. 9 | Quantification of blebbisome and migrasome formation. a**, Number of EVs formed per cells in each field of view over 24 h. EVs formed, blebbisome mean: 1.13 +/− 0.59 SEM, migrasome mean: 2.77 +/- 1.44 SEM. n = 3 independent experiments. **b**, Number of EVs formed during each cellular retraction event that formed either a blebbisome or migrasomes. EVs formed, blebbisome mean: 1 +/− 0 SEM, migrasome mean: 2.08 +/− 0.51 SEM. n = 3 independent experiments. Quantifications from DIC time-lapse movies. Source numerical data are available in source data.

Dylan T. Burnette

# Reporting Summary

## Statistics

For all statistical analyses, confirm that the following items are present in the figure legend, table legend, main text, or Methods section.

| n/a | Confirmed | |
|---|---|---|
| ☐ | ☒ | The exact sample size (*n*) for each experimental group/condition, given as a discrete number and unit of measurement |
| ☐ | ☒ | A statement on whether measurements were taken from distinct samples or whether the same sample was measured repeatedly |
| ☐ | ☒ | The statistical test(s) used AND whether they are one- or two-sided<br>*Only common tests should be described solely by name; describe more complex techniques in the Methods section.* |
| ☐ | ☒ | A description of all covariates tested |
| ☐ | ☒ | A description of any assumptions or corrections, such as tests of normality and adjustment for multiple comparisons |
| ☐ | ☒ | A full description of the statistical parameters including central tendency (e.g. means) or other basic estimates (e.g. regression coefficient) AND variation (e.g. standard deviation) or associated estimates of uncertainty (e.g. confidence intervals) |
| ☐ | ☒ | For null hypothesis testing, the test statistic (e.g. *F*, *t*, *r*) with confidence intervals, effect sizes, degrees of freedom and *P* value noted<br>*Give P values as exact values whenever suitable.* |
| ☒ | ☐ | For Bayesian analysis, information on the choice of priors and Markov chain Monte Carlo settings |
| ☒ | ☐ | For hierarchical and complex designs, identification of the appropriate level for tests and full reporting of outcomes |
| ☒ | ☐ | Estimates of effect sizes (e.g. Cohen's *d*, Pearson's *r*), indicating how they were calculated |

*Our web collection on statistics for biologists contains articles on many of the points above.*

## Software and code

Policy information about availability of computer code

| Data collection | VisiView Version 5.0.0.27 (Visitron Systems) was used to acquire all iSIM data. |
|---|---|
| Data analysis | FIJI, PRISM, and SuperPlots |

For manuscripts utilizing custom algorithms or software that are central to the research but not yet described in published literature, software must be made available to editors and reviewers. We strongly encourage code deposition in a community repository (e.g. GitHub). See the Nature Portfolio guidelines for submitting code & software for further information.

## Data

Policy information about availability of data

All manuscripts must include a data availability statement. This statement should provide the following information, where applicable:
- Accession codes, unique identifiers, or web links for publicly available datasets
- A description of any restrictions on data availability
- For clinical datasets or third party data, please ensure that the statement adheres to our policy

Mass spectrometry data have been deposited in ProteomeXchange with the primary accession code PXD059407 https://proteomecentral.proteomexchange.org/cgi/GetDataset?ID=PXD059407. All other data supporting the findings of this study are available from the corresponding author on reasonable request.

# Research involving human participants, their data, or biological material

Policy information about studies with human participants or human data. See also policy information about sex, gender (identity/presentation), and sexual orientation and race, ethnicity and racism.

| | |
|---|---|
| Reporting on sex and gender | N/A |
| Reporting on race, ethnicity, or other socially relevant groupings | N/A |
| Population characteristics | N/A |
| Recruitment | N/A |
| Ethics oversight | N/A |

Note that full information on the approval of the study protocol must also be provided in the manuscript.

# Field-specific reporting

Please select the one below that is the best fit for your research. If you are not sure, read the appropriate sections before making your selection.

☒ Life sciences   ☐ Behavioural & social sciences   ☐ Ecological, evolutionary & environmental sciences

For a reference copy of the document with all sections, see nature.com/documents/nr-reporting-summary-flat.pdf

# Life sciences study design

All studies must disclose on these points even when the disclosure is negative.

| | |
|---|---|
| Sample size | Sample sizes for experiments involving individual blebbisome measurements were not specifically calculated. Instead, every blebbisome in each data set was counted and assessed based on the parameters of the experiment. For the TMRE data, the sample size was chosen to be one because of the limitations of live cell imaging as well as likelihood of finding multiple blebbisomes in the same field of view as a cell for comparison. |
| Data exclusions | Data points were not excluded. |
| Replication | Each experiment has at least three biological replicates to provide adequate data and account for variability in sample preparation. Each attempt at replication was successful. |
| Randomization | Randomization is not relevant to this study, for the experiments were not designed to be compared to different treatment groups that would require randomization to eliminate bias. |
| Blinding | Blinding was not possible for these data sets due to the inherent design of the experiments in which microscopy images were taken specifically of blebbisomes and which were then assessed for characteristics of interest. The regions of the sample that were imaged were chosen blindly and blebbisomes were identified after acquisition. |

# Reporting for specific materials, systems and methods

We require information from authors about some types of materials, experimental systems and methods used in many studies. Here, indicate whether each material, system or method listed is relevant to your study. If you are not sure if a list item applies to your research, read the appropriate section before selecting a response.

## Materials & experimental systems

| n/a | Involved in the study |
|---|---|
| ☐ | ☒ Antibodies |
| ☐ | ☒ Eukaryotic cell lines |
| ☒ | ☐ Palaeontology and archaeology |
| ☐ | ☒ Animals and other organisms |
| ☒ | ☐ Clinical data |
| ☒ | ☐ Dual use research of concern |
| ☒ | ☐ Plants |

## Methods

| n/a | Involved in the study |
|---|---|
| ☒ | ☐ ChIP-seq |
| ☒ | ☐ Flow cytometry |
| ☒ | ☐ MRI-based neuroimaging |

# Antibodies

Antibodies used

| | | |
|---|---|---|
| Monoclonal Mouse anti-GM130 | BD Biosciences | 610822 |
| Polyclonal Rabbit anti-NMIIA | BioLegend | PRB-440P |
| Monoclonal Rabbit anti-RPS10 | Abcam | ab151550 |
| Monoclonal Mouse anti-Mitochondria | Abcam | ab92824 |
| Monoclonal Rabbit anti-HSP60 | Cell Signaling Technology | 12165 |
| Monoclonal Rabbit anti-Annexin A1 | Abcam | ab214486 |
| Monoclonal Rabbit anti-Annexin A2 | Abcam | ab178677 |
| Monoclonal Rabbit anti-Syntenin | Abcam | ab133267 |
| Polyclonal Rabbit anti-VDAC2 | Abcam | ab155803 |
| Monoclonal Rabbit anti-NMIIA | Abcam | ab138498 |
| Monoclonal Rabbity anti-RPS8 | Abcam | ab201454 |
| Monoclonal Rabbit anti-Lamin A/C | Abcam | ab108595 |
| Monoclonal Rabbit anti-Lamin A/C | Abcam | ab169532 |
| Monoclonal Rabbit anti-EEF2 | Abcam | ab75748 |
| Monoclonal Rabbit anti-Calreticulin | Abcam | Ab92516 |
| Monoclonal Rabbit anti- Alpha Tubulin | Abcam | Ab52866 |
| Monoclonal Rabbit anti- Cytokeratin 19 | Abcam | Ab52625 |
| Monoclonal Rabbit anti-LC3B | Abcam | Ab192890 |
| Monoclonal Rabbit anti-SQSTM1 / p62 | Abcam | Ab109012 |
| Monoclonal Rabbit anti-LAMP1 | Abcam | Ab108597 |
| Monoclonal Rabbit anti-LAMP1 | Abcam | Ab208943 |
| Monoclonal Rabbit anti-LAMP2 | Abcam | Ab199946 |
| Monoclonal Rabbit anti-GLUT1 | Abcam | Ab115730 |
| Monoclonal Rabbit anti-TSG101 | Abcam | Ab125011 |
| Monoclonal Rabbit anti-Alpha Actinin 4 | Abcam | Ab108198 |
| Monoclonal Rabbit anti-CD63 | Abcam | Ab217345 |
| Monoclonal Rabbit anti-Flotillin 1 | Abcam | Ab133497 |
| Monoclonal Rabbit anti-RAB13 | Abcam | Ab205528 |
| Monoclonal Rabbit anti-CD147 | Abcam | Ab108308 |
| Monoclonal Rabbit anti-CD147 | Abcam | Ab188190 |
| Monoclonal Mouse anti-HLA E | Abcam | Ab2216 |
| Monoclonal Rabbit anti-CD47 | Abcam | Ab300124 |
| Monoclonal Rabbit anti-Nectin 2 | Abcam | Ab135246 |
| Monoclonal Rabbit anti-CD73 | Abcam | Ab133582 |
| Monoclonal Rabbit anti-VISTA | Abcam | Ab300042 |
| Monoclonal Rabbit anti-PVR | Abcam | Ab205304 |
| Monoclonal Rabbit anti-PVR | Abcam | Ab267788 |
| Monoclonal Rabbit anti-B7H4 | Abcam | Ab252438 |
| Monoclonal Rabbit anti-CD276 | Abcam | Ab219648 |
| Monoclonal Rabbit anti-CD276 | Abcam | Ab134161 |
| Monoclonal Rabbit anti-PD-L2 | Abcam | Ab288298 |
| Monoclonal Rabbit anti-PD-L2 | Abcam | Ab256386 |
| Monoclonal Rabbit anti-PD-L1 | Abcam | Ab213480 |
| Monoclonal Rabbit anti-PD-L1 | Abcam | Ab213524 |
| Monoclonal Rabbit anti-VDAC | Cell Signaling Technology | #4661 |
| Monoclonal Rabbit anti-TGOLN2/TGN38 | Cell Signaling Technology | #95649 |
| Monoclonal Rabbit anti-CD81 | Cell Signaling Technology | #10037 |
| Monoclonal Rabbit anti-Rab27A | Cell Signaling Technology | #69295 |
| Monoclonal Rabbit anti-Rab27B | Cell Signaling Technology | #17572 |
| Polyclonal Rabbit anti-Rab27B | Cell Signaling Technology | #44813 |
| Monoclonal Mouse anti-Alix | Cell Signaling Technology | #2171 |
| Monoclonal Rabbit anti-CD73 | Cell Signaling Technology | #13160 |
| Monoclonal Rabbit anti-VISTA | Cell Signaling Technology | #64953 |
| Monoclonal Rabbit anti-PVR | Cell Signaling Technology | #13544 |
| Monoclonal Rabbit anti-PD-L1 | Cell Signaling Technology | #15165 |
| Monoclonal Rabbit anti-PD-L1 | Cell Signaling Technology | #29122 |
| Polyclonal Rabbit anti-CD147 | Thermofisher Scientific | #34-5600 |
| Polyclonal Rabbit anti-CD47 | Thermofisher Scientific | #PA5-116827 |
| Monoclonal Rabbit anti-CD155 | Thermofisher Scientific | #MA5-29762 |
| Monoclonal Mouse anti-β-Actin | Sigma-Aldrich | A5316 |
| Monoclonal Mouse anti-CD29 | BD Transduction Laboratories | 610467 |
| Monoclonal Mouse anti-TSG101 | BD Transduction Laboratories | 612696 |
| Monoclonal Mouse anti-CD63 | BD Transduction Laboratories | 556019 |
| Monoclonal Mouse anti-Flotillin-1 | BD Transduction Laboratories | 610820 |
| Monoclonal Rabbit anti-B7-H3/CD276 | Bethyl Laboratories | #A700-025 |
| Monoclonal Rabbit anti-VISTA | Bethyl Laboratories | #A700-035 |
| Goat anti-rabbit IgG, HRP-linked Antibody | Cell Signaling Technology | #7074 |
| Goat anti-mouse 488 | Life Technologies | A11001 |
| Goat anti-rabbit 488 | Life Technologies | A11034 |
| Goat anti-mouse 568 | Life Technologies | A11004 |
| Goat anti-rabbit 568 | Life Technologies | A11036 |
| Goat anti-mouse 647 | Life Technologies | A32728 |

Goat anti-rabbit 647    Life Technologies        A32733
Donkey anti-Mouse IgG (H+L) Highly Cross-Adsorbed Secondary Antibody, Alexa Fluor™ Plus 680     Thermo Fisher Scientific
A32788
Goat anti-Mouse IgG (H+L) Highly Cross-Adsorbed Secondary Antibody, Alexa Fluor™ Plus 800     Thermofisher Scientific
A32730
Donkey anti-Rabbit IgG (H+L) Highly Cross-Adsorbed Secondary Antibody, Alexa Fluor™ Plus 800     Thermofisher Scientific
A32808

| Validation | Each antibody used for Western blotting was validated by the company from which is was purchased for that application. Each antibody used for immunofluorescence  was validated by the company from which it was purchased for that application. |

# Eukaryotic cell lines

Policy information about cell lines and Sex and Gender in Research

| Cell line source(s) | Human DKO-1 (male) colon cancer ordered from ATCC, human Gli36 glioblastoma ordered from ATCC, human MDA-MB-231 (female) breast cancer cells lines ordered from ATCC, human CCD-18Co (female) colon fibroblast cells ordered from ATCC, murine B16-F1 (male) melanoma cells ordered from ATCC, MV3 melanoma cells from ATCC, and mouse embryonic fibroblasts from Dr. Jennifer Lippincott-Schwartz |
| --- | --- |
| Authentication | The DKO-1, MDA-MB-231, CCD-18Co, and B16-F1 were authenticated by ATCC. The mouse embryonic fibroblasts were not authenticated. |
| Mycoplasma contamination | Cells were tested for mycoplasma via a DAPI stain which did not reveal a non eukaryotic cell specific signal. |
| Commonly misidentified lines (See ICLAC register) | No cells used in this study are on this list. |

# Animals and other research organisms

Policy information about studies involving animals; ARRIVE guidelines recommended for reporting animal research, and Sex and Gender in Research

| Laboratory animals | The zebrafish line LH1066 was used according to institutional ethical guidelines and used from 0-72 hours post fertilization.  8-week-old female C57BL/6J mice were used according to institutional ethical guidelines. |
| --- | --- |
| Wild animals | N/A |
| Reporting on sex | For the bone marrow experiment, all mice used in this study were female. The experiment was repeated three times in which bone marrow was extracted from  a single mouse each time. For zebrafish experiments gender was not a variable. Embryos do not have differentiated gonads and have the potential to develop into either ovaries or testes; a process that does not happen until past 10 days post fertilization. |
| Field-collected samples | N/A |
| Ethics oversight | All animal studies were done in accordance with NIH, the US Department of Agriculture Animal Welfare Act, and the US Public Health Service Policy on Humane Care and Use of Laboratory Animals and were approved by Vanderbilt University Medical Center's Institutional Animal Care and Use Committee. Bone marrow from mice were collected according to M1800191-01. Zebrafish embryo experiments were conducted in accordance with M2100073-00-S2300172. |

Note that full information on the approval of the study protocol must also be provided in the manuscript.

