## [Peer Review File · Nature Cell Biology]

Blebbisomes are large, organelle-rich extracellular vesicles with cell-like properties

Corresponding Author: Dr Dylan Burnette

Version 0:

Decision Letter:

*Please delete the link to your author homepage if you wish to forward this email to co-authors.

Dear Dr Burnette,

Thank you for submitting your manuscript, "Blebbisomes: organelle-rich extracellular vesicles with cell-like properties that take up and secrete extracellular vesicles", to Nature Cell Biology. Your manuscript has now been seen by 2 referees, who are experts in EVs (Referee #2); and EVs (Referee #3). We had initially recruited a third expert (Reviewer #1) but withdrew the reviewer as that expert is not able to provide comments in a timely manner, and we apologize for the delay. As you will see from their comments (attached below), our two reviewers found the work of potential interest but have raised substantial concerns, which in our view would need to be addressed with considerable revisions before we can consider publication in Nature Cell Biology.

Nature Cell Biology editors discuss the referee reports in detail within the editorial team, including the chief editor, to identify key referee points that should be addressed with priority, and requests that are overruled as being beyond the scope of the current study. To guide the scope of the revisions, I have listed these points below. Our standard revision period is six months, and we are committed to providing a fair and constructive peer-review process, so please feel free to contact me if you would like to discuss any of the referee comments further or if you anticipate any issues or delays addressing the reviews.

In particular, in our view, it would be essential to address the following points:

1- Further characterizations of the blebbisomes are needed, including to validate that this is a different type of vesicle than migrasomes and other large EVs:

Rev#2 point#4

Rev#3 "The uniqueness of BLS is quite apparent, but it should be reinforced experimentally by head-to-head comparisons with migrasomes, large oncosomes and exopheres. Basing these comparisons on historical reports by others is not convincing. In this sense the existing comparisons with IEVs and sEVs is useful, but inconclusive."

"Do they enter biofluids? How numerous are they by comparison to other EVs?"

"The evidence as to the existence of BLS in vivo is relatively 'soft'. Images in Fig. 6b are based on a single cancer cell in the Zebrafish embryo, and at that are difficult to decipher. One can argue that long cultured cancer cells implanted into Zebrafish embryos could exhibit features that may or may not be indicative of natural processes in vivo in which BLS could be involved. Can BLS be unambiguously visualised in unperturbed tissues, such as mouse or human cancers, stroma or heart muscle, given the detection of BLS in corresponding cells in vitro."

2- We agree that insight into the mechanisms of formation would be informative, as requested by both reviewers. Definitive mechanistic insights and answers to the reviewers' questions should be provided even if the entire precise mechanism is not fully delineated after revision:

Rev#2 points 1-2

Rev#3 "The analysis of BLS biogenesis is rather descriptive and would benefit from more mechanistic insights. Why do BLS precursor domains detach from parental cells? What are at least some of the underlying molecular mechanisms? Can they

be manipulated and BLS formation perturbed or aborted?"

3- Both reviewers felt that functional analyses would go a long way to help researchers understand what these vesicles are. We agree and suggest you test potential immunosuppressive functions as per Rev#2's point #3 and Rev#3's questions ["What is the biological role of BLS? Do they interact with cells in any way? (...) The discussion of possible immunoregulatory roles of these intriguing EVs is interesting, but presently based on cargo analysis, and thereby rather speculative in the absence of functional data. This could be easily addressed in a similar manner as in several prior visible studies on EV-associated PD-L1 (e.g. Chen et al Nature 2018, Poggio et al Cell 2019; Guan et al Nature Comm. 2022 and others). In the absence of such direct data the lengthy section of the Discussion describing how BLS might affect immunoregulation in cancer is probably unwarranted."]

Rev#2 also asked about the functional contribution of mitochondria in blebbisomes – if this can be directly linked to a function for blebbisomes, it would be an interesting avenue to pursue, but we do not think you will need to incorporate functional data along these lines if immunoregulatory functions can be established.

4- Finally, please pay close attention to our guidelines on statistical and methodological reporting (listed below) as failure to do so may delay the reconsideration of the revised manuscript. In particular, please provide:

We would be happy to consider a revised manuscript that would satisfactorily address these points, unless a similar paper is published elsewhere, or is accepted for publication in Nature Cell Biology in the meantime.

- ensure that it conforms to our format instructions and publication policies (see below and <https://www.nature.com/nature/for-authors>).

- provide a point-by-point rebuttal to the full referee reports verbatim, as provided at the end of this letter.

- provide the completed Reporting Summary (found here <https://www.nature.com/documents/nr-reporting-summary.pdf>). This is essential for reconsideration of the manuscript will be available to editors and referees in the event of peer review. For more information see <http://www.nature.com/authors/policies/availability.html> or contact me.

Nature Cell Biology is committed to improving transparency in authorship. As part of our efforts in this direction, we are now requesting that all authors identified as 'corresponding author' on published papers create and link their Open Researcher and Contributor Identifier (ORCID) with their account on the Manuscript Tracking System (MTS), prior to acceptance. ORCID helps the scientific community achieve unambiguous attribution of all scholarly contributions. You can create and link your ORCID from the home page of the MTS by clicking on 'Modify my Springer Nature account'. For more information please visit www.springernature.com/orcid.

This journal strongly supports public availability of data. Please place the data used in your paper into a public data repository, or alternatively, present the data as Supplementary Information. If data can only be shared on request, please explain why in your Data Availability Statement, and also in the correspondence with your editor. Please note that for some data types, deposition in a public repository is mandatory - more information on our data deposition policies and available

repositories appears below.

Link Redacted

We hope that you will find our referees' comments and editorial guidance helpful. Please do not hesitate to contact me if there is anything you would like to discuss. Thank you again for considering our journal for your work,

Best wishes,

Melina

Melina Casadio, PhD
Senior Editor, Nature Cell Biology
ORCID ID: <https://orcid.org/0000-0003-2389-2243>

Reviewers' Comments:

Reviewer #2:

Remarks to the Author:

This study describes a novel organelle-rich EV subtype referred to as blebbisomes that are released by a spectrum of tumor cell lines and other cell types. This is an interesting finding. However, while data on the existence of blebbisomes in vitro and in vivo is provided in addition to some cargoes that are contained in these structures, the study does not provide insight into the biogenesis and/or function of blebbisomes.

1. How are blebbisomes formed? The structures pointed to in figures 1a and 3a appear to be quite different from that shown in figure 1e. The latter appears to form from a long protrusion or nanotube emanating from the cell body.
2. What are the regulators of blebbisome formation? Do mutants or cellular depletion of resident Rabs affect blebbisome formation?
3. Given the immune checkpoint proteins as molecular cargoes, do blebbisomes have immunosuppressive function?
4. What fraction of the EVs released are blebbisomes?
5. Are mitochondria actively incorporated into blebbisomes. What is the relevance of mitochondria in blebbisomes?

Reviewer #3:

Remarks to the Author:

In this article Jeppesen et al describe a new type of very large (10-20um) extracellular vesicles (EVs), which they term "blebbisomes" (BLS) because of the appearance of blebs on their surfaces. BLS are said to be anuclear, autonomous, mobile cellular fragments that pinch off moving cells when their membranes retract. They are described as having unique protein profiles, more closely resembling cells than smaller EVs. Based on protein markers, imaging (TEM, iSIM) and membrane polarization tests, BLS were found to contain intact and active mitochondria. They also contain Golgi, ER and other organelles, including endosomes and multivesicular bodies (MVBs), which actively release exosomes. BLS also ingest exogenously added exosomes. The consequence of this is unclear at the moment.

BLS appear to differ not only from canonical small EVs, but also from other larger EV subtypes, such as large oncosomes, migrasomes, and exophers, mainly due to their size and unique content of active mitochondria, whereas other large EVs have been described as carrying damaged mitochondria.

It is reported here that BLSs are produced not only by cancer cells, which are the main subject of the study, but also by cultured, non-transformed cells, such as human and mouse fibroblasts and cardiomyocytes. They are also found in Zebrafish embryos injected with cancer cells and in bone marrow cells upon isolation.

While the function of BLS is not directly analysed they are found to be enriched for immune checkpoint and regulatory molecules, such as PD-L1, PD-L2, B7-H3, VISTA, PVR, CD73, HLA-E, the levels of which vary between different cellular sources, and are also found in smaller EVs, but often at lower levels.

Overall, these are quite fascinating and novel observations, and the study appears to be well executed, of high experimental quality and well written. It is very likely that BLS will attract interest of the cell biology and EV research community. However, a few points would require additional experimental and conceptual clarification.

- The analysis of BLS biogenesis is rather descriptive and would benefit from more mechanistic insights. Why do BLS precursor domains detach from parental cells? What are at least some of the underlying molecular mechanisms? Can they be manipulated and BLS formation perturbed or aborted?
- What is the biological role of BLS? Do they interact with cells in any way? Do they enter biofluids? How numerous are they by comparison to other EVs? The discussion of possible immunoregulatory roles of these intriguing EVs is interesting, but presently based on cargo analysis, and thereby rather speculative in the absence of functional data. This could be easily addressed in a similar manner as in several prior visible studies on EV-associated PD-L1 (e.g. Chen et al Nature 2018, Poggio et al Cell 2019; Guan et al Nature Comm. 2022 and others). In the absence of such direct data the lengthy section of the Discussion describing how BLS might affect immunoregulation in cancer is probably unwarranted.
- The evidence as to the existence of BLS in vivo is relatively 'soft'. Images in Fig. 6b are based on a single cancer cell in the Zebrafish embryo, and at that are difficult to decipher. One can argue that long cultured cancer cells implanted into Zebrafish embryos could exhibit features that may or may not be indicative of natural processes in vivo in which BLS could be involved. Can BLS be unambiguously visualised in unperturbed tissues, such as mouse or human cancers, stroma or heart muscle, given the detection of BLS in corresponding cells in vitro.
- The uniqueness of BLS is quite apparent, but it should be reinforced experimentally by head-to-head comparisons with migrasomes, large oncosomes and exopheres. Basing these comparisons on historical reports by others is not convincing. In this sense the existing comparisons with IEVs and sEVs is useful, but inconclusive.
- The authors may wish to discuss their findings in the context of other known complex EVs, such as those described by Petersen et al in JExBio 2023 using endothelial cell cultures as a model. In this case the biogenesis of exosomes and content of MVBs in endothelial bleb derived, large EVs was documented by EM imaging. Expulsion of nuclei from cancer cells has also been described. Another reference point are blood platelets, which are effectively large EVs containing active mitochondria and capable of morphological transformation, including forms of blebbing. Are elements of the megakaryocyte program present in cells producing BLS?

AUTHOR CONTRIBUTIONS – must be included after the Acknowledgements, detailing the contributions of each author to

the paper (e.g. experimental work, project planning, data analysis etc.). Each author should be listed by his/her initials.

FINANCIAL AND NON-FINANCIAL COMPETING INTERESTS – the authors must include one of three declarations: (1) that they have no financial and non-financial competing interests; (2) that they have financial and non-financial competing interests; or (3) that they decline to respond, after the Author Contributions section. This statement will be published with the article, and in cases where financial and non-financial competing interests are declared, these will be itemized in a web supplement to the article. For further details please see <https://www.nature.com/licenceforms/hrg/competing-interests.pdf>.

Methods should be written concisely, but should contain all elements necessary to allow interpretation and replication of the results. As a guideline, Methods sections typically do not exceed 3,000 words. The Methods should be divided into subsections listing reagents and techniques. When citing previous methods, accurate references should be provided and any alterations should be noted. Information must be provided about: antibody dilutions, company names, catalogue numbers and clone numbers for monoclonal antibodies; sequences of RNAi and cDNA probes/primers or company names and catalogue numbers if reagents are commercial; cell line names, sources and information on cell line identity and authentication. Animal studies and experiments involving human subjects must be reported in detail, identifying the committees approving the protocols. For studies involving human subjects/samples, a statement must be included confirming that informed consent was obtained. Statistical analyses and information on the reproducibility of experimental results should be provided in a section titled "Statistics and Reproducibility".

All Nature Cell Biology manuscripts submitted on or after March 21 2016 must include a Data availability statement as a separate section after Methods but before references, under the heading "Data Availability". For Springer Nature policies on data availability see <http://www.nature.com/authors/policies/availability.html>; for more information on this particular policy see <http://www.nature.com/authors/policies/data/data-availability-statements-data-citations.pdf>. The Data availability statement should include:

- Accession codes for primary datasets (generated during the study under consideration and designated as "primary accessions") and secondary datasets (published datasets reanalysed during the study under consideration, designated as "referenced accessions"). For primary accessions data should be made public to coincide with publication of the manuscript. A list of data types for which submission to community-endorsed public repositories is mandated (including sequence, structure, microarray, deep sequencing data) can be found here <http://www.nature.com/authors/policies/availability.html#data>.
- Unique identifiers (accession codes, DOIs or other unique persistent identifier) and hyperlinks for datasets deposited in an approved repository, but for which data deposition is not mandated (see here for details <http://www.nature.com/sdata/data-policies/repositories>).
- At a minimum, please include a statement confirming that all relevant data are available from the authors, and/or are included with the manuscript (e.g. as source data or supplementary information), listing which data are included (e.g. by figure panels and data types) and mentioning any restrictions on availability.
- If a dataset has a Digital Object Identifier (DOI) as its unique identifier, we strongly encourage including this in the Reference list and citing the dataset in the Methods.

We recommend that you upload the step-by-step protocols used in this manuscript to the Protocol Exchange. More details can found at www.nature.com/protocolexchange/about.

All imaging data should be accompanied by scale bars, which should be defined in the legend. Cropped images of gels/blots are acceptable, but need to be accompanied by size markers, and to retain visible background

signal within the linear range (i.e. should not be saturated). The boundaries of panels with low background have to be demarked with black lines. Splicing of panels should only be considered if unavoidable, and must be clearly marked on the figure, and noted in the legend with a statement on whether the samples were obtained and processed simultaneously. Quantitative comparisons between samples on different gels/blots are discouraged; if this is unavoidable, it should only be performed for samples derived from the same experiment with gels/blots were processed in parallel, which needs to be stated in the legend.

Unprocessed scans of all key data generated through electrophoretic separation techniques need to be presented in a supplementary figure that should be labelled and numbered as the final supplementary figure, and should be mentioned in every relevant figure legend. This figure does not count towards the total number of figures and is the only figure that can be displayed over multiple pages, but should be provided as a single file, in PDF or TIFF format. Data in this figure can be displayed in a relatively informal style, but size markers and the figures panels corresponding to the presented data must be

indicated.

The total number of Supplementary Figures (not including the “unprocessed scans” Supplementary Figure) should not exceed the number of main display items (figures and/or tables (see our Guide to Authors and March 2012 editorial <http://www.nature.com/ncb/authors/submit/index.html#suppinfo>; <http://www.nature.com/ncb/journal/v14/n3/index.html#ed>). No restrictions apply to Supplementary Tables or Videos, but we advise authors to be selective in including supplemental data.

Each Supplementary Figure should be provided as a single page and as an individual file in one of our accepted figure formats and should be presented according to our figure guidelines (see above). Supplementary Tables should be provided as individual Excel files. Supplementary Videos should be provided as .avi or .mov files up to 50 MB in size. Supplementary Figures, Tables and Videos much be accompanied by a separate Word document including titles and legends.

GUIDELINES FOR EXPERIMENTAL AND STATISTICAL REPORTING

REPORTING REQUIREMENTS – We are trying to improve the quality of methods and statistics reporting in our papers. To that end, we are now asking authors to complete a reporting summary that collects information on experimental design and reagents. The Reporting Summary can be found here <https://www.nature.com/documents/nr-reporting-summary.pdf>. If you would like to reference the guidance text as you complete the template, please access these flattened versions at <http://www.nature.com/authors/policies/availability.html>.

We strongly recommend the presentation of source data for graphical and statistical analyses as a separate Supplementary Table, and request that source data for all independent repeats are provided when representative experiments of multiple independent repeats, or averages of two independent experiments are presented. This supplementary table should be in Excel format, with data for different figures provided as different sheets within a single Excel file. It should be labelled and numbered as one of the supplementary tables, titled “Statistics Source Data”, and mentioned in all relevant figure legends.

Version 1:

Decision Letter:

*Please delete the link to your author homepage if you wish to forward this email to co-authors.

Dear Dylan,

Thank you again for your patience while your revised manuscript, "Blebbisomes: organelle-rich extracellular vesicles with cell-like properties that take up and secrete extracellular vesicles", was being assessed by the original reviewers. As you know, they had persisting concerns with the depth of the characterizations and analyses. Thank you for taking the time to provide responses to their remarks, which we have now discussed with Rev#2. As you will see from their comments (attached below as well), they find your plan largely reasonable but provided additional feedback. Although we are also very interested in this study, we believe that their concerns should be addressed before we can consider publication in Nature Cell Biology, as we seek strong reviewer support for publication. While we limit our manuscripts to a single round of major experimental revision in order to limit the overall time spent in peer review, given interest and continued support and as the final revisions are relatively minor, we are open to a final round of minor revisions (1-2 months sounds fine).

Please address the remaining reviewer comments as per Rev#2's feedback below in addition to your plans to address Revs#2-3's points. Moreover, as always, please pay close attention to our guidelines on statistical and methodological reporting (listed below) as failure to do so may delay the reconsideration of the revised manuscript. In particular, please provide:

We therefore invite you to take these points into account when revising the manuscript. In addition, when preparing the revision please:

- ensure that it conforms to our format instructions and publication policies (see below and <https://www.nature.com/nature/for-authors>).

- provide a point-by-point rebuttal to the full referee reports verbatim, as provided at the end of this letter.

- provide the completed Reporting Summary (found here <https://www.nature.com/documents/nr-reporting-summary.pdf>). This is essential for reconsideration of the manuscript and will be available to editors and referees in the event of peer review. For more information see <http://www.nature.com/authors/policies/availability.html> or contact me.

Nature Cell Biology is committed to improving transparency in authorship. As part of our efforts in this direction, we are now requesting that all authors identified as 'corresponding author' on published papers create and link their Open Researcher and Contributor Identifier (ORCID) with their account on the Manuscript Tracking System (MTS), prior to acceptance. ORCID helps the scientific community achieve unambiguous attribution of all scholarly contributions. You can create and link your ORCID from the home page of the MTS by clicking on 'Modify my Springer Nature account'. For more information please visit www.springernature.com/orcid.

This journal strongly supports public availability of data. Please place the data used in your paper into a public data repository, or alternatively, present the data as Supplementary Information. If data can only be shared on request, please explain why in your Data Availability Statement, and also in the correspondence with your editor. Please note that for some data types, deposition in a public repository is mandatory - more information on our data deposition policies and available repositories appears below.

Link Redacted

We hope that you will find our referees' comments and editorial guidance helpful. Please do not hesitate to contact me if there is anything you would like to discuss.

Best wishes,

Melina

Melina Casadio, PhD
Senior Editor, Nature Cell Biology
ORCID ID: <https://orcid.org/0000-0003-2389-2243>

Reviewers' Comments:

Reviewer #2:

Remarks to the Author:

In their revised manuscript, Jeppesen and colleagues have provided additional information to characterize blebbisomes, a newly described subset of large extracellular vesicles. They now show that blebbisome formation is affected by knockdown of myosin II but not ESCRT proteins providing some insight into the mechanisms of blebbisome formation. They also provide additional data demonstrating potential roles in immune suppression, and suggest the presence of blebbisomes in bone marrow. These are all interesting findings.

Specific points.

It is appreciated that the authors have provided some evidence to differentiate blebbisomes from migrasomes. However this is described far better in the rebuttal than in the manuscript itself. If the formation of migrasomes in the cells they examine is a rare event, they should state as much. Show quantification of the numbers of blebbisomes relative to migrasomes.

Similar effort to distinguish blebbisomes from large oncosomes will also strengthen this study. This does not necessarily require separation of subpopulations of large EVs and generation of pure populations of each EV type, as the authors suggest. What about labeling cells forming blebbisomes for proteins unique to oncosomes (Minciacchi et al, 2025)?

Figure 3. Mitochondria, autophagosomes, multivesicular bodies, intact Golgi stacks and electron dense lysosomes/residual bodies are readily identified by TEM. This is not the case of other vesicular and tubular elements in cells (early, sorting or recycling endosomes, vesicular and tubular ER and Golgi structures). Further, how frequently are all these organelles identified in blebbisomes? Quantitation should be shown.

The authors suggest that blebbisomes are found in bone marrow. The data however are weak in this regard. How does one distinguish the blebbisome in the image and video from a blebbing BM cell? Are blebbisomes detected in biofluids?

The authors state—"Additionally, we have uncovered evidence that blebbisomes can undergo apoptosis with concurrent generation of EVs, highlighting that not only cells but also blebbisomes may be sources of apoptotic EVs." What is the evidence for this? That blebbisomes are sensitive to apoptosis-inducing reagents does not mean that they are a source of apoptotic EVs. Are apoptotic EVs found in or released from blebbisomes under normal conditions?

ADDITIONAL COMMENTS PROVIDED ON THE AUTHORS' RESPONSES

I've read the author's response and for the most part it seems reasonable. As indicated by both reviewers, it is important that the authors demonstrate how the "blebbisome" is distinct from (or similar to) other similarly sized EVs such as oncosomes and migrasomes. This is a critical point that needs to be addressed. They showed some data to suggest that the blebbisome is a distinct structure from migrasomes. Their additional findings that strengthen this point should also be shown.

Similarly, it would be helpful to morphologically distinguish nascent blebbisomes from oncosomes. Do proteins enriched in blebbisomes (based on their proteomics data) localize to emanating oncosomes? Conversely, do oncosome specific proteins localize to blebbisomes? They could use IF or live cell imaging or EM. They may need to use other cell types, such as prostate cancer cells that secrete oncosomes in abundance to address this point.

The "blebbisome" described here is similar to the large EV described by Petersen et al. (2023). The current study characterizes it further, so distinguishing it from other larger EVs is important, and some solid insight into the biogenesis or function significantly strengthens the study.

Reviewer #3:

Remarks to the Author:

The revised manuscript by Jeppesen et al provides additional valuable data characterising blebbisomes (BLS), a new class of complex extracellular vesicles the authors have recently discovered and characterized. Certainly, some of the mitochondrial death-like processes affecting BLS along with their previously described properties are quite fascinating. The authors also provide a comprehensive rebuttal of editorial and reviewers' critique, and one understands that some of the questions raised previously may take a larger effort to be comprehensively addressed. This said, for a submission to a leading cell biology journal the fundamental properties of BLS, such as key molecular biogenetic mechanisms, distinguishing features compared to other large and complex EVs (e.g. migrasomes or large oncosomes), biological significance and function of BLS in vitro and in vivo should not be "out of scope" of the present paper.

While the authors made some attempts to address these questions, not all of these efforts were entirely convincing. It is true that some of the preparative technologies are not fully developed, for instance to purify large oncosomes (LO) to homogeneity, but it is difficult to agree that these are unsurmountable challenges. For example, LOs are up to 10 μ m in size and therefore should be readily separable from other large EVs in systems where they are produced (Di Vizio et al 2009). Similarly, enrichment of cells expressing TSPAN4 in migrasomes suggests a causative effect of this tetraspanin, and an equivalent of this molecular mechanism for BLS would be quite informative (rather than "misleading"). Does expression of

TSPAN4 affect BLS? In this regard silencing of NMIIB looks promising, but one would wonder whether such a generic targeting of the myosin system would not affect other large EVs, including migrasomes.

Similarly, figure 6 of the revised manuscript continues to be centered on immune checkpoints expressed in BLS, in spite of complete absence of data that BLS possess any immunoregulatory functions. The latter may be difficult to conceptualize given the rarity of these EVs and absence of data that they interact with immune cells. This could have been easily tested. It is also not convincing to say that this question is a subject of an ongoing separate project and will be developed under a separate manuscript. If this is the case, why is fig. 6 included in the present manuscript? Ultimately, the existence and some sort of significant biological impact of BLS in vivo, or their formation as a reflection of some deeper cellular process are presumably the reasons why BLS should be regarded as more than cellular curiosity.

The newly presented pH-induced 'death' of BLS is quite intriguing, but of uncertain biological significance. Here again, as in other segments of the present study, the absence of a compelling reference point, such as another type of large and complex EVs (migrasomes) that may undergo similar fate under comparable conditions impacts the interpretation. Including these sorts of head to head comparisons, would make the present paper, as interesting as it is, still more convincing and mature.

GUIDELINES FOR SUBMISSION OF NATURE CELL BIOLOGY ARTICLES

ARTICLE FORMAT

ABSTRACT – should not exceed 150 words and should be unreferenced. This paragraph is the most visible part of the paper and should briefly outline the background and rationale for the work, and accurately summarize the main results and conclusions. Key genes, proteins and organisms should be specified to ensure discoverability of the paper in online searches.

TEXT – the main text consists of the Introduction, Results, and Discussion sections and must not exceed 3500 words including the abstract. The Introduction should expand on the background relating to the work. The Results should be divided in subsections with subheadings, and should provide a concise and accurate description of the experimental findings. The Discussion should expand on the findings and their implications. All relevant primary literature should be cited, in particular when discussing the background and specific findings.

REFERENCES – are limited to a total of 70 in the main text and Methods combined,. They must be numbered sequentially

as they appear in the main text, tables and figure legends and Methods and must follow the precise style of Nature Cell Biology references. References only cited in the Methods should be numbered consecutively following the last reference cited in the main text. References only associated with Supplementary Information (e.g. in supplementary legends) do not count toward the total reference limit and do not need to be cited in numerical continuity with references in the main text. Only published papers can be cited, and each publication cited should be included in the numbered reference list, which should include the manuscript titles. Footnotes are not permitted.

Methods should be written concisely, but should contain all elements necessary to allow interpretation and replication of the results. As a guideline, Methods sections typically do not exceed 3,000 words. The Methods should be divided into subsections listing reagents and techniques. When citing previous methods, accurate references should be provided and any alterations should be noted. Information must be provided about: antibody dilutions, company names, catalogue numbers and clone numbers for monoclonal antibodies; sequences of RNAi and cDNA probes/primers or company names and catalogue numbers if reagents are commercial; cell line names, sources and information on cell line identity and authentication. Animal studies and experiments involving human subjects must be reported in detail, identifying the committees approving the protocols. For studies involving human subjects/samples, a statement must be included confirming that informed consent was obtained. Statistical analyses and information on the reproducibility of experimental results should be provided in a section titled "Statistics and Reproducibility".

All Nature Cell Biology manuscripts submitted on or after March 21 2016, must include a Data availability statement as a separate section after Methods but before references, under the heading "Data Availability". For Springer Nature policies on data availability see <http://www.nature.com/authors/policies/availability.html>; for more information on this particular policy see <http://www.nature.com/authors/policies/data/data-availability-statements-data-citations.pdf>. The Data availability statement should include:

- Accession codes for primary datasets (generated during the study under consideration and designated as "primary accessions") and secondary datasets (published datasets reanalysed during the study under consideration, designated as "referenced accessions"). For primary accessions data should be made public to coincide with publication of the manuscript. A list of data types for which submission to community-endorsed public repositories is mandated (including sequence, structure, microarray, deep sequencing data) can be found here <http://www.nature.com/authors/policies/availability.html#data>.
- Unique identifiers (accession codes, DOIs or other unique persistent identifier) and hyperlinks for datasets deposited in an approved repository, but for which data deposition is not mandated (see here for details <http://www.nature.com/sdata/data-policies/repositories>).
- At a minimum, please include a statement confirming that all relevant data are available from the authors, and/or are included with the manuscript (e.g. as source data or supplementary information), listing which data are included (e.g. by figure panels and data types) and mentioning any restrictions on availability.
- If a dataset has a Digital Object Identifier (DOI) as its unique identifier, we strongly encourage including this in the Reference list and citing the dataset in the Methods.

We recommend that you upload the step-by-step protocols used in this manuscript to [protocols.io](https://www.protocols.io). More details can be found at <https://www.protocols.io/help/publish-articles>.

DISPLAY ITEMS – main display items are limited to 6-8 main figures and/or main tables. For Supplementary Information see below.

FIGURES – Colour figure publication costs \$395 per colour figure. All panels of a multi-panel figure must be logically connected and arranged as they would appear in the final version. Unnecessary figures and figure panels should be avoided (e.g. data presented in small tables could be stated briefly in the text instead).

All imaging data should be accompanied by scale bars, which should be defined in the legend. Cropped images of gels/blots are acceptable, but need to be accompanied by size markers, and to retain visible background signal within the linear range (i.e. should not be saturated). The boundaries of panels with low background have to be demarked with black lines. Splicing of panels should only be considered if unavoidable, and must be clearly marked on the figure, and noted in the legend with a statement on whether the samples were obtained and processed simultaneously. Quantitative comparisons between samples on different gels/blots are discouraged; if this is unavoidable, it has to be performed for samples derived from the same experiment with gels/blots were processed in parallel, which needs to be stated in the legend.

Figures should be provided at approximately the size that they are to be printed at (single column is 86 mm, double column is 170 mm) and should not exceed an A4 page (8.5 x 11"). Reduction to the scale that will be used on the page is not necessary, but multi-panel figures should be sized so that the whole figure can be reduced by the same amount at the

smallest size at which essential details in each panel are visible. In the interest of our colour-blind readers we ask that you avoid using red and green for contrast in figures. Replacing red with magenta and green with turquoise are two possible colour-safe alternatives. Lines with widths of less than 1 point should be avoided. Sans serif typefaces, such as Helvetica (preferred) or Arial should be used. All text that forms part of a figure should be rewritable and removable.

- For line art, graphs, charts and schematics we prefer Adobe Illustrator (.AI), Encapsulated PostScript (.EPS) or Portable Document Format (.PDF). Files should be saved or exported as such directly from the application in which they were made, to allow us to restyle them according to our journal house style.
- We accept PowerPoint (.PPT) files if they are fully editable. However, please refrain from adding PowerPoint graphical effects to objects, as this results in them outputting poor quality raster art. Text used for PowerPoint figures should be Helvetica (preferred) or Arial.
- We do not recommend using Adobe Photoshop for designing figures, but we can accept Photoshop generated (.PSD or .TIFF) files only if each element included in the figure (text, labels, pictures, graphs, arrows and scale bars) are on separate layers. All text should be editable in 'type layers' and line-art such as graphs and other simple schematics should be preserved and embedded within 'vector smart objects' - not flattened raster/bitmap graphics.
- Some programs can generate Postscript by 'printing to file' (found in the Print dialogue). If using an application not listed above, save the file in PostScript format or email our Art Editor, Allen Beattie for advice (a.beattie@nature.com).

Regardless of format, all figures must be vector graphic compatible files, not supplied in a flattened raster/bitmap graphics format, but should be fully editable, allowing us to highlight/copy/paste all text and move individual parts of the figures (i.e. arrows, lines, x and y axes, graphs, tick marks, scale bars etc). The only parts of the figure that should be in pixel raster/bitmap format are photographic images or 3D rendered graphics/complex technical illustrations.

Unprocessed scans of all key data generated through electrophoretic separation techniques need to be presented in a supplementary figure that should be labeled and numbered as the final supplementary figure, and should be mentioned in every relevant figure legend. This figure does not count towards the total number of figures and is the only figure that can be displayed over multiple pages, but should be provided as a single file, in PDF or TIFF format. Data in this figure can be displayed in a relatively informal style, but size markers and the figures panels corresponding to the presented data must be indicated.

The total number of Supplementary Figures (not including the "unprocessed scans" Supplementary Figure) should not exceed the number of main display items (figures and/or tables (see our Guide to Authors and March 2012 editorial <http://www.nature.com/ncb/authors/submit/index.html#suppinfo>; <http://www.nature.com/ncb/journal/v14/n3/index.html#ed>). No restrictions apply to Supplementary Tables or Videos, but we advise authors to be selective in including supplemental data.

Each Supplementary Figure should be provided as a single page and as an individual file in one of our accepted figure formats and should be presented according to our figure guidelines (see above). Supplementary Tables should be provided

as individual Excel files. Supplementary Videos should be provided as .avi or .mov files up to 50 MB in size. Supplementary Figures, Tables and Videos must be accompanied by a separate Word document including titles and legends.

GUIDELINES FOR EXPERIMENTAL AND STATISTICAL REPORTING

REPORTING REQUIREMENTS – We ask authors to complete a Reporting Summary that collects information on experimental design and reagents. We hope this will aid in your evaluation of the paper. The Reporting Summary can be found here (<https://www.nature.com/documents/nr-reporting-summary.pdf>) Please note that these forms are dynamic 'smart pdfs' and must therefore be downloaded and completed in Adobe Reader. We will then flatten them for ease of use. If you would like to reference the guidance text as you complete the template, please access these flattened versions at (<http://www.nature.com/authors/policies/availability.html>).

Version 2:

Decision Letter:

Our ref: NCB-A52205B

19th November 2024

Dear Dr. Burnette,

Thank you for submitting your revised manuscript "Blebbisomes: organelle-rich extracellular vesicles with cell-like properties that take up and secrete extracellular vesicles" (NCB-A52205B). It has now been seen by the original Referee #2 and their comments are below. The reviewer finds that the paper has improved in revision, and therefore we'll be happy in principle to publish it in Nature Cell Biology, pending minor revisions to comply with our editorial and formatting guidelines.

Please note that the current version of your manuscript is in a PDF format; could you please email us a copy of the file in an editable format (Microsoft Word or LaTeX) as we can not proceed with PDFs at this stage? Many thanks in advance.

Thank you again for your interest in Nature Cell Biology. Please do not hesitate to contact me if you have any questions.

Sincerely,

Melina

Melina Casadio, PhD
Senior Editor, Nature Cell Biology
Consulting Editor, Nature Structural & Molecular Biology
ORCID ID: <https://orcid.org/0000-0003-2389-2243>

Reviewer #2 (Remarks to the Author):

This study reports on a previously undescribed EV subtype, referred to by the authors, as blebbisomes. The effort made to distinguish blebbisomes from previously described, large EVs of >1 micron size, will be helpful to future investigations. All points I had raised have been addressed.

Version 3:

Decision Letter:

Dear Dr Burnette,

I am pleased to inform you that your manuscript, "Blebbisomes are large, organelle-rich extracellular vesicles with cell-like properties", has now been accepted for publication in Nature Cell Biology.

Please note that *Nature Cell Biology* is a Transformative Journal (TJ). Authors may publish their research with us through the traditional subscription access route or make their paper immediately open access through payment of an article-processing charge (APC). Authors will not be required to make a final decision about access to their article until it has been accepted. [Find out more about Transformative Journals](https://www.springernature.com/gp/open-research/transformative-journals)

If you have not already done so, we strongly recommend that you upload the step-by-step protocols used in this manuscript to protocols.io (<https://protocols.io>), an open online resource that allows researchers to share their detailed experimental know-how. All uploaded protocols are made freely available and are assigned DOIs for ease of citation. Protocols and Nature Portfolio journal papers in which they are used can be linked to one another, and this link is clearly and prominently visible in the online versions of both. Authors who performed the specific experiments can act as primary authors for the Protocol as they will be best placed to share the methodology details, but the Corresponding Author of the present research paper should be included as one of the authors. By uploading your Protocols onto protocols.io, you are enabling researchers to more readily reproduce or adapt the methodology you use, as well as increasing the visibility of your protocols and papers. You can also establish a dedicated workspace to collect your lab Protocols. Further information can be found at <https://www.protocols.io/help/publish-articles>.

Nature Cell Biology encourages authors presenting evidence for cell, biological, molecular, and genetic interactions to consider communicating these findings using Biofactoid (<https://biofactoid.org/>). This tool helps users share a searchable representation of interactions (e.g. binding, gene expression, post-translational modification) between genes, gene products, or chemicals. Information added to Biofactoid, with author attribution, is shared on social media and public databases, such as Pathway Commons, where it can be discovered and analyzed in the context of a large and growing corpus of knowledge.

With kind regards,

Melina Casadio, PhD
Senior Editor, Nature Cell Biology
Consulting Editor, Nature Structural & Molecular Biology
ORCID ID: <https://orcid.org/0000-0003-2389-2243>

** Visit the Springer Nature Editorial and Publishing website at http://editorial-jobs.springernature.com?utm_source=ejp_NCB_email&utm_medium=ejp_NCB_email&utm_campaign=ejp_NCB for more information about our career opportunities. If you have any questions please click [here](mailto:editorial.publishing.jobs@springernature.com).**

RESPONSE TO THE REVIEWERS

We would like to express our gratitude to the reviewers for their useful comments and insightful questions. Their feedback not only guided the revisions of this manuscript but also sparked spirited discussions in our lab regarding the future research directions for blebbsomes. Below is a summary of the revisions we have implemented during this period. With your valuable guidance, we believe that the manuscript has been significantly enhanced and now presents a clearer, more comprehensive exploration of the unique properties and potential functions of blebbsomes.

Specific queries from the editors:

1- Further characterizations of the blebbsomes are needed, including to validate that this is a different type of vesicle than migrasomes and other large EVs:

In order to further differentiate blebbsomes from migrasomes, we are including two supplementary movies that show the formation of migrasomes. In our hands, this was an incredibly rare event to observe. Previous studies have relied on the overexpression of tetraspanin-4 to induce the formation of migrasomes in vitro. The multiple cell lines that we present here do not express tetraspanin-4 at high levels. Our new data shows migrasomes forming from thin retraction fibers in which the migrasome blows up like a balloon after the retraction fiber is created, as previously reported. Importantly for comparisons with blebbsomes, our data shows that migrasomes are and non-motile. Video 12 shows migrasomes forming, and video 13 shows a rare field of view in which a blebbsome is present during the formation of migrasomes. As previously reported, a single retraction fiber can form multiple migrasomes. On the other hand, we have never observed multiple blebbsomes being formed at any one time. Indeed, we included in the original submission a movie in which we thought multiple blebbsomes would be formed; however, in the end, only one blebbsome formed (video 5). Given their smaller size and lack of motility, we feel that it is highly unlikely that we can group migrasomes and blebbsomes as the same kind of extracellular vesicle. Furthermore, the organelle composition is different between the two extracellular vesicles. For example, only a subset of migrasomes have mitochondria as reported in Jiao et al. Cell 2021, and those that do have these organelles appear to only have damaged mitochondria.

The point that reviewer 2 is making is not yet possible based on the existing protocols available for extracellular vesicle isolation. There have only been a handful of purification techniques for extracellular vesicles, and these only focus on the purification of "small EVs" and "large EVs". Although there are ways of isolating the groups of large EVs and small EVs, these methods are not well suited to specifically interrogating percentages of subsets of specific subtypes of EVs within these groups. To be able to address and provide data for this question, more advanced techniques would need to be developed that allow for simultaneous isolation of multiple EVs that can unambiguously purify one type of large EV away from another. We think we can reliably generate relatively pure samples of blebbsomes which we can then compare directly to large EVs overall, and small EVs, but we are not confident that pure samples of large oncosome, migrasomes, and exophers can be generated with current methods and published protocols. We are therefore opting for what we think is the most honest biochemical characterizations that can currently be made (blebbsomes vs. large EVs and blebbsome vs. small EVs) (see also our reply just below for more discussion on this topic).

Rev#3 " The uniqueness of BLS is quite apparent, but it should be reinforced experimentally by head-to-head comparisons with migrasomes, large oncosomes and exophers. Basing these comparisons on historical reports by others is not convincing. In this sense the existing comparisons with IEVs and sEVs is useful, but inconclusive."

We are pleased that the reviewer recognizes the uniqueness of blebbisomes compared to other types of IEVs. We also agree with the reviewer that such detailed biochemical comparisons between subpopulations of large to very large EVs would be very valuable. Unfortunately, we do not think it is currently technically possible to obtain pure populations of exophers, migrasomes and large oncosomes that could directly be compared to our purified blebbisomes. Current protocols for purification of exophers does not exist. We have asked the corresponding author on the initial discovery paper on exophers if their lab had protocols for specific isolation of pure exophers that could be used for biochemical analysis of exphors (proteomics, immunoblot etc.) but unfortunately no such protocol has been developed or validated at this point in time. Some proteomics for migrasomes have been published but like for all studies on migrasomes, has relied on overexpression of TSPAN4 to obtain enough migrasomes for study. We have observed very rare release of endogenously produced migrasomes (new videos 12 and 13) and it is now clear to us that it would not be feasible to purify enough endogenous migrasomes for biochemical analysis. Overexpression of TSPAN4 to induce artificial production of migrasomes may produce changes in protein expression and abundance in migrasomes, which to our mind, puts an asterisk on known protein content of migrasomes. We could in principle compare our blebbisomes to those published migrasome proteomics but we feel the value of that might be limited and could be potentially misleading. Protocols for large oncome preparations are similar to protocols for IEVs and will produce mixed samples of different IEVs. In light of this, we therefore feel that comparing blebbisomes by proteomics and immunoblots to purified IEVs and sEVs is the best that can technically currently be done, and crucially, therefore also the most honest comparison possible at this time.

“Do they enter biofluids?”

Although we have not yet technically worked out how to sufficiently enrich blebbisomes in biofluids such as blood for unambiguous identification to the same standard as we apply to cultured cells, we are confident that endogenous blebbisomes are present in mammalian bone marrow.

How numerous are they by comparison to other EVs?”

The reviewer raises an interesting but technically very difficult question to address. We can confidently say that blebbisome production is quantitatively much more common than release of endogenous migrasomes which in our hands turned out to be very rare events (new Videos 12 and 13). But we do now have at least enumeration of blebbisome release per cell over a 48-hour time period (see new figure below; Extended Data Figure 3a). Unlike blebbisome, we cannot directly observe, or follow release, of large oncosomes (or exophers) in real time, and purified large oncosome preparations are actually a mix of different types of large EVs further complicating assessment. We can say that compared to the total amount of “generic” (mixed populations) of all large or very large EVs (IEVs), blebbisomes are likely to be only a small subset. In particular, 100 – 1000 nm ectosomes (“microvesicles”) will be the numerically dominant type of IEVs in any sample mix we have so far observed. And of course blebbisomes are far less numerous in the extracellular environment than smaller, generic, sEVs. Numerically, then, blebbisomes are likely to represent a relatively small portion of all EVs released although based on our proteomic analyses blebbisome can represent a significant source of specific EV proteins that might otherwise have been ascribed to other types of EVs in the past.

“The evidence as to the existence of BLS in vivo is relatively ‘soft’. Images in Fig. 6b are based on a single cancer cell in the Zebrafish embryo, and at that are difficult to decipher. One can argue that long cultured cancer cells implanted into Zebrafish embryos could exhibit features that may or may not be indicative of natural processes in vivo in which BLS could be involved. Can BLS be unambiguously visualised in unperturbed tissues, such as mouse or human cancers, stroma or heart muscle, given the detection of BLS in corresponding cells in vitro.”

We agree with the reviewer that it would be significant to visualize blebbisomes. We attempted experiments in zebrafish using neural crest cells expressing mEGFP but found it difficult to distinguish putative blebbisomes from cells as we did not have the ability to localize nuclei and mitochondria. It became clear that we could not produce such data in a reasonable timeframe. However, this realization that we need more molecular markers led us to reexamine the strength of our other evidence that blebbisomes exist in vivo. In the original submission, we showed a montage and a movie of a blebbisome that we purified from bone marrow. This was done using only DIC and had no molecular markers. Given that we are culturing bone marrow, it was straightforward to label by DNA to detect cell nuclei and mitochondria. The revision now includes new data in the final figure (shown below) and a new supplemental video (Video 11). Bone marrow has cells with nuclei and mitochondria, as well as cell types that lack both (e.g., red blood cells), while blebbisomes lack a nucleus but have mitochondria. We feel this new labeling method makes it clearer that bone marrow isolated from mice contains blebbisomes.

c Blebbisomes from bone marrow

2- We agree that insight into the mechanisms of formation would be informative, as requested by both reviewers. Definitive mechanistic insights and answers to the reviewers' questions should be provided even if the entire precise mechanism is not fully delineated after revision:

We have now started the process of investigating the mechanistic details of blebbisome biogenesis and will include what insight we have gained in the revised version of the manuscript, although a complete description of all mechanistic steps in formation of blebbisomes is outside the scope of this article. We now know that the molecular motor that facilitates the formation of large cellular retractions, NMIIB, is required for the formation of blebbisomes while the ESCRT-III machinery that is involved in formation and scission of membrane vesicles at multivesicular endosomes and the plasma membrane is not necessary to form and release blebbisomes. (please see the detailed response to Reviewer 3, and new figure, below). The reviewers also questioned the proportion of blebbisomes among other EVs released from cells and whether we could establish a more direct comparison with other large EVs such as migrasomes, large oncosomes, and exophers. While current techniques allow for the isolation of mixed large EV populations, they do not support the precise quantification of each subtype. However, we provided a comparison based on available proteomic data and our newly added supplementary videos demonstrating the rarity of migrasome formation and their distinct characteristics from blebbisomes. We also highlighted that the overexpression techniques used in other studies to induce migrasome formation might alter their native properties, thus making direct comparisons potentially misleading.

Rev#3 "The analysis of BLS biogenesis is rather descriptive and would benefit from more mechanistic insights.

Why do BLS precursor domains detach from parental cells?

What are at least some of the underlying molecular mechanisms?

Can they be manipulated and BLS formation perturbed or aborted?"

The reviewer raises an important question. Although a complete investigation of blebbisome biogenesis is out of the scope of what can be accomplished in this manuscript we have now performed some experiments to broach the subject. Through the use of siRNA, we knocked down the expression of non-

muscle myosin IIB (NMIIB) which is a contractile protein of the myosin superfamily involved in cellular motility and retraction. Cells that were knocked down were then plated and left to move and grow for 48 hours after which they were stained for both actin and nuclei to identify blebbisomes from cells. We found that cells lacking expression of NMIIB, which inhibits the formation of large retraction fibers, produced significantly less blebbisomes compared to siRNA control cells highlighting the importance of this protein in blebbisome biogenesis. We also performed siRNA knockdown of CHMP2A and CHMP4B, two ESCRT-III proteins that mediates budding and scission of membranes to form vesicles, both at MVEs and the plasma membrane. However, neither knockdown of a single protein or both proteins produced a reduction in blebbisome formation, indicating that ESCRT-III is likely not required for blebbisome formation. We have included this data in the revised manuscript and for convenience we reproduce the new figure here below:

3- Both reviewers felt that functional analyses would go a long way to help researchers understand what these vesicles are. We agree and suggest you test potential immunosuppressive functions as per Rev#2's point #3 and Rev#3's questions ["What is the biological role of BLS? Do they interact with cells in any way? (...) The discussion of possible immunoregulatory roles of these intriguing EVs is interesting, but presently based on cargo analysis, and thereby rather speculative in the absence of functional data. This could be easily addressed in a similar manner as in several prior visible studies on EV-associated PD-L1 (e.g. Chen et al Nature 2018, Poggio et al Cell 2019; Guan et al Nature Comm. 2022 and others). In the absence of such direct data the lengthy section of the Discussion describing how BLS might affect immunoregulation in cancer is probably unwarranted."]

We thank the reviewers for highlighting an important direction. We agree that question as to what the potential immunoregulatory role blebbisomes play is of interest. While we considered this question it became apparent that the scope was large enough, not to just be its own study, but a series of studies. As

such, we are now working on a grant proposal to fund these studies as this lies well outside the scope and funds of our current grants. We acknowledge that the amount of text we used in the original manuscript describing the potential connections between the contents of blebosomes and the immune system was unwarranted given the lack of mechanistic data. As such, we have substantially reduced this section.

Rev#2 also asked about the functional contribution of mitochondria in blebosomes – if this can be directly linked to a function for blebosomes, it would be an interesting avenue to pursue, but we do not think you will need to incorporate functional data along these lines if immunoregulatory functions can be established.

This question led to some interesting conversations in our lab. Mitochondria are well known as hubs for cellular decision making. We thought that it would be significant if we could test if mitochondria could be making cell-like decisions in blebosomes. We had preliminary unpublished data suggesting that blebosomes could “die”. We were previously interested in whether cellular blebs made by cells in different contexts were different and were comparing blebbing during cytokinesis and during cell death. One of the ways we induced cell death was simply by increasing the pH of the cell culture media. While we did find any obvious differences in the dynamics of membrane blebs one of the movies did get picked by the Nikon Smallworld imaging contest: <https://www.nikonsmallworld.com/galleries/2022-small-world-in-motion-competition/dying-melanoma-cells>

In this movie, you will see that DIC reveals when the cells lose the integrity of their membranes when they die. We also had some blebosomes in the fields of view in these experiments and they also lost the integrity of their membranes. While pH changes were not specific to mitochondria, these data, along with the reviewer’s question, convinced us to design a series of experiments to test whether mitochondria could induce blebbsiomes to die. There are several small molecules that target mitochondria and induced apoptosis. We tested there and found that they caused blebbsiomes to die. This data (shown below) is now included in Figure 1, and adds another unique function to blebbsiomes: cell-like decisions.

4- Finally, please pay close attention to our guidelines on statistical and methodological reporting (listed below) as failure to do so may delay the reconsideration of the revised manuscript. In particular, please provide:

We have included labeled unprocessed images of all blots as a supplementary PDF file.

- a Supplementary Table including all numerical source data in Excel format, with data for different figures provided as different sheets within a single Excel file. The file should include source data giving rise to graphical representations and statistical descriptions in the paper and for all instances where the figures

present representative experiments of multiple independent repeats, the source data of all repeats should be provided.

We have included this supplementary table.

We would be happy to consider a revised manuscript that would satisfactorily address these points, unless a similar paper is published elsewhere, or is accepted for publication in Nature Cell Biology in the meantime.

Specific comments from the reviewers:

Reviewer #2:

Remarks to the Author:

This study describes a novel organelle-rich EV subtype referred to as blebbisomes that are released by a spectrum of tumor cell lines and other cell types. This is an interesting finding. However, while data on the existence of blebbisomes in vitro and in vivo is provided in addition to some cargoes that are contained in these structures, the study does not provide insight into the biogenesis and/or function of blebbisomes.

1. How are blebbisomes formed? The structures pointed to in figures 1a and 3a appear to be quite different from that shown in figure 1e. The latter appears to form from a long protrusion or nanotube emanating from the cell body.

We apologize for the confusion. Figures 1a and 3a show blebbisomes that were already formed when the experiment started, whereas Figure 1e shows the formation of a blebbisome by the exact mechanism suggested by the reviewer. The supplemental movies also show several examples of blebbisomes forming from such a cellular retraction event. The dynamics and morphology of the retraction led us to predict that two different classes of proteins could be driving blebbisome formation, as addressed in the response to the reviewer's next question

2. What are the regulators of blebbisome formation?

We predicted that two families of proteins could be responsible for the release of blebbisomes from cells. Large cellular retractions are driven by the molecular motor myosin II, with myosin IIB being a major family member in that process. Our data now show that depletion of myosin IIB from cells significantly reduces blebbisome formation. The ESCRT family of proteins was another set of potential players. Among other functions, these proteins drive the scission of the midbody after cytokinesis. However, depletion of major ESCRT-III family members in cells did not reduce blebbisome formation. Even though this is negative data, we chose to include it in the manuscript as the lack of involvement of of ESCRT-III machinery is informative for the field. We also acknowledge that this is only scratching the surface of the proteins that could be involved in blebbisome formation and that future dedicated studies are required for a more complete description of the regulators of blebbisome formation and biogenesis.

Do mutants or cellular depletion of resident Rabs affect blebbisome formation?

This is an intriguing question and is part of a larger ongoing study in the lab. In our current manuscript, we report that blebbisomes contain numerous organelles and possess the machinery required for membrane trafficking. However, how organelles, membrane vesicles, or cytoskeletal components are distributed to the site where a blebbisome is going to form is completely unknown. We predict that there will be directed traffic into the blebbisome. Consequently, we are now systematically testing how each organelle enters the blebbisome, starting with simple questions (e.g., are components moved down the retraction fiber to the end?). We will begin by identifying which Rabs are involved using a similar descriptive

methodology, and then deplete the Rabs that specifically move to forming blebbisomes.

3. Given the immune checkpoint proteins as molecular cargoes, do blebbisomes have immunosuppressive function?

We appreciate the reviewer for emphasizing a significant research direction. The potential immunoregulatory roles of blebbisomes present a compelling avenue of inquiry and we fully agree that potential immunosuppressive functions of blebbisomes should be a high priority for investigation. However, upon reflection, we realized that addressing this would extend beyond a single study. Consequently, we are in the process of drafting a grant proposal to support this extensive research, as it exceeds the scope and funding of our current projects. We acknowledge that the amount of text we used in the original manuscript discussing the contents of blebbisomes and the immune system was excessive, and we have therefore substantially reduced discussion of this topic in the revised manuscript.

4. What fraction of the EVs released are blebbisomes?

As outlined above in more detail, this is a difficult question to address as it is not possible to determine the numbers of all types of known EVs released from a particular population of cells. However, we can quantify, and now report, the number of blebbisomes released per cell. We present this data in the new Extended Data Figure 3. We used the number of blebbisomes released from cells to compare the effects of depleting myosin IIB and ESCRT proteins.

5. Are mitochondria actively incorporated into blebbisomes.

We are actively investigating whether mitochondria and other organelles are actively trafficked to the site of blebbisome formation. If they are, we think they are likely to be using specific kinesins and/or myosins. This is part of the larger project described above.

What is the relevance of mitochondria in blebbisomes?

Membrane blebbing and motility burn through ATP, which requires active mitochondria to replenish. This is what led us to originally test if the mitochondria were as active as they are in cells. We now present new data that suggests that mitochondria in blebbisomes can also facilitate cell-like decisions. Specifically, mitochondria can trigger apoptosis in blebbisomes.

Reviewer #3:

Remarks to the Author:

In this article Jeppesen et al describe a new type of very large (10-20um) extracellular vesicles (EVs), which they term "blebbisomes" (BLS) because of the appearance of blebs on their surfaces. BLS are said to be anuclear, autonomous, mobile cellular fragments that pinch off moving cells when their membranes retract. They are described as having unique protein profiles, more closely resembling cells than smaller EVs. Based on protein markers, imaging (TEM, iSIM) and membrane polarization tests, BLS were found to contain intact and active mitochondria. They also contain Golgi, ER and other organelles, including endosomes and multivesicular bodies (MVBs), which actively release exosomes. BLS also ingest exogenously added exosomes. The consequence of this is unclear at the moment.

BLS appear to differ not only from canonical small EVs, but also from other larger EV subtypes, such as large oncosomes, migrasomes, and exophers, mainly due to their size and unique content of active

mitochondria, whereas other large EVs have been described as carrying damaged mitochondria. It is reported here that BLSs are produced not only by cancer cells, which are the main subject of the study, but also by cultured, non-transformed cells, such as human and mouse fibroblasts and cardiomyocytes. They are also found in Zebrafish embryos injected with cancer cells and in bone marrow cells upon isolation.

While the function of BLS is not directly analysed they are found to be enriched for immune checkpoint and regulatory molecules, such as PD-L1, PD-L2, B7-H3, VISTA, PVR, CD73, HLA-E, the levels of which vary between different cellular sources, and are also found in smaller EVs, but often at lower levels.

Overall, these are quite fascinating and novel observations, and the study appears to be well executed, of high experimental quality and well written. It is very likely that BLS will attract interest of the cell biology and EV research community. However, a few points would require additional experimental and conceptual clarification.

- The analysis of BLS biogenesis is rather descriptive and would benefit from more mechanistic insights. Why do BLS precursor domains detach from parental cells? What are at least some of the underlying molecular mechanisms? Can they be manipulated and BLS formation perturbed or aborted?

Thank you for this question. We have added new data in which we address the potential protein families involved in blebbisome formation. We acknowledge that we are only scratching the surface of the numerous potential molecular players in this process. Since blebbisomes are formed from large cellular retractions, we first investigated whether the family of molecular motors responsible for driving these retractions, myosin II, plays a role in blebbisome formation. Myosin IIB is the myosin II family member responsible for retractions, so we depleted it from cells and found that it reduced blebbisome formation. We then predicted that the ESCRT machinery would also be required for blebbisome formation, as the nanotube from which the blebbisome detaches is similar in morphology to the midbody. ESCRTs are responsible for the scission of the midbody. Unfortunately, knocking down the highly expressed ESCRTs in our cells did not significantly reduce the production of blebbisomes. While this is negative data, we have included it in the revision in case anyone has a similar hypothesis in the future.

- What is the biological role of BLS? Do they interact with cells in any way? Do they enter biofluids? How numerous are they by comparison to other EVs? The discussion of possible immunoregulatory roles of these intriguing EVs is interesting, but presently based on cargo analysis, and thereby rather speculative in the absence of functional data. This could be easily addressed in a similar manner as in several prior visible studies on EV-associated PD-L1 (e.g. Chen et al Nature 2018, Poggio et al Cell 2019; Guan et al Nature Comm. 2022 and others). In the absence of such direct data the lengthy section of the Discussion describing how BLS might affect immunoregulation in cancer is probably unwarranted.

We agree that the lengthy discussion about how blebbisomes might affect immunoregulation given that we only report the cargo analysis. As such, we have substantially reduced this speculation. We included it originally as we do think a role in the immune system will be the next likely step in studying blebbisomes. As such, we are currently writing grants to try to fund those investigations. While we think that immunoregulation is the most assessable potential blebbisome function to study, the list of functions is likely to be a long one.

- The evidence as to the existence of BLS in vivo is relatively 'soft'. Images in Fig. 6b are based on a single cancer cell in the Zebrafish embryo, and at that are difficult to decipher. One can argue that long cultured cancer cells implanted into Zebrafish embryos could exhibit features that may or may not be indicative of

natural processes in vivo in which BLS could be involved. Can BLS be unambiguously visualised in unperturbed tissues, such as mouse or human cancers, stroma or heart muscle, given the detection of BLS in corresponding cells in vitro.

We agree with the reviewer that visualizing blebbisomes in native unperturbed tissues would be significant. We attempted experiments in zebrafish using neural crest cells expressing mEGFP, but we found it difficult to distinguish putative blebbisomes from cells due to the inability to localize nuclei and mitochondria. It became evident that producing such data in a reasonable timeframe was not feasible. However, this realization highlighted the need for more molecular markers, prompting us to reexamine the strength of our existing evidence that blebbisomes exist in vivo. In the original submission, we presented a montage and a movie of a blebbisome purified from bone marrow, using only DIC without molecular markers. Given that we are culturing bone marrow, it was straightforward to label DNA to detect cell nuclei and mitochondria. The revision now includes new data in the final figure and a new supplemental video (Video 11). Bone marrow contains cells with nuclei and mitochondria, as well as cell types that lack both (e.g., red blood cells), while blebbisomes lack a nucleus but have mitochondria. We believe this new labeling method makes it clearer that bone marrow isolated from mice contains blebbisomes

- The uniqueness of BLS is quite apparent, but it should be reinforced experimentally by head-to-head comparisons with migrasomes, large oncosomes and exophers. Basing these comparisons on historical reports by others is not convincing. In this sense the existing comparisons with IEVs and sEVs is useful, but inconclusive.

We are pleased that the reviewer recognizes the uniqueness of blebbisomes compared to other types of large extracellular vesicles (IEVs). We also agree with the reviewer that detailed biochemical comparisons between subpopulations of large to very large EVs would be highly valuable. Unfortunately, we do not believe it is currently technically possible to obtain pure populations of exophers, migrasomes, and large oncosomes that could be directly compared to our purified blebbisomes. At present, protocols for the purification of exophers do not exist. We contacted the corresponding author of the initial discovery paper on exophers to inquire if their lab had developed protocols for the specific isolation of pure exophers for biochemical analysis (e.g., proteomics, immunoblot), but no such protocol has been developed or validated at this time.

Some proteomics studies on migrasomes have been published, but like all studies on migrasomes, they have relied on the overexpression of TSPAN4 to obtain sufficient amounts of migrasomes. We have observed that the release of endogenously produced migrasomes is very rare (new videos 12 and 13), making it clear that purifying enough endogenous migrasomes for biochemical analysis is not feasible. Overexpression of TSPAN4 to induce artificial production of migrasomes may alter protein expression and abundance, which in our opinion, casts doubt on the known protein content of migrasomes. While we could compare our blebbisomes to published migrasome proteomics data, we believe the value of such a comparison might be limited and potentially misleading.

Protocols for preparing large oncosomes are similar to those for IEVs, resulting in mixed samples of different IEVs. In light of this, we believe that comparing blebbisomes by proteomics and immunoblots to purified IEVs and small EVs (sEVs) is the best approach that can currently be taken, and crucially, it is also the most honest comparison possible at this time.

- The authors may wish to discuss their findings in the context of other known complex EVs, such as those described by Petersen et al in JExBio 2023 using endothelial cell cultures as a model. In this case the biogenesis of exosomes and content of MVBs in endothelial bleb derived, large EVs was documented by EM imaging. Expulsion of nuclei from cancer cells has also been described. Another reference point are

blood platelets, which are effectively large EVs containing active mitochondria and capable of morphological transformation, including forms of blebbing. Are elements of the megakaryocyte program present in cells producing BLS?

The reviewer raises the point that other studies have documented that MVBs can be released from cells by various mechanisms and gives the example of Petersen et al, JExBio 2023. Another example is the *en bloc* release of MVB-like small extracellular vesicle clusters (Valcz et al. JEV 2019). Our reading of the literature is that these events are very distinct from blebbisome release, outside of the presence of MVBs, as these appears to be mechanisms for specific release of MVB while the presence of MVBs is not a necessary requirement for blebbisome formation nor does every blebbisome contain an MVB. The reviewer also mentions the ejection of nuclei from cancer cells. We are not of a mind, and it is not our intention to claim, that blebbisomes are the only cellular mechanism that can be responsible for release of a cellular organelle. Based on our reading of the literature blood platelets are specialized cells that prevent bleeding and minimize blood vessel injury. Large progenitor cells, megakaryocytes, in the bone marrow are the source of platelets. As blebbisome release is a phenomenon observed, from many normal and cancer cells, we do not believe, to the best of our current knowledge, that blebbisome release is related to blood platelets, though at this point in time we cannot rule out that some of the same basic molecular machinery required for platelet formation is also utilized for blebbisome formation. For example, the cytoskeleton is required to form platelets, as well as for other large cellular reorganization events, and we have in this revised manuscript documented a role for the specific cytoskeletal protein NMIIIB, and by their very nature blebbisome formation, like platelets, required cytoskeleton reorganization.

Reviewer #2:

Remarks to the Author:

In their revised manuscript, Jeppesen and colleagues have provided additional information to characterize blebbisomes, a newly described subset of large extracellular vesicles. They now show that blebbisome formation is affected by knockdown of myosin II but not ESCRT proteins providing some insight into the mechanisms of blebbisome formation. They also provide additional data demonstrating potential roles in immune suppression, and suggest the presence of blebbisomes in bone marrow. These are all interesting findings.

Specific points.

It is appreciated that the authors have provided some evidence to differentiate blebbisomes from migrasomes. However this is described far better in the rebuttal than in the manuscript itself. If the formation of migrasomes in the cells they examine is a rare event, they should state as much. Show quantification of the numbers of blebbisomes relative to migrasomes.

We appreciate the reviewer's suggestion to better quantify the frequency of retraction events that lead to the formation of blebbisomes compared to migrasomes. In our initial assessment, which we conducted visually and reported in our rebuttal, it appeared that more retraction events resulted in the formation of blebbisomes than migrasomes. However, upon performing a detailed quantification, we found that there is no statistically significant difference in the frequency of retraction events that generate blebbisomes versus migrasomes. This quantification has now been included in the revised manuscript as a new Extended Data Fig. 9a and reproduced here below for convenience.

Extended Data Fig. 9

Additionally, we have provided a new graph demonstrating that, unlike migrasomes, where multiple structures can form behind a cell, only a single blebbisome is generated per retraction event (Extended Data Fig. 9b). It was an oversight in our previous revision not to explicitly state that. Based on our observations, we have never detected multiple blebbisomes forming from a single cell which contrasts with what is known about migrasome formation in the literature.

Similar effort to distinguish blebbisomes from large oncosomes will also strengthen this study. This does not necessarily require separation of subpopulations of large EVs and generation of pure populations of each EV type, as the authors suggest. What about labeling cells forming blebbisomes for proteins unique to oncosomes (Minciacchi et al, 2025)?

We agree with the reviewer that comparing blebbisomes to large oncosomes will indeed strengthen our study. Both structures overlap in size and are derived from the plasma membrane and cytoplasm. Given this, we were not surprised that our proteomic analysis revealed that blebbisomes and large extracellular vesicles (IEVs), which would include any large oncosomes if present, were closely related. To address this point more thoroughly, we now present a new main figure (Fig. 3) and a new supplementary figure (Extended Data Fig. 5) directly comparing individual blebbisomes to oncosomes. Both figures reproduced here below.

Fig. 3

Extended Data Fig. 5

Initially, we did not delve deeply into the study of large oncosomes, as they form from membrane blebs (similar to microvesicles) whereas blebbisomes do not. Additionally, large oncosomes themselves do not exhibit blebbing. However, we acknowledge that the conclusion that large oncosomes do not bleb was based on published data rather than our own experiments.

To address this, we now present our own data from purified preparations of large oncosomes and blebbisomes. Consistent with the literature, our findings confirm that oncosomes do not exhibit membrane blebbing (Fig. 3a). Moreover, we localized several proteins using immunofluorescence that our proteomic analysis had identified as shared between large oncosomes and blebbisomes. The most intriguing of these was TUFM, a mitochondrial protein. This localization suggested the presence of active mitochondria in large oncosomes (Fig. 3c,d); however, TMRE staining revealed negligible activity, indicating that the mitochondria in oncosomes are non-functional and/or damaged (Fig. 3e, f).

In addition, we have quantitatively compared other proteins reported to be enriched in large oncosomes using both proteomics on purified blebbisomes and IEVs (including

large oncosomes) (Extended Data Fig. 5a) and immunofluorescence on purified blebbisomes and purified large oncosomes (Extended Data Fig. 5b,c). Most results were consistent across both techniques, however, we found that large oncosomes had more myosin IIA (MYH 9) based on proteomic data, but less based on immunofluorescence. A similar observation was made for CD63. This discrepancy is likely due to the proteomics data being derived from mixed populations of IEVs while the immunofluorescence data is based on purified large oncosomes that are specifically in the 1-10 um size range.

While these results demonstrate confirm clear differences between blebbisomes and large oncosomes, we acknowledge that there remains a possibility of a relationship between the two structures. As such, we have added the following statement to the Discussion section:

"Given their similarity in size, we have considered the possibility that large oncosomes might represent inactive or dead blebbisomes. While further investigation is required to fully explore this hypothesis, there are two key reasons why this relationship seems unlikely. First, blebbisomes lose membrane integrity when they undergo cell death and do not round up and detach from the cell to float into the media, as we might expect if they were to transform into large oncosomes. Second, blebbisomes do not form from plasma membrane blebs, unlike large oncosomes and microvesicles."

Figure 3. Mitochondria, autophagosomes, multivesicular bodies, intact Golgi stacks and electron dense lysosomes/residual bodies are readily identified by TEM. This is not the case of other vesicular and tubular elements in cells (early, sorting or recycling endosomes, vesicular and tubular ER and Golgi structures). Further, how frequently are all these organelles identified in blebbisomes? Quantitation should be shown.

We appreciate the reviewer's suggestion to quantify the frequency of specific organelles within blebbisomes. In response, we have conducted immunofluorescence-based quantification of several organelles. With the exception of the Golgi apparatus, which was absent in a small subset of blebbisomes (10%), all other examined organelles were consistently present in all blebbisomes analyzed. This quantification is now included in a new Table 1, reproduced here below.

Table 1: Percent of blebbisomes positive for specific organelles

Blebbisomes were stained for multiple organelle markers and assessed for the presence of each marker. $n = 3$ independent experiments with 10 blebbisomes assessed per n .

Organelle	n	Blebbisomes per n	% with each organelle	95%CI_lo	95%CI_hi
Golgi (GM130)	3	10	90	0.9	0.9
Mitochondria (TOM20)	3	10	100	1	1
Peroxisome (PEX14)	3	10	100	1	1
Ribosome (RPS8)	3	10	100	1	1
MVE/Exosome (CD63)	3	10	100	1	1

We acknowledge that further characterization is necessary to fully understand the functionality of these organelles within blebbisomes. Ongoing work in our lab is focused on investigating how membrane trafficking in blebbisomes compares to that in cells, particularly in terms of organelle dynamics and potential functional differences.

The authors suggest that blebbisomes are found in bone marrow. The data however are weak in this regard. How does one distinguish the blebbisome in the image and video from a blebbing BM cell? Are blebbisomes detected in biofluids?

We apologize for not fully explaining the rationale behind the bone marrow data in our revised manuscript. To differentiate cells from blebbisomes, we labeled both nuclei and mitochondria. Cells contain both a nucleus and mitochondria, whereas blebbisomes contain only mitochondria. There are, however, certain cells in bone marrow, such as red blood cells, that lack both a nucleus and mitochondria. After a thorough literature review, we found no reports of enucleated cell types that exhibit continuous blebbing. However, we acknowledge that such a cell type could theoretically exist. Therefore, in this revision, we refer to these structures as "blebbisome-like" extracellular vesicles.

The updated manuscript text reads as follows:

“Therefore, we extracted bone marrow from mice and immediately performed blebbisome purification. Several cell types in bone marrow are relatively small and in the same size range as blebbisomes. As such, we needed to be able to distinguish between cells and large blebbisome-like EVs. Therefore, we labeled mitochondria, as they are present in both blebbisomes and cells, as well as nuclei which only cells have (Fig. 7c). Of note, red blood cells contain neither mitochondria or nuclei but can be distinguished by their concave appearance in DIC and lack of membrane blebbing (Fig. 7c). Blebbisome-like EVs were identified by their characteristic blebbing nature as well as a positive signal for mitochondria and negative signal for nuclei.”

The authors state—“Additionally, we have uncovered evidence that blebbisomes can undergo apoptosis with concurrent generation of EVs, highlighting that not only cells but

also blebbisomes may be sources of apoptotic EVs.” What is the evidence for this? That blebbisomes are sensitive to apoptosis-inducing reagents does not mean that they are a source of apoptotic EVs. Are apoptotic EVs found in or released from blebbisomes under normal conditions?

We have removed the speculative statement in the Discussion suggesting that blebbisomes may produce apoptotic bodies and EVs, as we did not have sufficient time to fully investigate this possibility.

ADDITIONAL COMMENTS PROVIDED ON THE AUTHORS' RESPONSES

I've read the author's response and for the most part it seems reasonable. As indicated by both reviewers, it is important that the authors demonstrate how the “blebbisome” is distinct from (or similar to) other similarly sized EVs such as oncosomes and migrasomes. This is a critical point that needs to be addressed. They showed some data to suggest that the blebbisome is a distinct structure from migrasomes. Their additional findings that strengthen this point should also be shown.

We have now provided data quantifying the number of migrasomes formed during cellular retraction events (Extended Data Fig. 9, see above). A previous study by Jiao et al. (Cell, 2021, figure below) presented convincing evidence that migrasomes formed from control cells do not contain mitochondria. This study further demonstrated that a subset of migrasomes could contain damaged mitochondria if they originated from cells with damaged mitochondria.

[REDACTED]

Given these distinctions, we focused much of our second revision efforts on a detailed comparison between blebbisomes and large oncosomes, particularly regarding

mitochondrial functionality. While mitochondrial proteins have been reported in large oncosomes, it remains unclear if these mitochondria are functional. Our new data suggests that mitochondria in large oncosomes are non-functional/damaged (Fig. 3, see above)

Similarly, it would be helpful to morphologically distinguish nascent blebbisomes from oncosomes. Do proteins enriched in blebbisomes (based on their proteomics data) localize to emanating oncosomes? Conversely, do oncosome specific proteins localize to blebbisomes? They could use IF or live cell imaging or EM. They may need to use other cell types, such as prostate cancer cells that secrete oncosomes in abundance to address this point.

Please see the new data from IF experiments, and discussion of our comparison between large oncosomes and blebbisomes above. We found that MDA-MB-231 breast cancer cells produce an abundance of both blebbisomes and large oncosomes and therefore used these for our experiments.

The "blebbisome" described here is similar to the large EV described by Petersen et al. (2023). The current study characterizes it further, so distinguishing it from other larger EVs is important, and some solid insight into the biogenesis or function significantly strengthens the study.

The paper describes a "large EV" that is considerably smaller than blebbisomes and does not exhibit blebbing or contain mitochondria (Petersen et al. The Journal of Extracellular Biology 2023). Multiple of these EVs form at a time from the plasma membrane of HUVEC cells making them more similar to migrasomes than to blebbisomes, as only a single blebbisome is formed at a time. However, like blebbisomes, these EVs appear to contain multivesicular endosomes (MVEs), and we have now cited this study in our manuscript to inform the reader.

The study by Petersen et al. also claims that these EVs can secrete exosomes based solely on electron microscopy data showing vesicles within the EV that appear docked to the membrane (Figure is shown below for convenience). No experimental evidence is provided to directly test the hypothesis that these EVs can actively secrete vesicles or other contents.

[REDACTED]

REV#3 Comments to the Authors

The revised manuscript by Jeppesen et al provides additional valuable data characterising blebbisomes (BLS), a new class of complex extracellular vesicles the authors have recently discovered and characterized. Certainly, some of the mitochondrial death-like processes affecting BLS along with their previously described properties are quite fascinating.

The authors also provide a comprehensive rebuttal of editorial and reviewers' critique, and one understands that some of the questions raised previously may take a larger effort to be comprehensively addressed. This said, for a submission to a leading cell biology journal the fundamental properties of BLS, such as key molecular biogenetic mechanisms, distinguishing features compared to other large and complex EVs (e.g. migrasomes or large oncosomes), biological significance and function of BLS in vitro and in vivo should not be "out of scope" of the present paper.

No previous single study reporting a novel EV species has presented a description, purification scheme, proteomic analysis, mechanisms of formation, cellular function, and in vivo relevance. An example of this is the discovery of exophers, reported in Nature, which did not contain purification of exophers, did not describe the biogenesis of exophers, nor did it contain a biochemical characterization of exophers. This appears to be what the reviewer is asking for. It has taken multiple manuscripts—all in leading journals—to get such insights about migrasomes alone. Here we present data presenting description, purification, proteomic analysis, mechanisms of formation, and show that blebbisomes themselves have multiple cellular-like functions. We also show that blebbisomes can be purified from bone marrow. Although, we are willing to refer to these structures as blebbisome-like, as they bleb (DIC data), have mitochondria (fluorescent data), and lack a nucleus (fluorescent data), but we are not presenting a full characterization of them. As outlined below, we have also now added new data comparing blebbisomes to large oncosomes. We clearly demonstrate that, unlike any other type of EV described so far, only blebbisomes contain functional mitochondria. And functionally, we clearly demonstrate that blebbisomes can themselves secrete smaller EVs, and also take up smaller EVs from its environment. The only thing we can be said to be lacking in the current version is a function in vivo, that is, why cells release blebbisomes in the first place. Likely, as for every other type of EVs, numerous in vivo functions will eventually be uncovered.

While the authors made some attempts to address these questions, not all of these efforts were entirely convincing. It is true that some of the preparative technologies are not fully developed, for instance to purify large oncosomes (LO) to homogeneity, but it is difficult to agree that these are unsurmountable challenges. For example, LOs are up to 10 um in size and therefore should be readily separable from other large EVs in systems where they are produced (Di Vizio et al 2009).

We now present a new main figure (Fig. 3) and new supplementary figure (Extended Data Fig. 5) to directly compare individual purified blebbisomes and purified large oncosomes (please see also detailed response to Reviewer 2 above). Initially, we did not conduct an in-depth investigation into large oncosomes, as they originate from membrane blebs, unlike blebbisomes. Additionally, large oncosomes do not exhibit blebbing. However, we recognize that our conclusion about the absence of blebbing in oncosomes was based on previously published data rather than our own findings.

To remedy this, we now provide our own data from purified preparations of large oncosomes and blebbisomes. Consistent with the literature, our data confirm that large oncosomes do not exhibit membrane blebbing (Fig. 3a). Furthermore, we localized several proteins through immunofluorescence that our proteomic analysis identified as

shared between large oncosomes and blebbisomes. This was per reviewer 2's request. The most notable of these was TUFM, a mitochondrial protein. While its localization initially suggested active mitochondria in oncosomes, TMRE staining indicated minimal activity, suggesting that mitochondria in oncosomes are nonfunctional (Fig. 3e, f).

Additionally, we performed quantitative comparisons of other proteins reported to be enriched in large oncosomes using both proteomics and immunofluorescence, included in new Extended Data Fig 5.

While these results underscore clear distinctions between blebbisomes and large oncosomes, we acknowledge the possibility of a relationship between these structures. Therefore, we have added the following statement to the Discussion:

"Given their similarity in size, we have considered the possibility that large oncosomes might represent inactive or dead blebbisomes. While further investigation is required to fully explore this hypothesis, two key observations suggest that this relationship is unlikely. First, blebbisomes lose membrane integrity upon cell death and do not detach to float in the media as intact structures, which we would expect if they transformed into large oncosomes. Second, blebbisomes do not form from plasma membrane blebs, unlike large oncosomes and microvesicles."

Similarly, enrichment of cells expressing TSPAN4 in migrasomes suggests a causative effect of this tetraspanin, and an equivalent of this molecular mechanism for BLS would be quite informative (rather than "misleading"). Does expression of TSPAN4 affect BLS?

We appreciate the reviewer's suggestion regarding TSPAN4. The term "misleading" was used in our discussion with the reviewer and does not appear in the manuscript. Given our focus on comparing blebbisomes to large oncosomes, we did not have sufficient time to explore the potential role of specific tetraspanins, such as TSPAN4, in blebbisome formation. Our intent was to convey that the overexpression of TSPAN4 has led to the assumption that all cancer cells produce large amounts of migrasomes which may not be the case. TSPAN4 is not highly expressed across all cancer cell types.

Exploring how TSPAN4 expression might influence blebbisome formation could potentially be an interesting question. However, based on our proteomic data and previously published datasets, TSPAN4 is not enriched in cells, blebbisomes, or large EVs; it is only enriched in small EVs. Studies on migrasomes have used overexpression of TSPAN4 to artificially induce release of migrasomes. As a result, we have no reason to prioritize TSPAN4 in overexpression experiments to investigate molecular mechanisms of blebbisome formation as we deemed it to be unlikely.

As requested, we have already undertaken initial steps to identify relevant formation mechanisms, which are now included in the previous revised manuscript (specifically,

NMIIB, CHMP2A, and CHMP4B). These results clearly indicate involvement of myosin IIB but not ESCRT machinery in the biogenesis of blebbisomes.

In this regard silencing of NMIIB looks promising, but one would wonder whether such a generic targeting of the myosin system would not affect other large EVs, including migrasomes.

We appreciate the reviewer's insight regarding the specificity of targeting myosin IIB. Knocking down myosin IIB is a specific perturbation of the myosin II contractile system rather than a generic one. Most non-muscle cells express both myosin IIA and myosin IIB, which are typically organized into heterofilaments. Importantly, while knocking down myosin IIB does not prevent myosin IIA from forming filaments, reducing myosin IIA has been shown to decrease myosin IIB filament formation, as small filaments of myosin IIA may serve as templates for myosin IIB.

Additionally, myosin IIB is specifically implicated in driving large cellular retractions, which are associated with blebbisome formation. Myosin IIA, on the other hand, primarily generates cortical tension and intercellular pressure. We considered knocking down myosin IIA as well, but it is essential for both the formation and retraction of membrane blebs. Similarly to cells, without myosin IIA, blebs would likely not form in blebbisomes which would hinder our ability to identify blebbisomes, as bleb formation is a primary characteristic of blebbisomes. Given the scope of the revisions, myosin IIA did not rise to the top of our priority list.

Similarly, figure 6 of the revised manuscript continues to be centered on immune checkpoints expressed in BLS, in spite of complete absence of data that BLS possess any immunoregulatory functions. The latter may be difficult to conceptualize given the rarity of these EVs and absence of data that they interact with immune cells. This could have been easily tested. It is also not convincing to say that this question is a subject of an ongoing separate project and will be developed under a separate manuscript. If this is the case, why is fig. 6 included in the present manuscript? Ultimately, the existence and some sort of significant biological impact of BLS in vivo, or their formation as a reflection of some deeper cellular process are presumably the reasons why BLS should be regarded as more than cellular curiosity.

We appreciate the reviewer's concern regarding the inclusion of immune checkpoint data in Figure 6. In our study, we systematically identified and compared immune checkpoint protein expression not only in blebbisomes but also across cells, large EVs (lEVs), and small EVs (sEVs). This comprehensive comparison of immune checkpoint expression across different EV types represents very novel data that provides highly valuable insights into the unpublished presence and abundance of these proteins across a multitude of different EV types, and suggests important immunoregulatory

potential of these vesicles that points to many future avenues of study. Omitting this highly informative data would, in our view, would be a disservice to EV researchers interested in understanding the diverse functional roles of EV subtypes.

The newly presented pH-induced 'death' of BLS is quite intriguing, but of uncertain biological significance. Here again, as in other segments of the present study, the absence of a compelling reference point, such as another type of large and complex EVs (migrasomes) that may undergo similar fate under comparable conditions impacts the interpretation. Including these sorts of head to head comparisons, would make the present paper, as interesting as it is, still more convincing and mature.

We would like to clarify that we did not present data in the original revised manuscript or the current revision demonstrating that pH changes induce blebbisome death. This was discussed with the reviewer in the rebuttal to provide context on some preliminary observations. Specifically, we had noted in a separate experiment that a drop in pH appeared to cause blebbisome death, as blebbisomes happened to be in the field of view during an unrelated experiment.

In the current manuscript, we focus solely on demonstrating the effects of known apoptosis-inducing small molecules on blebbisomes. The observation of programmed death, a cell-like ability, in any type of extracellular vesicle (EV) is a very novel discovery with significant implications for the EV and cell biology fields. Obviously, future in vivo studies will be necessary to elucidate the biological relevance and specific functions of blebbisomes under different physiological conditions.